# Environmental drivers of increased ecosystem respiration in a warming tundra

S. L. Maes[1,2 ✉], J. Dietrich[1], G. Midolo[3], S. Schwieger[1,4], M. Kummu[5], V. Vandvik[6,7], R. Aerts[8], I. H. J. Althuizen[7,9], C. Biasi[10,11], R. G. Björk[12,13], H. Böhner[14], M. Carbognani[15], G. Chiari[15], C. T. Christiansen[16,17], K. E. Clemmensen[18], E. J. Cooper[19], J. H. C. Cornelissen[8], B. Elberling[17], P. Faubert[20], N. Fetcher[21], T. G. W. Forte[15], J. Gaudard[6,7], K. Gavazov[1,22], Z. Guan[23], J. Guðmundsson[24], R. Gya[6,7], S. Hallin[18], B. B. Hansen[25,26], S. V. Haugum[6,27], J.-S. He[23,28], C. Hicks Pries[29], M. J. Hovenden[30,31], M. Jalava[5], I. S. Jónsdóttir[32], J. Juhanson[18], J. Y. Jung[33], E. Kaarlejärvi[34], M. J. Kwon[35,36], R. E. Lamprecht[37], M. Le Moullec[26,38], H. Lee[39,40], M. E. Marushchak[37], A. Michelsen[16], T. M. Munir[41], E. M. Myrsky[42,43], C. S. Nielsen[17,44], M. Nyberg[30], J. Olofsson[4], H. Óskarsson[24], T. C. Parker[45], E. P. Pedersen[1,16], M. Petit Bon[46,47], A. Petraglia[15], K. Raundrup[38], N. M. R. Ravn[16], R. Rinnan[48], H. Rodenhizer[49], I. Ryde[16,17], N. M. Schmidt[50,51], E. A. G. Schuur[49,52], S. Sjögersten[53], S. Stark[42], M. Strack[54], J. Tang[55], A. Tolvanen[56], J. P. Töpper[57], M. K. Väisänen[42,58], R. S. P. van Logtestijn[8], C. Voigt[10,36], J. Walz[1], J. T. Weedon[8], Y. Yang[59], H. Ylänne[60], M. P. Björkman[12,13], J. M. Sarneel[4] & E. Dorrepaal[1]

Arctic and alpine tundra ecosystems are large reservoirs of organic carbon[1,2]. Climate warming may stimulate ecosystem respiration and release carbon into the atmosphere[3,4]. The magnitude and persistency of this stimulation and the environmental mechanisms that drive its variation remain uncertain[5–7]. This hampers the accuracy of global land carbon–climate feedback projections[7,8]. Here we synthesize 136 datasets from 56 open-top chamber in situ warming experiments located at 28 arctic and alpine tundra sites which have been running for less than 1 year up to 25 years. We show that a mean rise of 1.4 °C [confidence interval (CI) 0.9–2.0 °C] in air and 0.4 °C [CI 0.2–0.7 °C] in soil temperature results in an increase in growing season ecosystem respiration by 30% [CI 22–38%] ($n = 136$). Our findings indicate that the stimulation of ecosystem respiration was due to increases in both plant-related and microbial respiration ($n = 9$) and continued for at least 25 years ($n = 136$). The magnitude of the warming effects on respiration was driven by variation in warming-induced changes in local soil conditions, that is, changes in total nitrogen concentration and pH and by context-dependent spatial variation in these conditions, in particular total nitrogen concentration and the carbon:nitrogen ratio. Tundra sites with stronger nitrogen limitations and sites in which warming had stimulated plant and microbial nutrient turnover seemed particularly sensitive in their respiration response to warming. The results highlight the importance of local soil conditions and warming-induced changes therein for future climatic impacts on respiration.

The fate of the globally important carbon (C) stock of the arctic and alpine tundra biome (hereafter referred to as 'the tundra') will be determined by the balance between climatic impacts on C uptake (plant photosynthesis) and release (ecosystem respiration (ER)[2,9–12]; that is, the sum of plant or autotrophic respiration and microbial or heterotrophic respiration[13]. As the tundra stores vast amounts of carbon and is warming at rates higher than the global average[14,15], the consequences of global warming for C release from this biome have been a topic of hot debate over the past decades[4,11,16,17]. The overall magnitude, persistency and drivers causing variability of the ER increase with warming, however, remain highly uncertain, especially for multidecadal timescales. This is due to large variability in the ER response to experimental warming across single- or multisite studies (for example, increases of 8–52%; refs. 5,9,18–20), as well as poor representation of tundra sites in multisite meta-analyses of respiration[5,6,9,21,22].

## Indirect warming effects

Variability in the ER response to warming among tundra sites can be caused by differences in direct effects of warming on ER, for instance through changing kinetic rates of the underlying biological processes, as well as by indirect effects, through differences in warming-induced changes in abiotic (for example, microclimate[23], soil conditions[24,25] and permafrost thawing[26,27]) or biotic (vegetation and decomposer community composition[25,28–30]) conditions. For example, warming-associated soil drying can increase heterotrophic respiration by increasing oxygen

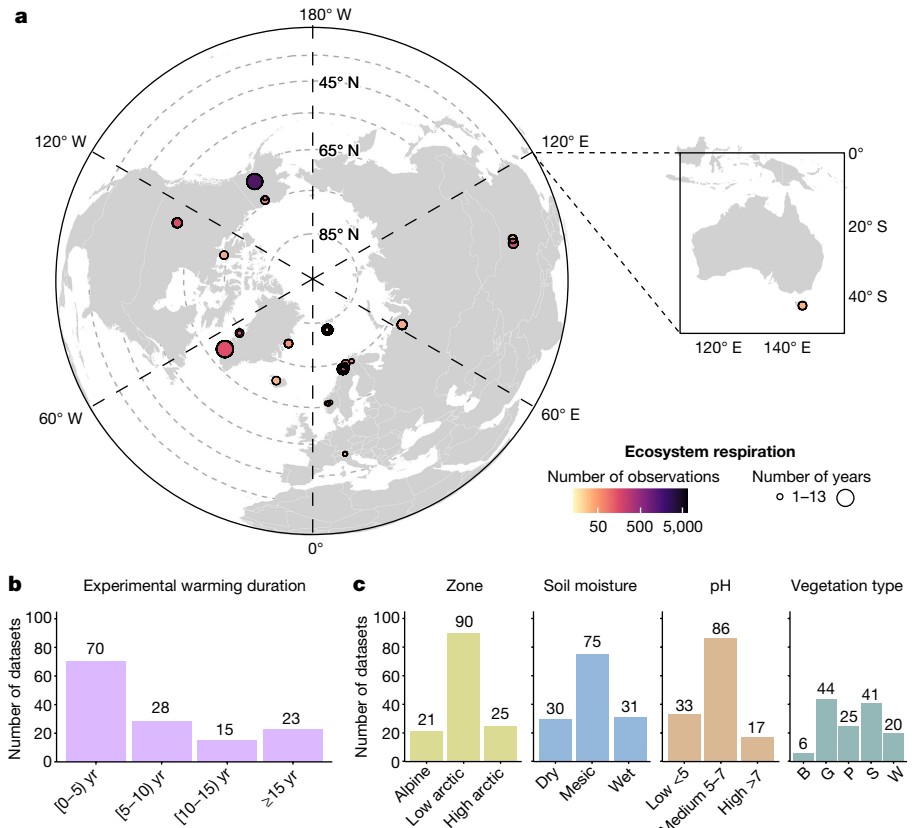

**Fig. 1 | Study area, duration and environmental context of warming experiments across the tundra biome. a–c**, ER was collected in 136 growing season measurement years across 56 independent OTC warming experiments, covering 28 distinct geographical sites across the tundra biome (**a**), capturing a wide range of experimental warming durations at the time of ER measurements (**b**) and environmental heterogeneity (**c**). The radius of the dots in **a** indicates the number of years in which ER was measured per experiment (range 1–13 years; 63% of experiments with range greater than 1 year), the colour of the dots indicates the number of observations per dataset, that is, number of ER measurements in the growing season across all years in which measurements

were conducted (details in Extended Data Fig. 1, Supplementary Methods 1 and Supplementary Tables 1 and 2). The distribution of datasets is shown across the duration of experimental warming before ER measurements were taken, grouped into four classes (duration = ER measurement year – OTC treatment start year; note that '0' refers to the first summer of experimental warming) (**b**) and categorical variables describing environmental contexts (**c**), from left to right: zone; soil moisture class; soil pH class; and vegetation type (B, barren; G, graminoid; P, prostrate shrub; S, erect shrub; W, wetland tundra). The maps were made with R.

supply for microbes in waterlogged or anaerobic soils, whereas it can decrease respiration by promoting water limitation of microbial metabolism in dry or aerobic soils[23–25]. Also, warming-driven changes in vegetation cover or composition may affect ER responses by altering the quantity of plant roots or the quantity and quality of litter, ultimately influencing autotrophic and heterotrophic respiration rates[28,30–32].

## Context-dependencies

Tundra ecosystems, covering a wide range of environmental conditions[33], are strongly heterogeneous in their climatic and soil conditions, as well as in their vegetation and microbial communities. ER responses to warming may vary as a result of these context-dependencies or context-specific environmental conditions[3,11,27,34,35]. For example, ER should theoretically increase more with warming in colder macroclimates[4,36–38] or in soils with greater soil organic carbon (SOC) pools, although this remains debated and probably depends on interacting factors such as soil moisture[5,11,39] or SOC lability[16,34]. Similarly, soil pH may cause variability in the ER response magnitude by influencing the availability of key nutrients for plants and decomposers[31,40]. Further, the vegetation[27,36,41,42] and microbial[27,29,40,42] community abundance or composition can affect the autotrophic respiration and heterotrophic respiration potential[27,40] and thus the ER response magnitude, through

differences in net primary productivity (NPP), litter quantity and quality and carbon-use efficiency.

Understanding such indirect warming effects as well as context-dependencies might explain the variability in ER responses to warming across the tundra. Further, it might elucidate important underlying mechanisms of the responses. Most multisite meta-analyses have focused strongly on the magnitude, direction and climatological drivers of the ER response[5,6,9,21,22], whereas understanding the ecological mechanisms behind the ER response is essential to accurately predict the long-term impact of climate change on the terrestrial carbon cycle[27,43]. To improve predictions of C emissions for the tundra[3,4,7,8,16], we urgently need better quantification and greater mechanistic understanding of how warming influences ER across the biome.

## Objective and study design

Here we aim to quantify with meta-analysis the magnitude and variability of the impact of warming on ER across the tundra and to understand with metaregression models how variation in the ER response depends on warming duration, indirect effects of warming and context-dependencies. We therefore collated 24,035 daytime, growing season ER measurements from 28 sites across the tundra biome, covering 56 passive in situ open-top chamber (OTC) warming experiments[44] across four continents, with warming duration ranging from

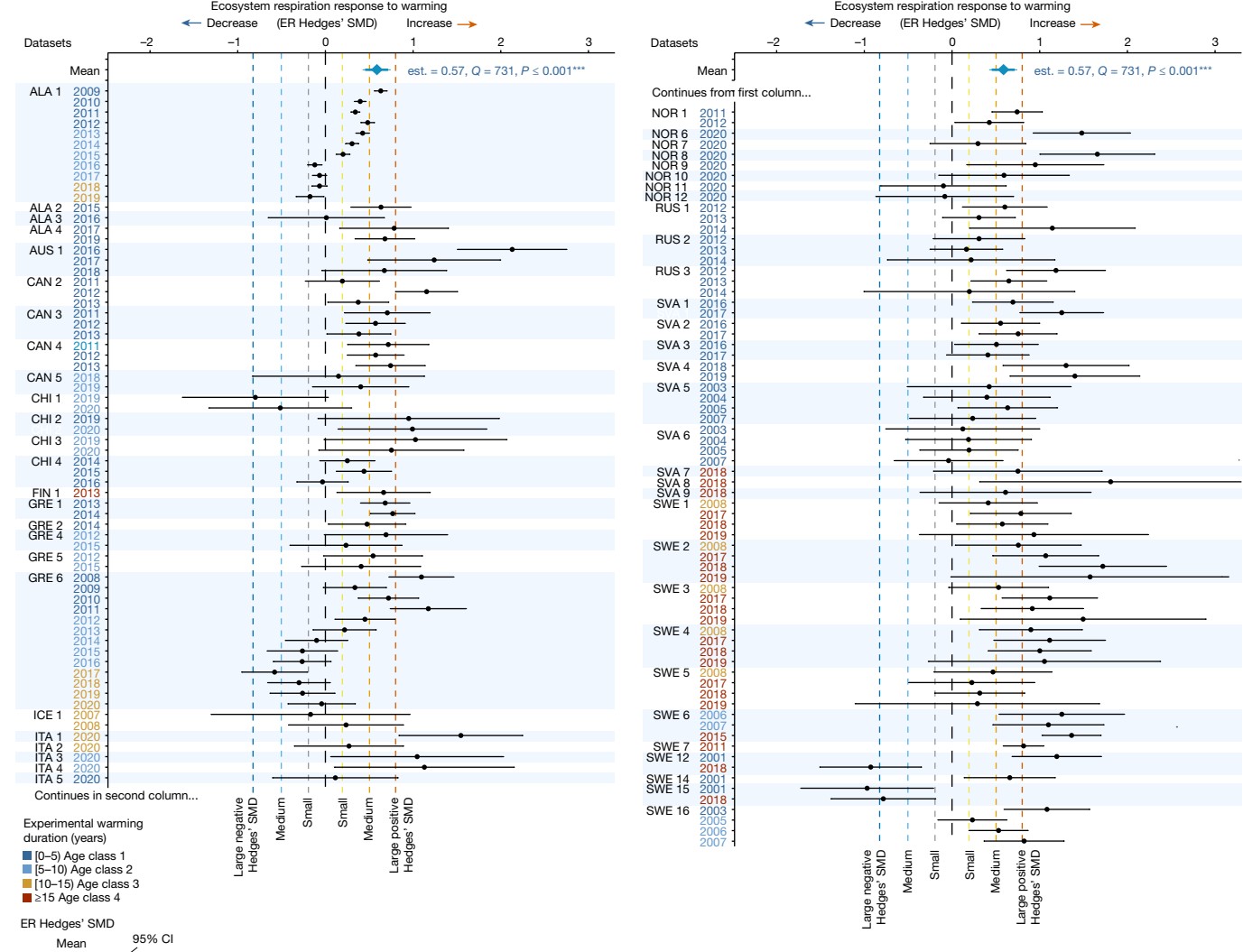

**Fig. 2 | Effects of experimental OTC warming on ecosystem ER.** Experimental warming increased ER across the tundra biome but the magnitude of the response varied across time and space. Effect of OTC warming on ER Hedges' SMD calculated as (mean ER of the warmed plots − mean ER of the control plots)/pooled standard deviation across the 136 growing season datasets (that is, unique experiment × ER measurement year combinations). On the top of the graph, a blue diamond shows the mean estimate (est. = 0.57 and 95% CI [0.44–0.70], error bars) of the ER response across the 136 datasets, as well as the *Q* value testing for heterogeneity and *P* value from the meta-analysis. Black dots represent ER Hedges' SMDs of individual datasets and 95% CIs (black error bars) in alphabetical and chronological order. Individual datasets are

represented by the experiment ID in black (left) and ER measurement year (right) in a colour scale ranging from dark blue, light blue, orange to red which represents increasingly longer warming duration at the time of ER measurements. Experiments with more than 1 year of ER data are grouped. See Supplementary Tables 1, 2 and 4 for details on the datasets and SMD and CI values. The black dashed vertical line (SMD = 0) represents no change in ER with warming whereas the areas to the right and left of it represent increased (SMD > 0) versus decreased (SMD < 0) ER with warming. Dashed vertical lines (*x* = 0.2, 0.5, 0.8 and −0.2, −0.5, −0.8) reflect small, medium and large positive and negative Hedges' SMD, that is, increasingly greater ER increases and decreases with warming.

one summer to 25 years at the time of ER measurements (Fig. 1a and Supplementary Table 1). No single study has assessed the ER response to in situ experimental warming yet across a large spatiotemporal environmental gradient in the tundra, with a standardized experimental setup while including also an extensive set of environmental drivers[9,11,45]. On the basis of previous studies, we expect an overall positive ER response to warming (both autotrophic respiration and heterotrophic respiration) across all experiments but significant variation in the magnitude of the response across time and space. We expect these differences in the magnitude to be driven (1) indirectly by variation in warming-induced changes in local environmental conditions, as well as (2) directly by context-dependent variation in environmental conditions. We focus on the growing season (June–August), which is when most ER measurements are taken and soils

are most active. We calculated the ER response as Hedges' *g* standardized mean difference (SMD)[46], the difference in mean growing season ER of the unmanipulated control and the warmed OTC plots, divided by the pooled standard deviation. ER Hedges' SMD values were calculated for every dataset, which was a distinct ER measurement year within an experiment and within a larger site, resulting in 136 ER SMD values (Fig. 1). Using a meta-analysis approach assigning higher importance to datasets with higher precision (that is, lower variance), we evaluated the magnitude and variability in the ER response to warming across the tundra biome. The long duration of warming at the time of ER measurements in several experiments (Fig. 1b and Supplementary Table 1) and the availability of repeated measurements in many of the experiments (up to 13 datasets or repeated ER measurement years per experiment; Supplementary Table 1) enabled

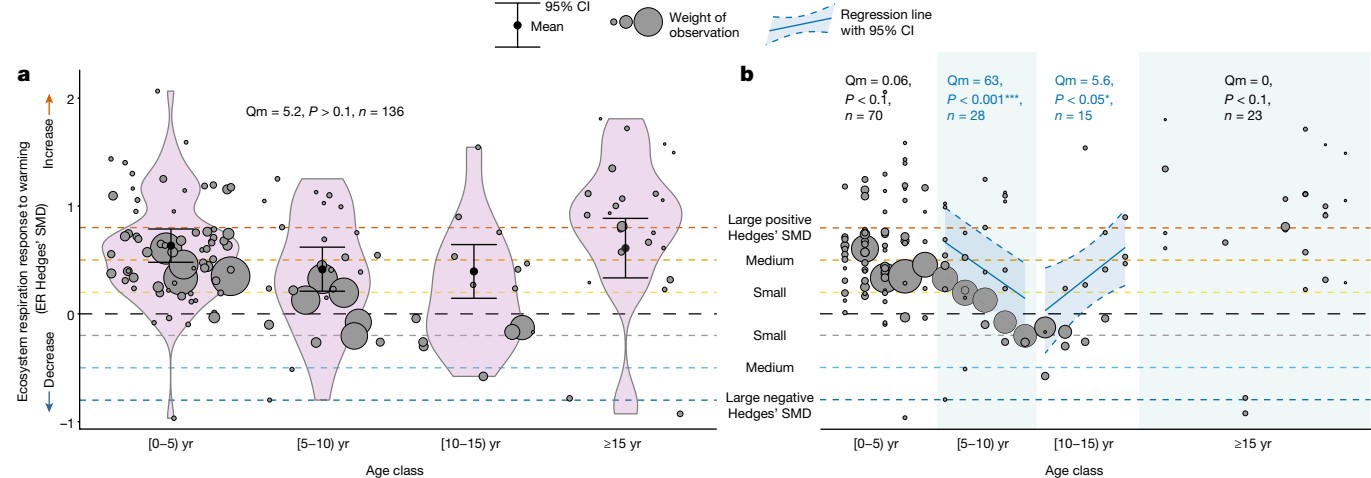

**Fig. 3 | Temporal patterns in ER responses with experimental warming duration.** ER responses to warming faltered around the end of the first decade of warming but did not wane in the long term. **a**, Mean ER response to warming (ER Hedges' SMD) across the four experimental duration age classes, showing the kernel density estimation of the underlying distributions in violin plots as well as the single-factor metaregression model estimates and 95% CIs (in black). **b**, Relation between experimental warming duration and ER response to warming across the four age classes (Extended Data Table 2) with 95% CI (light grey area). Hedges' SMDs for individual datasets (grey bubbles) are shown in **a** and **b**, calculated as (mean of the warmed plots − mean of the control plots)/ pooled standard deviation. Bubble size denotes the weight of the observation used in the metaregression model, quantified as the inverse of the square root of within-study variance, with greater bubbles indicating greater weights. Qm values, *P* values and sample sizes (*n*) are shown for the regression models, with Qm representing the *Q* value of importance of duration age class across all data in **a** and of experimental warming duration per age class in **b**. Significant regression lines in **b** are shown in blue. The black dashed horizontal line (SMD = 0) represents no change in ER with warming whereas the areas above and below represent increased (SMD > 0) versus decreased (SMD < 0) ER with warming. Dashed horizontal lines (*y* = 0.2, 0.5, 0.8 and −0.2, −0.5, −0.8) reflect small, medium and large positive and negative Hedges' SMD, that is, increasingly greater ER increases and decreases with warming. For detailed model output, see Extended Data Table 2.

us to evaluate whether and how the ER response to warming varied over time (Extended Data Table 2). Furthermore, the high number and wide geographic, climatic and habitat range of the included tundra sites (Fig. 1c), allowed us to analyse the drivers of variability in the ER response using metaregression (Extended Data Tables 3 and 4). These drivers included (1) changes in abiotic and biotic conditions induced by the warming treatment (that is, climatic, soil, vegetation and microbial community properties) and (2) the abiotic and biotic environmental context (that is, climatic, soil and vegetation community properties; Supplementary Methods 1). These two types of drivers allowed us to assess how the variation in the ER response magnitude depended on, respectively, indirect effects of warming as well as on context-dependencies. The database does not include gross primary productivity (GPP) data, as its inclusion in ER datasets is challenging because of the light-dependency of photosynthesis and associated high spatiotemporal variability of GPP measurements (Methods). However, by focusing on ER and accounting for GPP indirectly through NPP and plant biomass, our analysis incorporates the response of plant growth to warming in the ecosystem carbon balance.

## Experimental warming treatment

Warming with OTCs led to a mean 1.4 °C (95% confidence interval (CI) [0.9–2.0 °C]) (*P* < 0.001; *n* = 77) increase in growing season air temperatures, a mean 0.4 °C [CI 0.2–0.7 °C] (*P* < 0.001; *n* = 118) increase in growing season soil temperature and a mean 1.6% [CI 0.8–2.4%] (*P* < 0.001; *n* = 111) decrease in growing season soil moisture compared to ambient conditions (based on meta-analysis models; Supplementary Table 3 and Supplementary Methods 1). In line with previous studies[5,28,30], experimental warming also increased soil organic matter (SOM) content of the mineral layer, aboveground biomass and mean height of the vegetation community, whereas it decreased the lichen cover (Supplementary Table 3).

## Increased ecosystem respiration

As a result of the warming treatment, ecosystem respiration increased by a mean Hedges' SMD of 0.57 [CI 0.44–0.70] (*P* < 0.001) (*n* = 136), reflecting that 30% [CI 22–38] (*P* < 0.001) more $CO_2$ was respired, on average, in the experimentally warmed plots than in the unmanipulated controls (Fig. 2 and Extended Data Table 1). This mean magnitude is nearly four times greater than the previous 8% increase for the tundra, calculated in a meta-analysis based on 18 tundra sites[9]. Further meta-analyses of all data on partitioned ER into plant-related (autotrophic) and heterotrophic respiration available in all experiments included in this study (*n* = 9) demonstrated that both autotrophic and heterotrophic respiration increased significantly with warming (mean Hedges' SMD for autotrophic respiration = 0.44 [CI 0.08–0.80], *P* < 0.001 and for heterotrophic respiration = 0.92 [CI 0.36–1.48], *P* < 0.001; Extended Data Table 1 and Supplementary Discussion 1). On the basis of these limited data, the warming-induced increase in ER probably resulted from a significant increase in both plant-related and heterotrophic respiration, each contributing to the overall effect of warming on ER. Although the ER response to warming was overall positive and strong, there was significant heterogeneity across the datasets (*Q* value 731, *P* < 0.001; Fig. 2 and Extended Data Table 1), implying that ER responses to warming vary across time and/or space. The mean ER response remained positive and did not differ significantly across the four age classes (Fig. 3a), hence the overall positive warming effect persisted for the 2.5 decades timescale of these experiments. Investigating the temporal patterns further showed indications for nonlinear, positive responses over time (Fig. 3a,b and Extended Data Table 2): a decrease in the magnitude of the (still) positive ER response during 5–9 years of warming (moderator importance value or Qm = 63, *P* < 0.001, *n* = 28; Fig. 3b) was followed by an increasing magnitude during 10–14 years of warming (Qm = 5.6, *P* < 0.05, *n* = 15; Fig. 3b). Combined with the lack of change in the magnitude of the ER response over time during 0–5 and more than 15 years of warming,

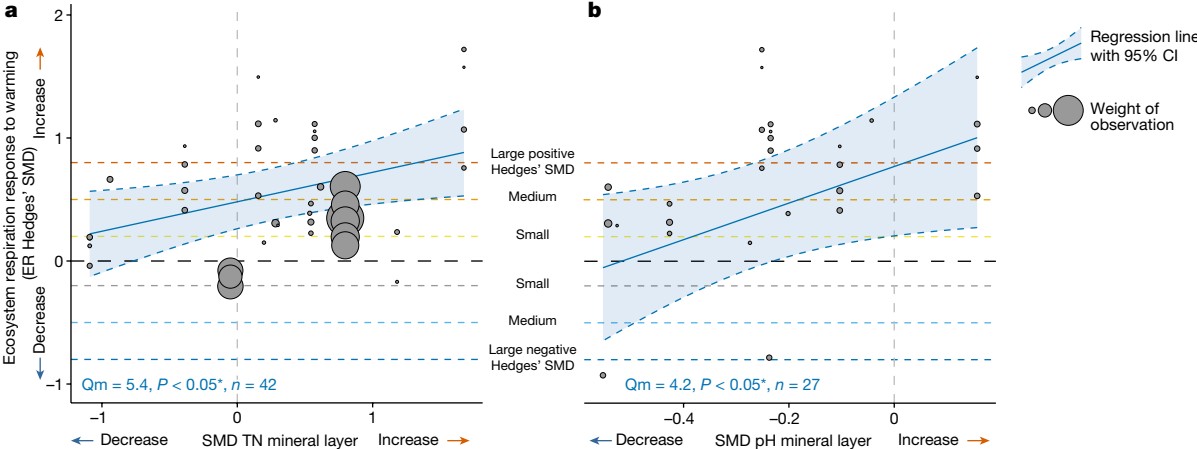

**Fig. 4 | Indirect warming effects on the ER response to warming.** **a**,**b**, Warming-induced changes in TN concentration (**a**) and soil pH of the mineral layer (**b**) affected the ER response to experimental warming. Warming-induced changes in TN and pH on the *x* axis are quantified as Hedges' SMDs of the soil conditions between the warmed and control plots, for example, (mean TN of the warmed plots – mean TN of the control plots)/pooled standard deviation. ER Hedges' SMDs for individual datasets are shown on the *y* axis with grey bubbles, calculated as (mean ER of the warmed plots – mean of the control plots)/pooled standard deviation. Bubble size denotes the weight of the observation used in the metaregression quantified as the inverse of the square root of within-study variance, with larger bubbles indicating greater weights. The significant regression lines with 95% CI are shown with blue lines and shaded areas, respectively. Bottom left in each panel shows the Qm (*Q* value of importance of the environmental drivers), *P* value of the metaregression model and the sample size (*n*, number of datasets). The black dashed horizontal line (*y* = 0) represents no change in ER with warming whereas the areas above and below represent increased (SMD > 0) versus decreased (SMD < 0) ER with warming. Dashed horizontal lines (*y* = 0.2, 0.5, 0.8 and −0.2, −0.5, −0.8) reflect small, medium and large positive and negative Hedges' SMD, respectively[50], or increasingly greater ER increases and decreases with warming. The black dashed vertical line (*x* = 0) reflects no change in the soil condition with warming (with areas right and left of it representing increased versus decreased conditions with warming). For detailed model output, see Extended Data Table 3.

the overall slope across all years was non-significant, indicating that ER continued to be positive in locations with longer warming histories (Fig. 3a and Extended Data Table 2). Although the ER response to experimental warming may thus falter around the end of the first decade of warming, our data do not provide evidence that ER responses to experimental warming wane in the longer term (that is, multidecadal duration).

## Indirect warming effects

Variation in the magnitude of the ER response to warming in the tundra was driven by variation in how warming indirectly changed soil conditions of the mineral layer, that is, of total N (TN) concentration and pH (Fig. 4 and Extended Data Table 3). In contrast, the variation in the ER response was not driven by differences in how warming changed the microclimate or the vegetation community structure (for example, biomass) and composition (for example, lichen cover) (Extended Data Table 3), despite warming inducing changes in the latter (Supplementary Table 3). Specifically, ER increased more with warming in experiments in which warming promoted greater increases in mineral layer TN concentration (Qm = 5.4, *P* < 0.05, *n* = 42; Fig. 4a) or caused smaller reductions in mineral layer pH (Qm = 4.2, *P* < 0.05, *n* = 27; Fig. 4b). Greater increases in TN concentration and smaller reductions in pH with warming suggest larger increases in N mineralization and N cycling with warming or stimulated microbial activity[5,47–49]. Combined with our ER partitioning results showing increased heterotrophic as well as autotrophic respiration with warming (*n* = 9), stimulation of both microbial and plant activity might lead to the observed increase in ER with warming (Extended Data Table 1 and Supplementary Discussion 1). This result is based on 9 out of the 136 datasets and should be verified with future large-scale ER partitioning studies. These nine datasets represent all the partitioning data that are now available from the 56 experiments, highlighting the methodological challenges in measuring ER partitioning in a long-term warming experiment.

## Context-dependent respiration responses

Surprisingly, variation in the ER response to warming was not related to any of the expected context-specific environmental drivers that reflect the tundra's heterogeneous climatic conditions or vegetation community composition. None of the climatic (for example, ambient mean temperature and permafrost probability) or vegetation community drivers (for example, vegetation class and NPP) influenced the ER response across the large spatiotemporal environmental gradient covered by the 56 experiments (Fig. 1, Extended Data Fig. 2 and Extended Data Table 4). Yet, our analyses did corroborate some expected context-dependencies in the ER response related to the soil conditions (Fig. 5, Extended Data Table 4 and Supplementary Methods 1). Nutrient-poor sites, that is, sites with lower TN concentration or with higher C:N ratios in the mineral layer, showed greater ER increases with warming than did nutrient-rich sites (TN: Qm = 6.3, *P* < 0.05, *n* = 43, Fig. 5a; C:N ratio: Qm = 4.7, *P* < 0.05, *n* = 39, Fig. 5b). Greater ER increases with warming in nutrient-poor sites may be linked to changes in stimulated belowground C allocation by the plants and subsequent soil priming by root leachates[47]. That is, warming has been shown to increase root production and thus labile C availability for rhizosphere organisms (that is, rhizosphere priming[47–49]). Especially in nutrient-poor conditions, this priming enhances decomposition, thereby increasing nutrient availability for plants (and potentially TN in the mineral layer) and ultimately stimulating autotrophic and heterotrophic respiration. This priming effect plays a larger role in these systems because of higher N-acquisition costs and thus greater 'return on investment' of priming, for plants in N-limited systems[47,48,50,51]. Thus, our results indicate that rhizosphere priming and the associated ER responses that come with the accelerated decomposition through priming, might depend on the degree of N limitation in tundra sites. Even though we did not find vegetation or microbial community drivers to influence the magnitude of the ER response, the importance of these soil parameters indirectly suggest that both plant and microbial processes play a role in the sensitivity of the ER response to warming.

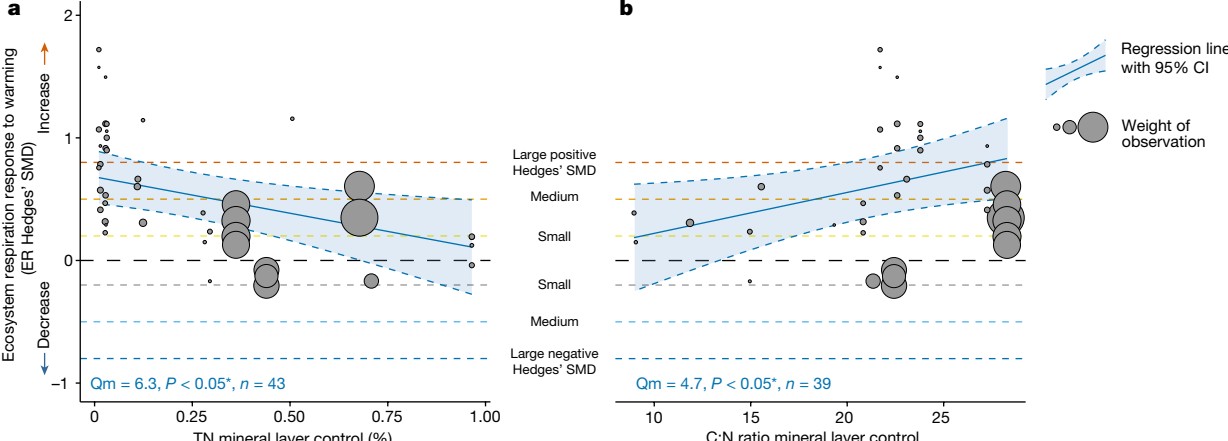

**Fig. 5 | Context-dependencies in the ER response to warming. a,b,** TN concentration (%) (**a**) and C:N ratio (**b**) of the mineral soil layer affected the ER response to experimental warming. The environmental drivers on the *x* axis reflect mean values of measured soil conditions at the control plots of each experiment with available data. ER Hedges' SMDs for individual datasets are shown on the *y* axis with grey bubbles, calculated as (mean ER of the warmed plots − mean of the control plots)/pooled standard deviation. Bubble size denotes the weight of the observation used in the metaregression quantified as the inverse of the square root of within-study variance, with larger bubbles indicating greater weights. The significant regression lines with 95% CI are shown with blue lines and blue shaded areas, respectively. Bottom left in each panel shows the Qm (*Q* value of importance of the environmental drivers), *P* value of the metaregression model and the sample size (*n*, number of datasets). The black dashed horizontal line (*y* = 0) represents no change in ER with warming whereas the areas above and below represent increased (SMD > 0) versus decreased (SMD < 0) ER with warming. Dashed horizontal lines (*y* = 0.2, 0.5, 0.8 and −0.2, −0.5, −0.8) reflect small, medium and large positive and negative Hedges' SMD, respectively[50], or increasingly greater ER increases and decreases with warming. For detailed model output, see Extended Data Table 4.

## Global implications of plant–soil linkages

The rate of warming in the tundra biome might be as much as 0.73 °C per decade (arctic tundra[14,15]), which greatly surpasses the global rate of 0.19 °C per decade[52]. The significant experimental warming of 1.4 °C [95% CI 0.9–2.0 °C] (*P* < 0.001; *n* = 77), that is, equivalent to two decades of regional warming across the tundra, increased growing season ER on average by 30% but with substantial variation in the magnitude of this response. Environmental drivers of variation in the ER response to warming were related to the extent to which warming had affected N-related soil parameters and to context-specific soil biogeochemistry in the mineral layer. Importantly, variation in these environmental drivers influenced the magnitude but not the direction of the ER response to warming, as the overall ER response remained positive.

Our results demonstrate the importance of local soil conditions in mediating future warming impacts on respiration, specifically of the mineral soil layer deeper in the soil profile. Experimental warming can disproportionately stimulate both microbial and plant root activity, and thus ER, in deeper or mineral soil layers[19,43,53] but only few studies have determined the depth at which most of the increase in ER originates because of methodological constraints. Our data show that the magnitude of the warming-induced ER increase in tundra sites is tightly linked to biogeochemical conditions, especially of the N cycle and changes therein, in the deeper mineral layer. This corroborates earlier findings that warming may have the strongest impact on biogeochemical conditions at depth[43], instead of at surface or subsurface organic soil layers. Particularly the relative availability of N in the mineral soil layer may determine warming-induced N turnover in tundra systems through plant–soil linkages involved in decomposition and rhizosphere priming processes[51,54–56]. We postulate that N-limited tundra sites, characterized by higher N-acquisition costs, are more responsive in their ER response to warming through stronger impacts of warming on decomposition and rhizosphere priming in these sites[47,57,58]. This is corroborated by the negative correlation between TN concentration in the mineral layer and the change in TN with warming (correlation coefficient = −0.43), indicating that N-limited sites also showed stronger stimulation of plant and microbial nutrient turnover under warmer conditions.

## Spatial patterns

By upscaling our metaregression results, which predict the ER response to warming on the basis of the significant context-dependent drivers (total N and C:N in the mineral layer, ratio of means (ROM) = 0.05–0.16 × TN + 0.01 × C:N, Qm = 6.7, *P* < 0.05, *n* = 39) (Fig. 5), we visualize spatial differences in the sensitivity of the ER response to warming across the arctic and circumarctic alpine region. Particularly sensitive areas to warming are the western and eastern parts of Siberia and parts of the Canadian arctic archipelago (Fig. 6). High uncertainty on these estimates (Fig. 6b), however, highlights an urgent need to address spatial sampling gaps in follow-up studies[59]. We also show that most of the uncertainty originates from the input data to the upscaling (Fig. 6c), that is, from the global gridded soil data (Methods), accounting for about 82% of total uncertainty. Considering these uncertainties, we can still use our model to provide an approximation of how much more C would be respired by 1.4 °C warming compared to present temperatures, across the tundra region (Fig. 6 and Supplementary Fig. 7). For the arctic tundra region alone, our upscaling exercise suggests that a temperature rise of 1.4 °C would enhance ER from 1.2 to 1.5 PgC yr⁻¹ (increase of 0.37 PgC yr⁻¹ with s.d. of 0.99 PgC yr⁻¹), which corresponds to an increase of 32% (s.d. = 85%). Across the arctic and alpine tundra regions combined, ER would increase from 3.4 to 4.3 PgC yr⁻¹ (increase of 0.86 PgC yr⁻¹ with s.d. of 1.36 PgC yr⁻¹) corresponding to an increase of 25% (s.d. = 40%).

Global and regional climate models which predict future C emissions require robust understanding of the mechanisms behind spatial differences in the ER response to warming. The mechanistic understanding provided by this study and based on field experimental data in real-world settings (for example, the relationship between soil nitrogen and pH and the ER response to warming), should therefore facilitate model development and improve predictions of changes in carbon cycling with future warming[3,7,8,16]. By estimating spatial differences in sensitivity of the ER response to warming, our results can be used to benchmark key components of the C cycle for tundra systems in these climate models. Our study focuses on growing season patterns but investigating how ER changes with warming across the whole year will also be important to improve climate models[60–62]. Because expansion

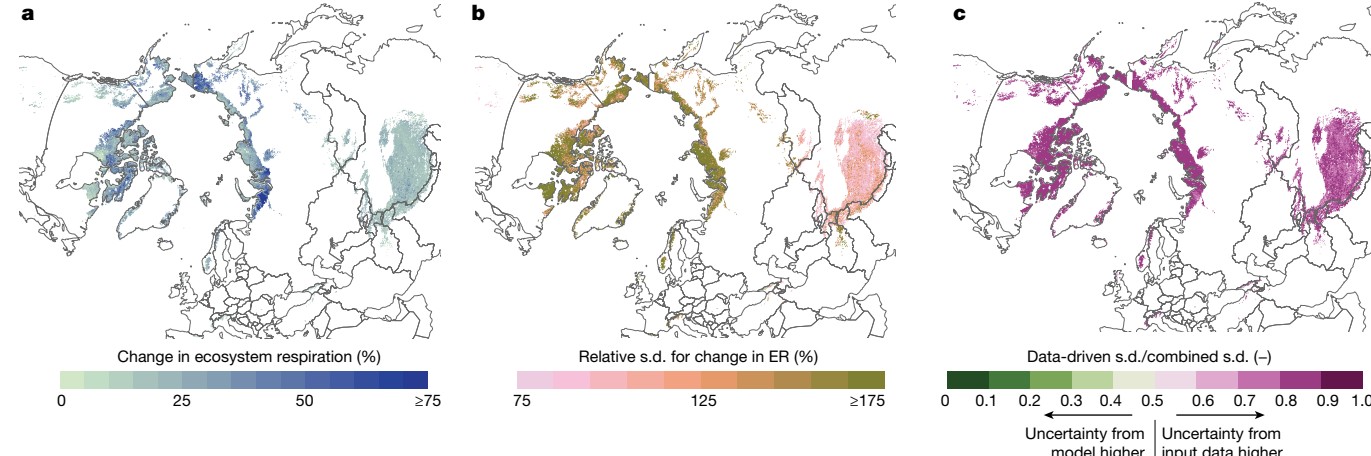

**Fig. 6 | Spatial patterns and uncertainty in the sensitivity of ER to warming across the tundra.** Maps showing spatial extrapolation of metaregression results for warming effects on ecosystem respiration (ER) and associated uncertainties across the arctic and circumarctic alpine tundra. **a**, Spatial distribution of predicted relative changes in ER resulting from 1.4 °C air warming. Values were obtained by combining the significant metaregression multifactor model with global gridded soil data for TN concentration and C:N ratio of the mineral layer (Methods). Plotted values are predicted mean values expressed as percentage change from present ER levels. For the whole region, ER increased from 3.4 to 4.3 PgC yr$^{-1}$ (increase of 0.86 PgC yr$^{-1}$ with s.d. of 1.36 PgC yr$^{-1}$) or by 25% (s.d. = 40%). **b**, Uncertainty of estimated ER changes expressed as coefficients of variation (standard deviations of model predictions divided by mean predicted values). These standard deviations are computed with Monte Carlo error propagation incorporating uncertainty associated with (1) metaregression model parameters and (2) soil database input values (Methods). **c**, Relative contribution of model parameter uncertainty versus input soil data uncertainty to Monte Carlo estimates of standard deviation. Error propagation was conducted twice, first combining both uncertainty sources as in **b** and subsequently by fixing metaregression parameters to their mean estimates and only allowing soil input data to vary. The plotted values show the ratio of the resulting standard deviations, that is, soil input uncertainty only/combined uncertainty. Colours closer to green indicate grid cells for which uncertainty in ER responses to warming are primarily driven by error in model metaregression model parameter estimates; colours closer to purple indicate grid cells for which ER prediction uncertainty is driven more by imprecision in soil database data. The maps were generated using Natural Earth (https://www.naturalearthdata.com/).

of the database across time and space will only further strengthen our understanding[27,59], we welcome contributions of ER measurements taken in new warming experiments across the tundra, or in new years for included experiments: http://www.tundrafluxdatabase.com/.

The magnitude of the long-term land carbon–climate feedback critically depends on the balance between C uptake through changes in photosynthesis activity and C release through changes in ER[1,43,63]. We focused on ER as a critical component in this balance and evaluated how ER might change with future warming. We found sustained ER increases across 25 years of experimental warming. Short-lived respiration increases to 5–10 years of experimental warming have been observed in several but not all single-site studies[5,9,22,64] as well as global[21] or high-latitude[16] meta-analyses and have in some longer term studies been followed by renewed respiration increases as the duration of warming continued[64]. We found a decreasing and increasing temporal pattern in the magnitude of the positive ER response to warming between 5–10 years and 10–15 years of warming, respectively, and no significant overall slope when evaluating across 25 years of warming (Fig. 3). These results highlight two key findings. First, in tundra ecosystems, ER remains enhanced with continued warming, at least up to 2.5 decades. Second, the ER response to warming shows a nonlinear trend over time between 5 and 15 years of warming, during which the positive respiration response drops in magnitude, after which it rises again. This nonlinear pattern of ER increase over time may be due to the underlying microbial and plant processes responding to warming at different rates, for example, ranging from more immediate effects of warming on microbial and plant respiration through accelerated decomposition versus slower effects through changes in biogeochemical and hydrological soil conditions as well as in microbial or vegetation community composition[7,43,64]. The latter is corroborated by the clear importance of changes in soil biogeochemical conditions as drivers of variation in the ER increase. Our results indicate that several of these slower, indirect mechanisms of warming impacts on ER, that is, indirect

effects of warming on ER through changes in soil nutrient availability, are already apparent during the time-span of up to 25 years of experimental warming studies in the tundra.

Finally, our ER partitioning analyses demonstrated that the ER increases with warming were probably driven by both increased plant-related and heterotrophic respiration[18,65,66]. Taken together, these findings imply that the large soil C stocks in tundra systems have the potential to transform from acting as a global C sink to a source if warming were to increase ER to such an extent that the extra C respiration cannot be compensated by equally strong increases in plant CO$_2$ uptake[43,65,67].

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

¹Climate Impacts Research Centre, Department of Ecology and Environmental Science, Umeå University, Abisko, Sweden. ²Forest Ecology and Management Group (FORECOMAN), Department of Earth and Environmental Sciences, KU Leuven, Leuven, Belgium. ³Department of Spatial Sciences, Faculty of Environmental Sciences, Czech University of Life Sciences Prague, Praha-Suchdol, Czech Republic. ⁴Department of Ecology and Environmental Science, Umeå University, Umeå, Sweden. ⁵Water and development research group, Aalto University, Espoo, Finland. ⁶Department of Biological Sciences, University of Bergen, Bergen, Norway. ⁷Bjerknes Centre for Climate Research, University of Bergen, Bergen, Norway. ⁸Amsterdam Institute for Life and Environment (A-LIFE), Vrije Universiteit, Amsterdam, The Netherlands. ⁹NORCE Climate and Environment, Norwegian Research Centre AS, Bergen, Norway. ¹⁰Department of Environmental and Biological Sciences, University of Eastern Finland, Kuopio, Finland. ¹¹Department of Ecology, University of Innsbruck, Innsbruck, Austria.

[12]Department of Earth Sciences, University of Gothenburg, Gothenburg, Sweden. [13]Gothenburg Global Biodiversity Centre, Gothenburg, Sweden. [14]Department of Arctic and Marine Biology, Faculty of Biosciences, Fisheries and Economics, The Arctic University of Norway, Tromsø, Norway. [15]Department of Chemistry, Life Sciences and Environmental Sustainability, University of Parma, Parma, Italy. [16]Terrestrial Ecology Section, Department of Biology, University of Copenhagen, Copenhagen, Denmark. [17]Center for Permafrost, Department of Geosciences and Natural Resource Management, University of Copenhagen, Copenhagen, Denmark. [18]Department of Forest Mycology and Plant Pathology, Swedish University of Agricultural Sciences, Uppsala, Sweden. [19]Department of Arctic and Marine Biology, UiT—the Arctic University of Norway, Tromsø, Norway. [20]Carbone Boréal, Département des Sciences Fondamentales, Université du Québec à Chicoutimi, Chicoutimi, Quebec, Canada. [21]Institute for Environmental Science and Sustainability, Wilkes University, Wilkes-Barre, PA, USA. [22]Swiss Federal Institute for Forest, Snow and Landscape Research WSL, Lausanne, Switzerland. [23]State Key Laboratory of Herbage Improvement and Grassland Agro-Ecosystems and College of Pastoral Agriculture Science and Technology, Lanzhou University, Lanzhou, China. [24]Agricultural University of Iceland, Reykjavik, Iceland. [25]Department of Terrestrial Ecology, Norwegian Institute for Nature Research, Trondheim, Norway. [26]Gjærevoll Centre for Biodiversity Foresight Analyses & Department of Biology, Norwegian University of Science and Technology, Trondheim, Norway. [27]The Heathland Centre, Alver, Norway. [28]Institute of Ecology, College of Urban and Environmental Sciences, Key Laboratory for Earth Surface Processes of the Ministry of Education, Peking University, Beijing, China. [29]Department of Biological Sciences, Dartmouth College, Hanover, NH, USA. [30]Biological Sciences, School of Natural Sciences, University of Tasmania, Hobart, Tasmania, Australia. [31]Australian Mountain Research Facility, Canberra, Australian Capital Territory, Australia. [32]Life and Environmental Sciences, University of Iceland, Reykjavík, Iceland. [33]Division of Life Sciences, Korea Polar Research Institute, Incheon, South Korea. [34]Research Centre for Ecological Change, Organismal and Evolutionary Biology Research Programme, Faculty of Biological and Environmental Sciences, University of Helsinki, Helsinki, Finland. [35]Korea Polar Research Institute, Incheon, Korea. [36]Institute of Soil Science, Universität Hamburg, Hamburg, Germany. [37]University of Eastern Finland, Department of Environmental and Biological Sciences, Kuopio, Finland. [38]Greenland Institute of Natural Resources, Nuuk, Greenland. [39]NORCE, Norwegian Research Centre AS, Bjerknes Centre for Climate Research, Bergen, Norway. [40]Department of Biology, Norwegian University of Science and Technology, Trondheim, Norway. [41]Department of Geography, University of Calgary, Calgary, Alberta, Canada. [42]Arctic Centre, University of Lapland, Rovaniemi, Finland. [43]Faculty of Biological and Environmental Sciences, University of Helsinki, Helsinki, Finland. [44]SEGES Innovation P/S, Aarhus, Denmark. [45]Ecological Sciences, The James Hutton Institute, Aberdeen, UK. [46]Department of Wildland Resources, Quinney College of Natural Resources and Ecology Center, Utah State University, Logan, UT, USA. [47]Department of Arctic Biology, University Centre in Svalbard, Longyearbyen, Norway. [48]Center for Volatile Interactions, Department of Biology, University of Copenhagen, Copenhagen, Denmark. [49]Center for Ecosystem Science and Society, Northern Arizona University, Flagstaff, AZ, USA. [50]Department of Ecoscience, Aarhus University, Roskilde, Denmark. [51]Arctic Research Centre, Aarhus University, Aarhus, Denmark. [52]Department of Biological Sciences, Northern Arizona University, Flagstaff, AZ, USA. [53]School of Biosciences, University of Nottingham, Sutton Bonington Campus, Loughborough, UK. [54]Department of Geography and Environmental Management, University of Waterloo, Waterloo, Ontario, Canada. [55]The Ecosystems Center, Marine Biological Laboratory, Woods Hole, MA, USA. [56]Natural Resources Institute Finland, Helsinki, Finland. [57]Norwegian Institute for Nature Research, Bergen, Norway. [58]Ecology and Genetics Research Unit, University of Oulu, Oulu, Finland. [59]State Key Laboratory of Vegetation and Environmental Change, Institute of Botany, Chinese Academy of Sciences, Beijing, China. [60]School of Forest Sciences, University of Eastern Finland, Joensuu, Finland. ✉e-mail: sybryn.maes@gmail.com

## Methods

### Data collection

**Study design.** We quantified and analysed the effect of warming on ecosystem $CO_2$ respiration (ER) across tundra ecosystems by collecting original, previously published and unpublished, data from OTC warming experiments in the tundra biome. OTCs are small greenhouses which passively increase air temperatures during the snow-free season while allowing free entry of precipitation[44]. They are commonly used to simulate climate warming at a plot-scale in low-stature alpine and arctic tundra systems (International Tundra Experiment network, ITEX[68]). We contacted potential data contributors (1) through email using the networks ITEX, WARM, InterAct and Permafrost Carbon, (2) at the 20th ITEX Meeting 9–13 September 2019 in Parma, Italy, and (3) by retrieving contact details of authors of earlier published meta-analyses of ER measured in warming experiments. ER datasets were only included if (1) experiments were located in alpine or arctic tundra, (2) original ER measurements were available from both OTC and unmanipulated control (ambient temperature) plots and (3) ER was measured during the plant growing season (that is, June through August for all sites, except for the Australian site AUS_1: October through February; Supplementary Fig. 2). This resulted in 24,035 data points from 136 ER datasets (that is, a unique site × experiment × ER measurement year; Supplementary Tables 1 and 2) from 56 unique OTC experiments across 28 tundra sites.

One site could consist of one or several warming experiments (for example, covering local differences in vegetation communities or soil conditions) and contribute ER measurements from one or several occasions during one or several years (Supplementary Table 1 and Supplementary Figs. 1 and 2). Experiments differed in abiotic (for example, permafrost presence, ambient climate and soil type) and/or biotic (for example, vegetation and size of microbial communities) conditions, as well as in warming treatment specifications. First, at the time of ER measurements, experiments differed in duration of their warming treatment (0–25 years, mean 7). Second, despite the common experimental warming setup, experiments differed in OTC size (0.3–0.7 m height), duration of OTC deployment (for example, year-round placement or winter removal of OTCs) and methodology of flux measurements. We therefore collected methodological data on (1) OTC height, (2) whether or not OTCs were removed during winter, (3) $CO_2$ analyser producer (Los Gatos Research, LICOR, PP-system or Vaisala), (4) $CO_2$ analyser type (infrared or laser), (5) measurement system (automated or manual and closed or open), (6) the timing of ER measurements (daytime or night time) and (7) ER measurement plot size. Details on the procedure used to check for methodological biases introduced by the factors described above are presented in the Supplementary Discussion 2. Because we observed no significant effects of these methodological factors on the ER response, we assume there is no methodological bias in our results.

**Ecosystem respiration.** For each dataset, we requested plot-specific daily average ER data and standardized all individual data points to the same unit, that is, $gCO_2\,m^{-2}\,d^{-1}$. We removed 339 outliers (0.8% of the data) to account for measurement errors (data points outside the range of mean ER ± 3 × s.d. in a dataset following ref. 69). Subsequently, we calculated growing season average ER per experimental treatment (control versus warmed) per dataset as grams of carbon in $CO_2\,m^{-2}\,d^{-1}$. For details on the sampling frequency, see Supplementary Fig. 2 and Supplementary Table 2. All experiments measured daytime ER using dark or opaque chambers, except for ALA_1, which used automated clear chambers. For this experiment, we extracted night-time measurements based on the photosynthetically active radiation values (PAR < 5 µmol $m^{-2}\,s^{-1}$) and included these as ER measurements.

### Effect size calculation

We calculated effect sizes[46] of the mean growing season ER response to warming, separately for each dataset. We used Hedges' $g$ SMD as primary effect size across all models and the log-transformed[5] ROM as additional effect size to quantify percentage change in ER with warming. Hedges' SMD for each dataset was calculated as the (mean growing season average ER in warmed plots − mean growing season average ER in control plots)/pooled standard deviation. This is deemed an appropriate effect size to use here because it overcomes data measured on different scales through standardization, as well as corrects for variance heterogeneity that may be introduced with small sample sizes ($n < 20$) by including the pooled standard deviation[46,70]. Mean Hedges' SMD values for each dataset were accompanied by 95% CIs of the standard errors, which reflect the precision of the effect size for each dataset[70]. The ROM was calculated as the log-transformed mean growing season average ER in warmed plots/mean growing season average ER in control plots, which we transformed to a mean percentage increase as $100\times(\exp(ROM) - 1)$, that is, a more intuitive measure of changes in ER. See Supplementary Table 2 for the number of day-average ER measurements and number of plots used for each of the 136 datasets and see Fig. 2 and Supplementary Table 4 for an overview of mean Hedges' SMD and 95% CIs per dataset.

### Temporal patterns

To assess the ER response to warming over time, we quantified the duration of experimental warming for each dataset as the difference between the year of ER measurement and the year in which the warming treatment started (Supplementary Fig. 1 and Supplementary Table 1). We further categorized the datasets into four 'age classes' on the basis of warming duration, with class 1 including datasets in which 0–5 years of warming had been ongoing at the time of ER measurements and classes 2, 3 and 4 in which 5–10, 10–15 and more than or equal to 15 years of warming had been ongoing, respectively (Fig. 1). Note that duration = 0 refers to when less than 1 year of warming took place, that is, one single summer of warming.

### Environmental drivers ER

To quantify environmental drivers that might explain the variation in the growing season ER response to warming, we obtained several types of environmental data on a site-, experiment- or ER-measurement level and used these to analyse indirect warming effects and context-dependencies as drivers of variability in ER response to warming. See Supplementary Table 5 for sample sizes of all environmental drivers used to assess indirect warming effects and context-dependencies and Supplementary Methods 1 for a detailed explanation on how the different drivers were obtained and calculated.

**Indirect warming effects.** To investigate whether and how changes in environmental conditions induced by the warming treatment influenced the ER response to warming, we quantified changes in the below-mentioned climatic and soil conditions as well as in vegetation and microbial communities for each dataset, by calculating effect sizes for these drivers: effect size calculation of ER data (above), for example, (mean air temperature warmed plots − mean air temperature in control plots)/pooled standard deviation. In doing so, we then tested whether changes in the environmental drivers due to warming influenced the changes in ER due to warming (see later section on 'Metaregression'). The included environmental drivers for this part of our analyses are related to (1) climatic conditions at ER measurement time, that is, air temperature (°C), soil temperature (°C) and soil moisture (%); (2) soil conditions at plot level, that is, SOM (%), total C concentration (TC) (%), total N concentration (TN) (%), C:N ratio, bulk density (BD) (g $cm^{-3}$) and pH from the mineral, and/or organic, layer; as well as organic layer depth (OLdepth) (cm); (3) the vegetation community

at plot level, that is, percentage cover per functional group (graminoids, forbs, deciduous shrubs, evergreen shrubs, mosses and lichens), aboveground biomass and mean community height, at plot level; and (4) the microbial community at plot level, that is, proxies for bacterial and fungal biomass and derived fungal:bacterial (FB) ratios (Extended Data Table 3). Although ER can be strongly correlated to GPP of the vegetation, our database does not include GPP data, as its inclusion and comparison across time and across several experiments is challenging because of the light-dependency of photosynthesis and associated high spatiotemporal variability of GPP measurements. However, both processes are (partly) driven by the activity of the vegetation, which in turn is driven by direct responses to their environment (for example, microclimate) as well as by, for example, its biomass and community composition. We therefore argue that to increase our mechanistic understanding of the underlying ecological drivers of ER response to warming, it is more meaningful to incorporate the role of the response of plant growth to warming for ecosystem respiration by including (changes in) plant biomass and community composition, as well as NPP (see section 'Context-dependencies') as potential drivers of ER, as we do here. Note that we refer to the soil total C and N concentration throughout the manuscript, because these drivers were measured on solid (dried) soil with CNS element analysers. Therefore, they refer to the sum of concentrations of both inorganic and organic compounds of C and N, even though TC most likely reflects primarily the SOC concentration because tundra soils contain limited-to-no inorganic C.

**Context-dependencies.** To evaluate whether and how the environmental context influenced the ER response to warming, we included drivers quantifying the climate, soil conditions and vegetation community (Extended Data Table 4 and Supplementary Methods 1). For each dataset, (1) climate drivers included zone (low arctic, high arctic and alpine) and permafrost probability (values between 0 and 1) at site level and mean air temperature, soil temperature and soil moisture in the control plots at ER measurement time. Further, (2) soil conditions drivers comprised mean values of the SOM, TC, TN, CN, BD, pH and OLdepth data from the control plots per dataset (see above). In addition, we included soil pH class (pH low less than 5, medium 5–7, high more than 7), soil moisture class (dry, mesic and wet) and soil carbon stock (t ha$^{-1}$) here at the experiment level. Finally, (3) vegetation community drivers included a vegetation class (five categories: barrens (B), graminoid tundra (G, grasses and sedges), prostrate shrub tundra (P), erect shrub tundra (S) and wetlands (W), that is, based on the circumpolar arctic vegetation map (CAVM)[71]) and NPP (kgC m$^{-2}$ yr$^{-1}$) at the experiment level.

### Statistical analyses
All analyses were performed using R v.4.0.3 unless specified otherwise[72].

**Modelling approach. Meta-analysis.** To evaluate the effects of experimental warming on ER, we performed multivariate meta-analysis with the rma.mv function from the metafor R package[73], using the Hedges' SMD of the growing season average ER data as primary effect size (Fig. 2 and Extended Data Table 1). In doing so, we obtained a mean pooled effect size of warming, taking into account the nestedness and repeated measurements in our 136 datasets. Each dataset was weighted by the inverse of its estimated sampling variance to increase both the precision of the estimated effect sizes and the statistical power in the meta-analyses[74]. We also performed a meta-analysis on the ROM as secondary effect size (see previous section on 'Effect size calculation'), which provided a mean pooled ROM of warming (Extended Data Table 1). Further, to test whether environmental conditions (climate, soil, vegetation and microbial: see previous section on 'Environmental drivers') significantly changed themselves because of the warming treatment, we performed separate meta-analyses testing the effect of warming on each of these drivers, using Hedges' SMD as primary effect

size and raw mean differences (that is, the difference in absolute values of the driver between treatment and control means) as secondary effect size because the latter provides estimates of absolute changes in the drivers (Supplementary Table 3).
**Metaregression. *Temporal patterns.*** To evaluate whether and how the ER response varied with duration of experimental warming, we performed two types of single-factor metaregression models. First, we tested whether duration (number of years of warming at time of ER measurements) influenced the magnitude of the ER response (ER Hedges' SMD). These models were performed across experiments and both (1) across the four age classes of duration, to analyse overall long-term patterns of warming effects and (2) in each age class (0–4, 5–9, 10–14 and more than 15 years). This enabled to explore potential nonlinearity of patterns in ER response rates with warming over time (Fig. 3b and Extended Data Table 2a). Second, we tested (1) whether the age class influenced the ER response, that is, whether the mean ER response differed among the four age classes in sign or direction, as well as (2) whether the mean ER response per age class was significantly different from zero (Fig. 3a and Extended Data Table 2b). This enabled to evaluate the persistency (continued existence) of the ER response over time.
***Indirect warming effects.*** To assess whether the variation in the ER response to warming was driven by indirect effects of warming on environmental conditions, we ran single-factor metaregression models using the Hedges' SMD of the different environmental drivers (reflecting warming-induced changes in climate, soil, vegetation and microbial community conditions) as individual predictors and ER Hedges' SMD as response (see previous section on 'Indirect warming effects'; Fig. 4 and Extended Data Table 3).
***Context-dependencies.*** To assess whether variation in the ER response to warming could be explained by context-specific environmental drivers related to climate, soil or vegetation, we performed single-factor metaregression models testing the influence of each of these drivers on ER Hedges' SMD (see previous section on 'Context-dependencies'; Fig. 5 and Extended Data Table 4).

We chose single-factor models to analyse the indirect warming effects and context-dependencies because the environmental driver data were unevenly spread across datasets (Supplementary Table 5) with different drivers missing from different datasets. Hence, multifactor models would have led to significant reductions in sample size and thus power. Before all metaregressions, we checked for confounding and/or collinearity issues through boxplots and correlograms.

**Random-effects structure.** Because we did not assume all datasets to have the same true effect size and to account for methodological differences across the datasets, we included a random-effects structure in all models[5]. Models were fitted with the following crossed and nested random effect structure: (-1 | experiment/dataset) + (-year | experiment). The first component (the multilevel structure) corresponds to a three-level meta-analysis, allowing the individual estimates of each observation (dataset) to be nested in the experiment grouping-level[75,76]. The second component (the autoregressive component) accounts for repeated samplings in the same site over several time points (years), by using a continuous autoregressive variance structure of true effects over time[76–78]. We modelled such a structure by setting struct = 'CAR' in the rma.mv function from the metafor R package[73]. This significantly improved the main meta-analysis model, that is, the model with the CAR structure had lower Akaike information criterion values than the model without; also, the acf-plot of computed 'normalized or Cholesky residuals' showed that autocorrelation was removed by adding the CAR. This random effect structure was then used for all meta-analyses and metaregression models.

**Model visualization and validation.** We considered and interpreted results as significant in our study if $P < 0.05$, as is commonly done in

meta-analyses[79–81], whereas trends ($P < 0.1$) are described but not interpreted as significant (Extended Data Fig. 3). We visualized meta-analysis results through forest plots using the forest function from the metafor package[8]. Significant metaregression results were visualized by showing the actual observations with scatterplots and violin plots for continuous and categorical drivers, respectively; as well as the model predictions, that is, regression lines or group means ± 95% CIs, respectively. Models were validated by evaluating funnel and profile plots and testing the residuals for the assumption of normality and variance homogeneity[73,82]. Funnel plots of the main meta-analysis showed asymmetry, indicating possible 'publication bias' but because we did not work with only published datasets (that is, not a meta-analysis sensu stricto) we argue that this detected asymmetry most likely originated from true heterogeneity rather than publication bias[83]. We did not observe any other irregularities. Further, we calculated $R^2$ of the significant metaregression models using McFadden's pseudo-$R^2$ by comparing log-likelihood values of the models to the one of the null model fitted through maximum-likelihood estimation. We report the omnibus test statistic Qm to estimate the amount of residual heterogeneity explained by each driver in the metaregression models, as well as the $Q$ or QE value test statistic for heterogeneity for meta-analysis or metaregression models, respectively.

## Supplementary analyses

To test the robustness of our results and to gain mechanistic insights which could help with further interpretation of our results, we performed several supplementary analyses (Supplementary Discussion). First, to answer whether the increased ER was driven by increased plant (autotrophic) or microbial (heterotrophic) respiration, or by both, we collected all the available data on ER partitioning in tundra warming experiments and performed meta-analyses on ER, autotrophic respiration and heterotrophic respiration for this subset (Extended Data Table 1b and Supplementary Discussion 1). Respiration partitioning measurements are destructive and hence alter ecosystem dynamics, making long-term measurements difficult. Hence, there is limited availability of continuous, simultaneous measurements of total ER and ER partitioning at the same site. Second, we analysed whether the differences in ER measurement methodology or OTC experiment setup resulted in methodological bias in our results (Supplementary Fig. 3 and Supplementary Discussion 2). Third, we analysed potential interactions of OTC microclimate effects on the ER response (Supplementary Discussion 3). Fourth, we assessed whether mismatching of ER with environmental driver data from different measurement years influenced our results through performing a sensitivity analysis, that is, rerunning our metaregression models on the basis of several restrictive sample size scenarios (Supplementary Fig. 4 and Supplementary Discussion 4). Fifth, we assessed whether the unbalanced sampling design from including two experiments with a much longer time series of repeated ER measurements (ALA_1 and GRE_6; Supplementary Table 2) than other experiments had a strong influence on the meta-analysis outcome; Supplementary Discussion 5). Finally, we analysed the temporal patterns in experiments as well, to see if they differed from our results when analysing these patterns across experiments (Supplementary Fig. 5 and Supplementary Discussion 6). Overall, these supplementary analyses confirmed the robustness of our results.

## Spatial upscaling

We estimated spatial patterns in the sensitivity of ER to experimental warming and the resulting change in ER, by upscaling our findings on context-dependencies across the arctic[6] and circumarctic alpine[84] tundra region (Supplementary Fig. 6). We performed the upscaling on the basis of the two context-dependent drivers which were significant predictors of ER responses to warming based on our single-factor metaregression models, that is, TN concentration and C:N ratio of the mineral soil layer. For the upscaling, a two-factor metaregression model was used (see details below). The detailed procedure is outlined below and visualized in Supplementary Methods 2.

**Soil input data.** As input for the upscaling, we downloaded the mean, 5th percentile and 95th percentile soil data (of TN, SOC concentration (%) and BD) for the study area with 250 m resolution from the ISRIC soil data hub[85]. Given that the study area was about 10 million km², the 250 m resolution was not computationally feasible for Monte Carlo uncertainty analysis (see section 'Monte Carlo uncertainty analysis') and thus, we aggregated the data to 1 km resolution (mean over each 1 km grid cell, ignoring the grid cells without values). A supercomputer was used to perform the analysis on the basis of the aggregated 1 km resolution. We first extracted the mean TN and SOC concentration for the mineral layer of each grid cell by defining the mineral soil layers within 0–60 cm depth using a TN threshold of $0.01 \, g \, g^{-1}$, a SOC threshold of $0.1 \, g \, g^{-1}$ and a BD threshold of $1 \, g \, cm^{-3}$. We then calculated depth-weighted averages of each soil driver (TN and SOC) over the mineral layer for each 1 km grid cell. To estimate standard deviation (s.d.) for each grid cell, we used the 5th and 95th percentiles of the SOC and TN layers calculated as s.d. = $(q.95 - q.05)/(2 \times qnorm(0.95))$, where q.95 and q.05 are 95th and 5th percentiles and qnorm is the inverse cumulative distribution function of a standard normal distribution. Because there was no dataset available for C:N ratio, we used the means and standard deviations of TN and SOC data to derive grid-cell-specific C:N ratios as SOC/TN, as advised by ISRIC soil data hub personnel. This derivation and the related uncertainty are described below in the section Monte Carlo uncertainty analysis.

**Upscaling model.** The regression model for upscaling was built with the log-transformed ROM as effect size (response variable), which allowed to calculate projected respiration as percentage change in ER with 1.4 °C warming (that is, the average warming achieved by the OTCs in our database), starting from a baseline ER. We used a two-factor metaregression model based on all ER datasets for which both TN concentration and C:N ratios of the mineral soil layer were available, with the following estimated coefficients: ROM = $0.05 - 0.16 \times$ TN + $0.01 \times$ C:N (QM = 6.7 ($P < 0.05$), QE = 274 ($P < 0.001$), $n = 39$).

**Monte Carlo uncertainty analysis.** To incorporate the uncertainty in the soil data and their relationship with ER changes into our upscaling estimates, we performed Monte Carlo simulations. To propagate uncertainty in the input soil data, we generated 100 values for TN and SOC for each 1 km² grid cell using the means and standard deviations derived from the ISRIC soil data. For each cell we sampled from a truncated multivariate random normal distribution (rtmvnorm from R package tmvtnorm), truncating the distribution of both variables to minimum of 5% of the mean value to avoid negatives. Because truncation alters the characteristics of the sample, we adjusted the means so that the mean of the truncated distribution matched the target mean obtained from ISRIC. The correlation between TN and SOC was set at 0.8273. For each sample of TN and SOC, a C:N ratio was calculated. We took the potential bias of using C:N ratio based on TN and SOC as a proxy for C:N ratio into account by fitting a linear regression between the reported C:N values from the database of this study and the calculated C:N values based on the reported TN and TC (Supplementary Fig. 8). We fitted 100 regressions to the scatter plot and used each regression to estimate the C:N ratio of each iteration ($n = 100$) in each grid cell.

To propagate more uncertainty derived from the estimated relationship between ROM and TN and C:N, we then used the 100 pairs of TN and C:N of each 1 km² grid cell as input for the upscaling model using predict.rma function from the metafor R package[73]. This resulted in 100 predictions for ER percentage change (with associated prediction standard errors) for each grid cell. These 100 parameter sets were combined to produce an overall predicted value by taking the average, with the associated standard deviation computed by treating the prediction

distribution as a mixture of the 100 prediction intervals. The standard deviation of the resulting distribution is given by:

$$SD_{pred} = \sqrt{\frac{1}{n} \sum (\mathbf{MEAN}^2 + \mathbf{SE}^2) - \left(\frac{1}{n} \sum \mathbf{MEAN}\right)^2}$$

where $n$ is the number of simulated soil datasets and **MEAN** and **SE** are the vectors of the corresponding predicted means and standard errors.

This resulted in a predicted relative change in induced by 1.4 °C warming averaged over the 100 predictions pairs and the corresponding combined standard deviation, for each $1 \times 1\,km^2$ grid cell (Fig. 6 and Supplementary Fig. 7). To estimate spatial patterns in the absolute change in ER, we then multiplied this relative change in respiration with spatially explicit baseline ER (obtained by summing gridded data of soil respiration[86] (data compiled from 1960 to 2011) and plant respiration[87] (data from 2015)).

We then estimated the relative contribution of model parameter uncertainty versus input soil data uncertainty to Monte Carlo estimates of the ER standard deviation (Fig. 6). Model predictions were conducted twice on the 100 input sets, first propagating both uncertainty sources (Fig. 6b) and subsequently by fixing metaregression parameters to their mean estimates and only allowing soil input data to vary. The same input data simulations were used in both runs to keep the data-related uncertainty constant. Figure 6c shows the ratio of the resulting standard deviations, that is, soil input uncertainty only/combined uncertainty.

Finally, we explored how well the distribution of our dataset compares to the distribution of gridded soil data used for upscaling. Thereto, we created a scatter plot for TN and C:N ratio in which we overlaid both the observed (from our dataset) and gridded values (gridded soil dataset used for upscaling) (Supplementary Fig. 9). Although the observed values from our dataset are slightly biased towards low TN values, overall, the observed values cover the main concentration of TN-C:N ratio distribution (about 64% when measured with Convex Hull polygon).

## Data availability

Data will be available on Zenodo at https://doi.org/10.5281/zenodo.10572479 (ref. 88). The maps in Fig. 1 and Extended Data Fig. 1 were made with R (packages ggplot2 and country code) and for the maps in Fig. 6 and Supplementary Figs. 6 and 7 country borders from Natural Earth were used and maps were done with R (package tmap).

## Code availability

R scripts will be available on GitHub at https://github.com/mjalava/tundraflux (ref. 89).

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

**Acknowledgements** We wish to thank the Abisko Scientific Research Station for hospitality and logistic support. For feedback on the statistical analyses, we are grateful to the R-sig-meta-analysis mailing list. We also wish to acknowledge CSC—IT Center for Science, Finland, for computational resources and the many students and field assistants involved in the monitoring, measurements and sample processing at the tundra sites over the decades of data collection included in this study. Thank you to U. Molau and O. Khitoun for data contribution from the SWE_12, 14, 15 sites and to Dries VDL for permanent support. We acknowledge that some of the fieldwork has been conducted on traditional lands. We deeply regret the passing of J. Tang, one of our co-authors, during the writing of the manuscript and wish to highlight the fundamental contributions he has made as an ecosystem ecologist to the field of carbon cycle science. During the preparation of the manuscript, S.L.M. was funded by the Flemish Research Foundation (FWO grant no. 12ZZV21N) and E.D. was funded by the Swedish Research Council VR (grant no. 2018-04004) and Knut and Alice Wallenberg Foundation (grant no. KAW 2020.0126). We also want to thank the following funding agencies for support in field- or laboratory-work or salaries as part of the in situ data collection and data synthesis: Climate Impact Research Centre, European Research Council (grant no. 819202 awarded to M.K. and grant no. 77102 to I.R.), Academy of Finland (grant nos. N-PERM 341348 to C.B., THAW-N 349503 and ACCC 337550 to M.E.M., and no. 347188 to E.K.), the Österreichischer Wissenschaftsfonds FWF (Lise Meitner grant no M-3335 to C.B.), the Research Council of Finland's Flagship Programme under project Digital Waters (grant no. 359248 to M.K.), European Commission Directorate-General for Research (grant no. EVK2-2001-00235 to E.J.C. and grant no. 771012 to R.R.), Danish National Research Foundation (grant no. CENPERM DNRF 100 awarded to B.E. and A.M. and C.S.N.), National Science Foundation of United States (grant nos. PLR 1418010 to N.F., PLR 1417763 to T.C.P. and ANS-2113641 supported M. Petit Bon), Norwegian Research Council (FRIMEDBIO grant no. 274712 to J.G. and J.P.T. and V.V., grant no. 274712 to R.G. and S.V.H., grant nos. 276080 and 343398 to B.B.H., grant nos. SFF-III 223257 to Norwegian University of Science and Technology to M.L.M., grant nos. FEEDBACK, 250740 and EMERALD, 294948 to H.L. and Arctic Field grant no. 269957 to M. Petit Bon), Greenland Ecosystem Monitoring funds to K.R., ARCUM, Stiftelsen Ymer-80 and SNSF (grant no. PZ00P2_174047 to K.G.), Swedish Research Council FORMAS (grant nos. 2013-655 to S.H. and 2021-02449 to J.M.S.), Alberta Innovates Technology Futures and Canada Research Chairs programme to M.S., Australian Research Council Discovery Projects (grant no. DP220100915 to M.J.H.), Icelandic Research Fund (to I.S.J.), National Research Foundation of Korea (grant nos. NRF-2021M1A5A1075508 and KOPRI-PN22012 to J.Y.J.), Svalbard Environmental Protection Fund (grant no. 15/128 to M. Petit Bon and grant no. 16/113 to B.B.H.), Danish Environmental Protection Agency (for funding N.M.S.), Academy of Finland (grant no. MUFFIN 332196 to C.V.), Maj and Tor Nessling Foundation (for funding H.Y.), Maaja vesitekniikan tukiry (to M.J.), European Union's Horizon 2020 research and innovation programme under the Marie Skłodowska-Curie agreement (grant no. 657627 to M.P. Björkman), FORMAS Swedish Research Council for Sustainable Development (for funding M.P. Björkman), Climate-ecological Observatory for Arctic Tundra-COAT and UNIS (logistical support for fieldwork M.L.M. and B.B.H.). The strategic research environment BECC—Biodiversity and Ecosystem Services in a Changing Climate (to M. P. Björkman) and for funding T.S., we thank the US Department of Energy, Office of Biological and Environmental Research (Terrestrial Ecosystem Science Program nos. DE-SC0006982, DE-SC0014085 and DE-SC0020227 and the Permafrost Carbon Network (NSF project nos. 1331083 and 1931333).

**Author contributions** E.D., J.M.S. and M.P. Björkman conceived the idea and S.L.M., M.P. Björkman, J.M.S. and E.D. designed the study. E.D., G.M., J.D., J.M.S., M.P. Björkman, S. Schwieger, and V.V. gave extensive feedback on the analyses and manuscript. S.L.M. and J.D. assembled the ER data for meta-analysis and metaregression performed by S.L.M. S. Schwieger assembled and analysed the AR/HR data for meta-analysis. M.J., J.T.W. and M.K. performed the spatial upscaling and designed all related figures. S. Schwieger, J.D. and S.L.M. codesigned Fig. 1. S. Schwieger and S.L.M. codesigned the Graphical Abstract, made by S. Schwieger. J.M.S. produced Supplementary Fig. 4. S. Schwieger produced Supplementary Fig. 5. S.L.M. designed all other figures and tables and wrote the manuscript. M.K. did the final formatting of the figures. All authors, excluding G.M., J.D., J.M.S., S.L.M. and S. Schwieger, contributed data. V.V., R.A., I.A., R.B., M.C., G.C., C.T.C., P.F., T.F., K.G., S.V.H., C.H.P., I.S.J., J.Y.J., T.M., C.S.N., T.C.P., E.P.P., M. Petit Bon, A.P., K.R., N.M.R.R., R.R., I.R., M.V., C.V., H.O., S. Sjögersten,

M.P. Björkman and E.D. also participated in in situ data collection. I.A., C.B., J.H.C., C.T.C., E.D., H.L., S. Sjögersten, V.V. and C.V. collected in situ flux partitioning data. M.J. developed the data and code repositories. All authors reviewed the manuscript, excluding G.C., H.B., H.O., J.T., J. Guðmundsson, M.J., M.N., N.M.R.R., R.E.L., R.S.V.L., R.A. and Z.G. who only contributed data.

**Funding** Open access funding provided by Umea University.

**Competing interests** The authors declare no competing interests.

**Additional information**
**Correspondence and requests for materials** should be addressed to S. L. Maes.

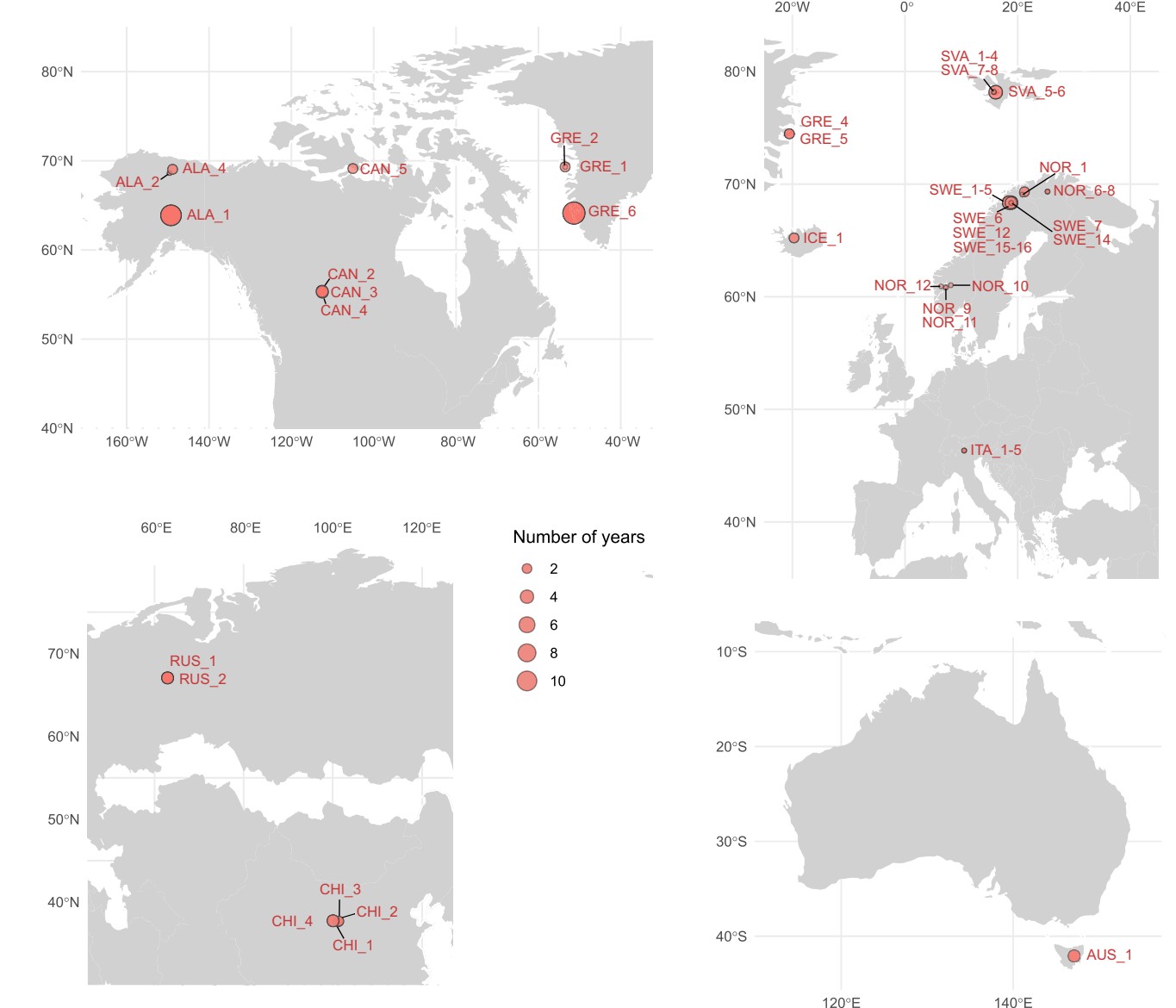

**Extended Data Fig. 1 | Maps showing the locations of the 28 sites (red dots) used for meta-analysis across four different continents: America (top left), Europe (top right), Asia (bottom left) and Australia (bottom right).** Details in Supplementary Table 1. The radius of the red dots reflects the number of years that ecosystem respiration (*ER*) data was measured at each site, resulting in a total of 136 datasets.

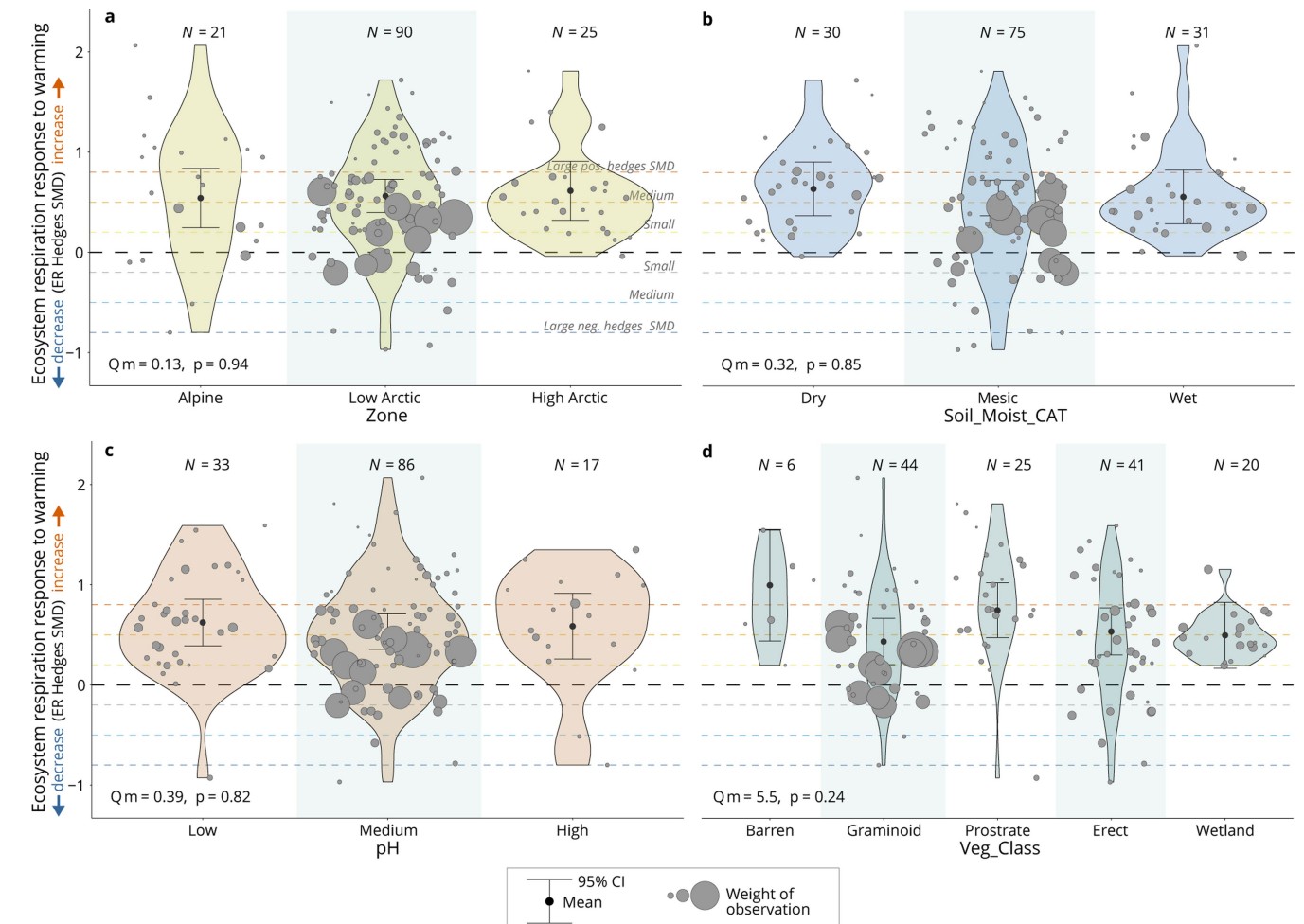

**Extended Data Fig. 2 | Uniform distribution of *ER* response to warming across the environmental context of all datasets.** *ER* response to experimental warming across the spatial environmental context (See Fig. 1 in Main text). Violin plots of the actual data across the categorical environmental drivers, that is, Climate *Zone* (a), Soil moisture class (b), Soil *pH* class (c) and Vegetation class (d), are displayed, showing the kernel density estimation of the underlying distributions. *ER Hedges SMD*s for individual datasets are displayed with grey bubbles, calculated as (mean *ER* of the warmed plots - mean *ER* of the control plots)/pooled standard deviation. Bubble size denotes the weight of the observation used in the metaregression models quantified as the inverse of the square root of within-study variance, with greater bubbles indicating greater

weights. Within the violin plots, single-factor metaregression model estimates and 95% confidence intervals are displayed with black circles and error bars. Above the x-axis, the 'Qm' value represents the importance of the moderator or environmental driver with *p*-value ('*p*-val') for each model. Number of datasets ('N') for each environmental driver per category is shown above the violin plots. The black dashed horizontal line (SMD = 0) represents no change in ER with warming while the areas above and below represent increased (SMD > 0) vs. decreased (SMD < 0) ER with warming. Dashed horizontal lines (y = 0.2, 0.5, 0.8 and −0.2, −0.5, −0.8) reflect small, medium and large positive and negative *Hedges SMD* effect sizes or increasingly greater *ER* increases and decreases with warming. For detailed model output, see Extended Data Table 4.

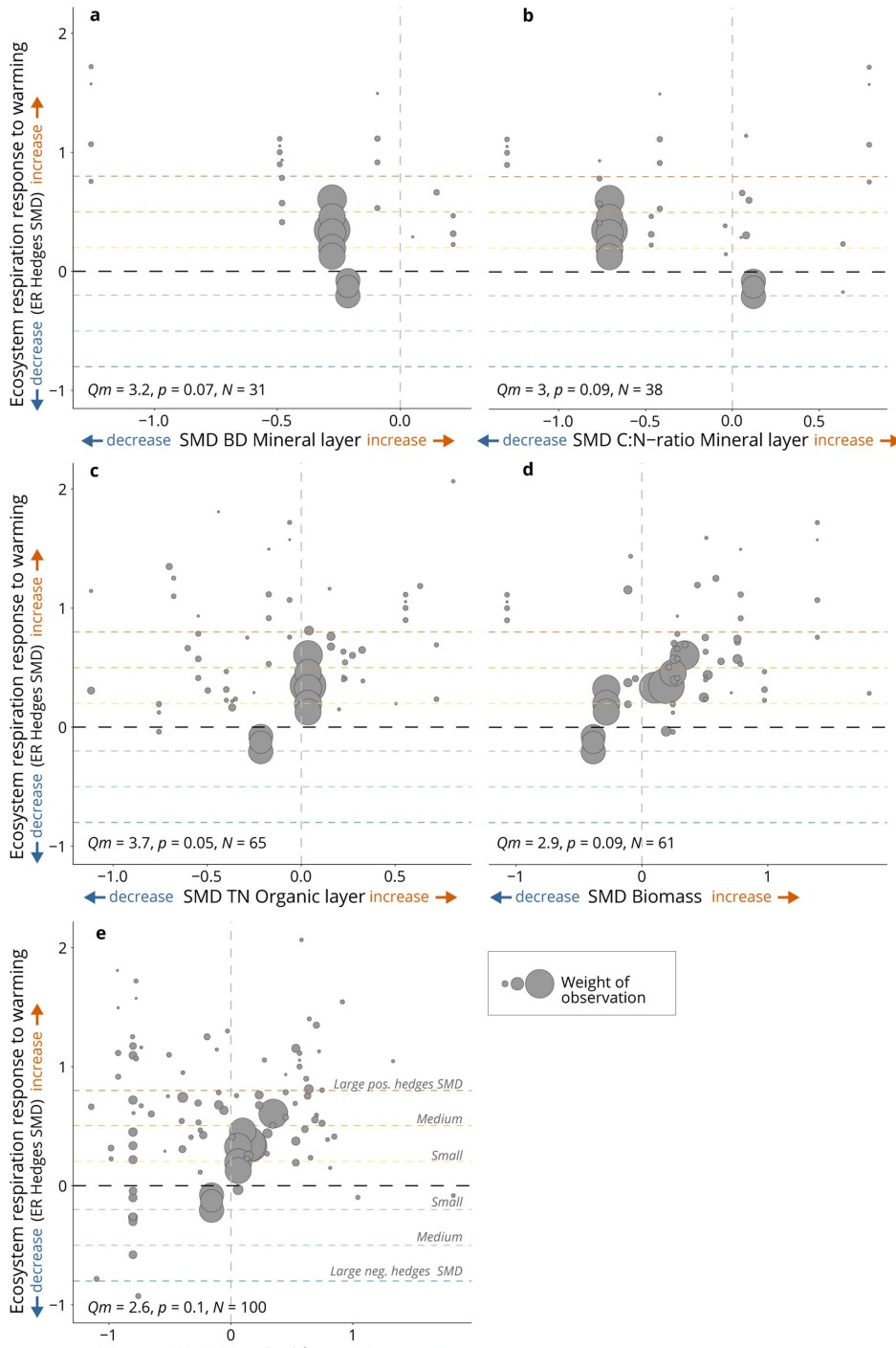

**Extended Data Fig. 3 | Warming-induced changes in soil conditions drive *ER* response.** Trends in warming-induced changes in soil conditions and in the vegetation community driving the *ER* response to experimental warming (*p* < 0.1): *Hedges SMD* of bulk density (*BD*) (a) and C:N ratio (b) of the soil mineral layer and of total N (*TN*) concentration of the soil organic layer (c) and aboveground biomass (d) and graminoid cover (e) of the vegetation community. *ER Hedges SMD*s for individual datasets are displayed on the y-axis with grey bubbles, calculated as (mean *ER* of the warmed plots - mean of the control plots)/pooled standard deviation. Bubble size denotes the weight of the observation used in the metaregression quantified as the inverse of the square root of within-study variance, with larger bubbles indicating greater weights. Top left in each panel shows the sample size ('N', number of datasets) and bottom left shows the 'Qm' (Q-value of importance of the environmental drivers) and '*p*'-value of the metaregression models. The black dashed horizontal line (y = 0) represents no change in ER with warming while the areas above and below represent increased (SMD > 0) vs. decreased (SMD < 0) ER with warming. Dashed horizontal lines (y = 0.2, 0.5, 0.8 and −0.2, −0.5, −0.8) reflect small, medium and large positive and negative *Hedges SMD*, respectively[50] or increasingly greater *ER* increases and decreases with warming. The black dashed vertical line (x = 0) reflects no change in the environmental condition with warming (with areas right and left of it representing increased vs. decreased conditions with warming). For detailed model output, see Extended Data Table 3.

**Extended Data Table 1 | Meta-analyses evaluating effects of experimental warming on Ecosystem (*ER*), Autotrophic (*Ra*) and Heterotrophic Respiration (*Rh*)**

| Response | Slope [95%CI]* | Q value | Multilevel structure | | Autoregressive component (inner \| outer) | | N | Percentage change[†] |
|---|---|---|---|---|---|---|---|---|
| | | | $\sigma^2_1$ | $\sigma^2_2$ | tau^2 | rho | | |
| *Main meta-analysis* | | | | | | | | |
| Ecosystem Respiration (*ER*) | *Hedges SMD* ↑ **0.57 [0.44, 0.70]** *** | 731 | 0.00 | 0.00 | 0.19 | 0.92 | 136 | NA |
| | *Ratio of Means (ROM)* ↑ **0.26 [0.20, 0.32]** *** | 907 | 0.00 | 0.00 | 0.04 | 0.94 | 136 | ↑ **29.8 [22.3, 37.8] %** |
| *Sub meta-analysis* | | | | | | | | |
| Ecosystem Respiration (*ER*) | *Hedges SMD* ↑ **0.72 [0.42, 1.02]** *** | 19 | 0.02 | 0.09 | 0.00 | 0.00 | 9 | NA |
| | *ROM* ↑ **0.33 [0.18, 0.48]** *** | 23 | 0.03 | 0.00 | 0.00 | 0.35 | 9 | ↑ **39.2 [19.6, 62.1] %** |
| Autotrophic Respiration (*Ra*) | *Hedges SMD* ↑ **0.44 [0.08, 0.80]** *** | 26 | 0.00 | 0.20 | 0.00 | 0.00 | 9 | NA |
| | *ROM* ↑ **0.35 [0.08, 0.83]** *** | 27 | 0.07 | 0.13 | 0.00 | 0.00 | 9 | ↑ **56.9 [6.9, 130.2] %** |
| Heterotrophic Respiration (*Rh*) | *Hedges SMD* ↑ **0.92 [0.36, 1.48]** *** | 43 | 0.00 | 0.00 | 0.50 | 0.84 | 9 | NA |
| | *ROM* ↑ **0.44 [0.28, 0.60]** *** | 19 | 0.03 | 0.00 | 0.00 | 1.00 | 9 | ↑ **54.5 [31.8, 81.2] %** |

*Significant results are highlighted in bold: * *p*<0.05; ** <0.01; *** <0.001.
[†]Percentage change = 100*(exp(*ROM*)-1)

*From top to bottom*: The different types of respiration responses used for meta-analyses, that is, ecosystem (*ER*) vs. autotrophic or plant (*Ra*) vs. heterotrophic or microbial (*Rh*) respiration, with *ER*=*Ra*+*Rh*; the different types of effect sizes used for meta-analyses, that is, *Hedges SMD* vs. *R*atio of Means (*ROM*). *From left to right*: The type of respiration investigated ('Response'); the model results for each response, testing whether respiration significantly changes with experimental warming ('Slope'); the Q-value of heterogeneity; the output of the multilevel model structure (represented by $\sigma^2_1$ and $\sigma^2_2$) and of the autoregressive model component (tau² and rho); the sample size ('N' or number of datasets); and the percentage change calculated from the *ROM* as 100*(exp(*ROM*)-1). Meta-analysis results are presented as the mean model estimate (slope) and 95% confidence intervals, with a down/upward arrow (↓/↑) indicating significantly decreased or increased respiration with warming and the significance level based on *p*-values. Significant drivers and results are in **bold**.

**Extended Data Table 2 | Metaregression models evaluating effects of experimental warming duration on *ER* response to warming**

| Driver | Slope [95%CI] [*] | QE | QM | R² | Multilevel structure | | Autoregressive component (inner \| outer) | | N |
|---|---|---|---|---|---|---|---|---|---|
| | | | | | $\sigma^2_1$ | $\sigma^2_2$ | tau^2 | rho | |
| *a Warming duration* | Across age classes (all data) | | | | | | | | |
| | 0.00 [-0.02, 0.01] | 683 | 0.2 | 0.00 | 0.00 | 0.00 | 0.20 | 0.93 | 136 |
| | Within age classes | | | | | | | | |
| | [0-5 years) | | | | | | | | |
| | ↓ -0.01 [-0.08, 0.06] | 186 | 0.1 | 0.00 | 0.06 | 0.03 | 0.00 | 0.05 | 70 |
| | **[5-10 years)** | | | | | | | | |
| | **↓ -0.13 [-0.16, -0.10] ***** | **63** | **63** | **0.45** | **0.23** | **0.00** | **0.00** | **1.00** | **28** |
| | **[10-15 years)** | | | | | | | | |
| | **↑ 0.15 [0.02, 0.27] *** | **36** | **5.6** | **0.33** | **0.13** | **0.00** | **0.00** | **1.00** | **15** |
| | ≥15 years | | | | | | | | |
| | 0.00 [-0.13, 0.13] | 85 | 0.0 | 0.00 | 0.51 | 0.00 | 0.00 | 1.00 | 23 |
| *b Age class* | Age class effect | 468 | 5.2 | 0.03 | 0.00 | 0.00 | 0.20 | 0.93 | 136 |
| | Significance from zero | | | | | | | | |
| | **[0-5 years)** | | | | | | | | |
| | **0.63 [0.48, 0.79]*** ** | | | | | | | | |
| | **[5-10 years)** | | | | | | | | |
| | **0.41 [0.21, 0.62]*** ** | | | | | | | | |
| | **[10-15 years)** | | | | | | | | |
| | **0.40 [0.14, 0.64]*** ** | | | | | | | | |
| | **≥15 years** | | | | | | | | |
| | **0.61 [0.33, 0.89]*** ** | | | | | | | | |

[*]Significant results are highlighted in bold:* *p*<0.05; ** <0.01; *** <0.001.

*From top to bottom*: Metaregression model results testing the effects of **a)** experimental warming duration and **b)** age classes based on experimental warming duration, on *ER* Hedges SMD for all data or for separate age classes. *From left to right*: The specific driver used as predictor to test the effect on *ER* Hedges SMDs ('Driver'); the model results for each driver, testing whether *ER* Hedges SMDs are significantly influenced by the driver ('Slope'); the 'QE'-value of heterogeneity, as well as the 'QM'-value, reflecting importance of the driver and the McFadden's pseudo-R² ('R²'); the output of the multilevel model structure (represented by σ²₁ and σ²₂) and of the autoregressive model component (tau² and rho); and the sample size ('N' or number of datasets). Metaregression results are presented as the mean model estimate and 95% confidence intervals ('Slope (95%CI)'), with a down/upward arrow (↓/↑) indicating that higher values of the driver significantly decreased or increased *ER* responses to warming and the significance level based on *p*-values. Significant drivers and results are in **bold**.

**Extended Data Table 3 | Metaregression models evaluating effects of environmental drivers (indirect warming effects) on *ER* response to warming**

| Driver | Slope [95%CI] [*] | QE | QM | R² | Multilevel structure | | Autoregressive component (inner \| outer) | | N |
|---|---|---|---|---|---|---|---|---|---|
| | | | | | $\sigma^2_1$ | $\sigma^2_2$ | tau^2 | rho | |
| *Type* | | | | | | | | | |
| *Climate* | | | | | | | | | |
| *SMD* Air temperature | ↓ -0.05 [-0.28, 0.18] | 467 | 0.2 | 0.00 | 0.00 | 0.00 | 0.16 | 0.90 | 77 |
| *SMD* Soil temperature | ↑ 0.11 [-0.04, 0.26] | 536 | 2.0 | 0.02 | 0.00 | 0.00 | 0.12 | 0.95 | 118 |
| *SMD* Soil moisture | ↓ -0.03 [-0.14, 0.08] | 621 | 0.3 | 0.00 | 0.00 | 0.00 | 0.18 | 0.93 | 111 |
| *Soil* | | | | | | | | | |
| *SMD* SOM (min, org) | ↑ 0.20 [-0.40, 0.79] | 80 | 0.4 | 0.01 | 0.49 | 0.00 | 0.00 | 1.00 | 28 |
| | ↑ 0.19 [-0.29, 0.66] | 93 | 0.6 | 0.01 | 0.00 | 0.00 | 0.26 | 1.00 | 41 |
| *SMD* TC (min, org) | ↑ 0.17 [-0.17, 0.52] | 270 | 1.0 | 0.04 | 0.00 | 0.00 | 0.14 | 0.95 | 42 |
| | ↓ -0.03 [-0.24, 0.18] | 379 | 0.1 | 0.00 | 0.00 | 0.00 | 0.12 | 0.94 | 65 |
| **SMD TN (min**, org) | **↑ 0.24 [0.04, 0.44] *** | **186** | **5.4** | **0.22** | **0.00** | **0.00** | **0.11** | **0.95** | **42** |
| | ↑ 0.27 [0.00, 0.54] (*) | 341 | 3.7 | 0.08 | 0.00 | 0.00 | 0.11 | 0.95 | 65 |
| *SMD* CN (min, org) | ↓ -0.21 [-0.46, 0.03] (*) | 185 | 3.8 | 0.12 | 0.00 | 0.00 | 0.13 | 0.96 | 38 |
| | ↓ -0.04 [-0.16, 0.08] | 404 | 0.6 | 0.00 | 0.00 | 0.00 | 0.11 | 0.94 | 71 |
| **SMD pH (min**, org) | **↑ 1.49 [0.06, 2.91] *** | **65** | **4.2** | **0.12** | **0.37** | **0.00** | **0.00** | **1.00** | **27** |
| | ↓ -0.03 [-0.17, 0.11] | 115 | 0.2 | 0.00 | 0.00 | 0.00 | 0.18 | 0.99 | 53 |
| *SMD* Bulk density | ↓ -0.53 [-1.11, 0.05] (*) | 231 | 3.2 | 0.27 | 0.00 | 0.00 | 0.12 | 0.94 | 31 |
| (min, org) | ↑ 0.18 [-0.13, 0.50] | 379 | 1.3 | 0.03 | 0.00 | 0.00 | 0.31 | 0.98 | 49 |
| *SMD* Org layer depth | ↓ -0.14 [-0.99, 0.71] | 115 | 0.1 | 0.00 | 0.30 | 0.00 | 0.00 | 1.00 | 44 |
| *Vegetation* | | | | | | | | | |
| *SMD* Graminoids | ↑ 0.16 [-0.04, 0.36] (*) | 565 | 2.6 | 0.02 | 0.00 | 0.00 | 0.18 | 0.91 | 100 |
| *SMD* Forbs | ↑ 0.04 [-0.17, 0.25] | 593 | 0.1 | 0.00 | 0.00 | 0.00 | 0.18 | 0.91 | 111 |
| *SMD* Decid shrubs | ↓ -0.05 [-0.19, 0.08] | 579 | 0.6 | 0.01 | 0.00 | 0.00 | 0.20 | 0.93 | 90 |
| *SMD* Evergr shrubs | ↑ 0.08 [-0.11, 0.28] | 566 | 0.7 | 0.01 | 0.00 | 0.00 | 0.19 | 0.91 | 88 |
| *SMD* Mosses | ↑ 0.09 [-0.05, 0.23] | 585 | 1.6 | 0.02 | 0.00 | 0.00 | 0.20 | 0.91 | 88 |
| *SMD* Lichens | ↑ 0.10 [-0.13, 0.33] | 518 | 0.7 | 0.01 | 0.00 | 0.00 | 0.20 | 0.93 | 78 |
| *SMD* Biomass | ↑ 0.21 [-0.03, 0.45] (*) | 222 | 2.9 | 0.06 | 0.00 | 0.00 | 0.12 | 0.95 | 61 |
| *SMD* Comm height | ↓ -0.07 [-0.28, 0.15] | 102 | 0.4 | 0.01 | 0.00 | 0.00 | 0.16 | 1.00 | 43 |
| *Microbial* | | | | | | | | | |
| *SMD* Bacterial Abund | ↑ 0.16 [-0.05, 0.37] | 11 | 2.2 | 0.27 | 0.01 | 0.00 | 0.00 | 0.99 | 16 |
| *SMD* Fungal Abund | ↓ -0.10 [-0.52, 0.33] | 14 | 0.2 | 0.03 | 0.04 | 0.00 | 0.00 | 0.94 | 16 |
| *SMD* FB-*Ratio* | ↓ -0.11 [-0.48, 0.25] | 13 | 0.4 | 0.07 | 0.04 | 0.00 | 0.00 | 1.00 | 16 |

[*]Significant results are highlighted in bold: **\* p<0.05; \*\* <0.01; \*\*\* <0.001.** Trends (p<0.1) are indicated with (*).

*From top to bottom*: Metaregression model results testing the effects of warming-induced changes in local environmental conditions on *ER Hedges SMD* ('indirect warming effects'). *From left to right*: The type of environmental driver investigated; the specific driver used as predictor to test the effect on *ER Hedges SMD*s ('Driver'); the model results for each driver, testing whether *ER Hedges SMD*s are significantly influenced by the driver ('Slope'); the 'QE'-value of heterogeneity, as well as the 'QM'-value, reflecting importance of the driver and the McFadden's pseudo-R² ('R²'); the output of the multilevel model structure (represented by $\sigma^2_1$ and $\sigma^2_2$) and of the autoregressive model component (tau² and rho); and the sample size ('N' or number of datasets). Metaregression results are presented as the mean model estimate and 95% confidence intervals ('Slope (95%CI)'), with a down/upward arrow (↓/↑) indicating that higher values of the driver significantly decreased or increased *ER* responses to warming and the significance level based on *p*-values. Significant drivers and results are in **bold**.

**Extended Data Table 4 | Metaregression models evaluating effects environmental drivers (context-dependencies) on *ER* response to warming**

| Driver | Slope [95%CI][*] | QE | QM | R² | Multilevel structure | | Autoregressive component (inner \| outer) | | N |
|---|---|---|---|---|---|---|---|---|---|
| | | | | | $\sigma^2_1$ | $\sigma^2_2$ | tau^2 | rho | |
| *Type* | | | | | | | | | |
| *Climate* | | | | | | | | | |
| Zone | Non-significant | 713 | 0.1 | 0.00 | 0.00 | 0.00 | 0.20 | 0.93 | 136 |
| Permafrost prob | ↑ 0.07 [-0.25, 0.38] | 709 | 0.2 | 0.00 | 0.00 | 0.00 | 0.19 | 0.93 | 133 |
| Air temperature | 0.00 [-0.02, 0.03] | 479 | 0.1 | 0.00 | 0.00 | 0.00 | 0.16 | 0.90 | 77 |
| Soil temperature | ↑ 0.01 [-0.01, 0.04] | 500 | 1.2 | 0.01 | 0.00 | 0.00 | 0.12 | 0.94 | 118 |
| *Soil* | | | | | | | | | |
| Soil moisture | 0.00 [0.00, 0.00] | 612 | 0.1 | 0.00 | 0.00 | 0.00 | 0.18 | 0.93 | 111 |
| Soil moisture class | Non-significant | 673 | 0.3 | 0.00 | 0.00 | 0.00 | 0.20 | 0.93 | 136 |
| SOM (min, org) | ↑ 0.05 [-0.14, 0.24] | 79 | 0.2 | 0.01 | 0.46 | 0.00 | 0.00 | 1.00 | 28 |
| | 0.00 [-0.02, 0.01] | 96 | 0.8 | 0.02 | 0.00 | 0.00 | 0.27 | 1.00 | 41 |
| Soil Carbon stock | 0.00 [-0.00, 0.00] | 682 | 0.0 | 0.00 | 0.00 | 0.00 | 0.20 | 0.93 | 131 |
| TC (min, org) | ↓ -0.02 [-0.04, 0.01] | 263 | 1.9 | 0.08 | 0.00 | 0.00 | 0.11 | 0.92 | 42 |
| | 0.00 [-0.01, 0.01] | 357 | 0.1 | 0.00 | 0.00 | 0.00 | 0.17 | 0.96 | 70 |
| **TN (min**, org) | **↓ -0.60 [-1.06, -0.13] *** | **302** | **6.3** | **0.27** | **0.00** | **0.00** | **0.08** | **0.91** | **43** |
| | 0.00 [-0.26, 0.26] | 394 | 0.0 | 0.02 | 0.00 | 0.00 | 0.17 | 0.96 | 70 |
| **C:N (min**, org) | **↑ 0.03 [0.00, 0.06] *** | **244** | **4.7** | **0.22** | **0.00** | **0.00** | **0.13** | **0.96** | **39** |
| | ↓ -0.01 [-0.03, 0.01] | 370 | 1.1 | 0.02 | 0.00 | 0.00 | 0.13 | 0.95 | 75 |
| pH class | Non-significant | 666 | 0.4 | 0.00 | 0.00 | 0.00 | 0.20 | 0.93 | 136 |
| pH (min, org) | ↑ 0.13 [-0.39, 0.64] | 82 | 0.2 | 0.01 | 0.46 | 0.00 | 0.00 | 1.00 | 29 |
| | ↑ 0.11 [-0.06, 0.30] | 133 | 1.6 | 0.02 | 0.00 | 0.00 | 0.20 | 0.93 | 55 |
| Bulk density | ↑ 0.09 [-0.62, 0.81] | 280 | 0.1 | 0.01 | 0.00 | 0.00 | 0.16 | 0.95 | 38 |
| (min, org) | ↓ -0.26 [-2.62, 2.10] | 449 | 0.0 | 0.00 | 0.00 | 0.00 | 0.24 | 0.96 | 69 |
| Org layer depth | 0.00 [0.00, 0.00] | 485 | 0.3 | 0.00 | 0.00 | 0.00 | 0.25 | 0.96 | 78 |
| *Vegetation* | | | | | | | | | |
| Vegetation class | Non-significant | 624 | 5.5 | 0.04 | 0.00 | 0.00 | 0.19 | 0.92 | 136 |
| Net primary prod | 0.00 [-0.89, 0.88] | 721 | 0.0 | 0.00 | 0.00 | 0.00 | 0.20 | 0.93 | 136 |

[*]Significant results are highlighted in bold: **\* p<0.05; ** <0.01; *** <0.001.** Trends (p<0.1) are indicated with (\*).

*From top to bottom*: Metaregression model results testing the effects of context-specific environmental conditions on *ER Hedges SMD* ('context-dependencies'). *From left to right:* The type of environmental driver investigated; the specific driver used as predictor to test the effect on *ER Hedges SMD*s ('Driver'); the model results for each driver, testing whether *ER Hedges SMD*s are significantly influenced by the driver ('Slope'); the 'QE'-value of heterogeneity, as well as the 'QM'-value, reflecting importance of the driver and the McFadden's pseudo-R² ('R²'); the output of the multilevel model structure (represented by $\sigma^2_1$ and $\sigma^2_2$) and of the autoregressive model component (tau² and rho); and the sample size ('N' or number of datasets). Metaregression results are presented as the mean model estimate and 95% confidence intervals ('Slope (95%CI)'), with a down/upward arrow (↓/↑) indicating that higher values of the driver significantly decreased or increased *ER* responses to warming and the significance level based on *p*-values. Significant drivers and results are in **bold**.