## [Peer Review File · Nature]

Manuscript Title: Environmental drivers of increased ecosystem respiration in a warming tundra

Reviewer Comments & Author Rebuttals

Reviewer Reports on the Initial Version:

Referees' comments:

Referee #1 (Remarks to the Author):

This manuscript analyzes a new dataset of warming experiments across tundra ecosystems. This important effort compiled information from 56 open-top-chamber in situ warming experiments. The authors performed meta-analysis techniques and proposed that a rise in air temperature (~1.4oC) causes an increase in growing-season ecosystem respiration by 30% that persisted for at least 25 years. These results build on previous studies that have analyzed a selected number of experiments; therefore, this manuscript expands those analyses and updates and improves the estimates and observed ecosystem responses. The authors also identified potential relationships of ER with other soil physical factors and vegetation community composition. Then the authors performed an upscaling approach to present the global sensitivity of warming in tundra ecosystems. The authors compiled an important dataset, and the results are provocative. That said, there are some weaknesses in the strength of the results, and there is missing information on the underlying assumptions for some calculations. Therefore, the authors should revise the use of statistics and how uncertainties are propagated, especially for the upscaling approach. My main concern is that the statement of persistence of high ER needs to be better supported by the analysis and thus requires revision. Here I include a series of comments to improve the manuscript.

Major Comments

Line 148 clearly explains the overall objective of the manuscript. That said, it would be important to include testable hypotheses. Lines 103-146 provide a clear introduction to the problem and the knowledge gaps. Then, what are the hypotheses associated with the overall objective based on our current understanding? Are the results consistent with current expectations, or are they surprisingly different from what was expected?

Lines 197-200 – It is unclear where the value of 30% more ER comes from Fig. 2. This figure has six panels with positive and negative slopes. Clarification is needed to better explain where this number comes from, as it is a primary result of the manuscript.

Line 201 - Q10 value needs to include uncertainty; ideally as 95%CI

Lines 204-209 – I think the authors need to be careful about interpreting the p-value of this trend. There has been a long history of discussing how the scientific community uses this information (i.e., p-values) and problems with “p-hacking.” The challenge with this study is that the authors interpret

a “marginally significant” slope for ER response to warming across all 25 years as a weak positive trend. This positive trend is then interpreted and discussed in the manuscript as an essential effect despite a p-value of 0.049. In contrast, the supplementary materials show that the authors made arbitrary decisions to exclude potential confounding variables with “marginally significant” effects (e.g., flux measurement methodology; or lines 149-152 in Supplementary Discussion). Furthermore, supplementary figure 5 shows the classical example that a combination of different slopes (e.g., negative and positive trends) is not the same as when data are plotted together. One can envision different fits (e.g., polynomial), piecewise, or arguably no trend. Finally, Figure 3e shows that the CI of the 3 data points for experiments of 25-30 years do not show clear evidence of increasing ER in these long-term experiments, although this could be an artifact of the low sample size. Thus, the implications of these results depend on how data is fitted and interpreted.

The authors must consider the scientific and ethical interpretation of a marginally significant effect for a potential warming trend, especially for a publication under review in Nature. Is it too early to provide strong evidence for this trend? Is the interpretation scientifically sound, and is it ethically correct to move forward with such an interpretation? Lines 209-212 state that despite this weak trend, the authors speculate that ER responses to experimental warming will not decrease in the longer term (i.e., multi-decadal duration). This is a statement that the authors and Nature editors need to evaluate, considering the evidence provided and how “marginal results” are interpreted throughout the manuscript.

Upscaling of ER due to increase in temperature. This is an add-on to the manuscript with multiple underlying assumptions that must be explicitly listed in the methods. Furthermore, there are issues of uncertainty derived from the meta-analysis (i.e., uncertainty in the metaregression) and with the datasets used as covariates (i.e., uncertainty in the original estimation of the covariates, and then the uncertainty in upscaling from the native resolution of these datasets [e.g., from 1x1 km resolution to 5x5 km resolution]). I see the appeal of doing this exercise. Still, at this moment, there are many missing details for this analysis which arguably results in a “back of the envelope” calculation as is currently presented. The authors must clearly explain the methods and the assumptions and think about error propagation in the input variables and model. Finally, think about uncertainty beyond that calculated from the metaregression (which I think is presented in supplementary figure 7).

The website <http://arcticflux.org/> describes the goal, the team, and the locations. I strongly support the commitment by the authors to make the data and scripts available on Figshare or other data FAIR repositories.

Comments in detail.

Supplementary Figure 4 – Are all these slopes statistically significant? One plot seems to have only 3 points.

Line 231- arbuscular mycorrhizae is a type of mycorrhizae. Do you mean ectomycorrhizae?

Line 254 – The effect of total nitrogen is also “marginally significant”? Then how does this borderline

effect influence the interpretation in lines 257-262?

Lines 290-292 – This is not a surprising result as it is well-known that local biophysical factors influence the responses of ecosystems. If this is a novel result, the authors must prove that previous knowledge contradicts this. Or rephrase the statement by saying that the meta-analysis confirms expectations of how local soil conditions influence ecosystem responses.

Lines 297-301 – These statements are interesting, but the authors should consider the strength of the results, especially in light of Total Nitrogen.

Lines 302 – I find the statement “our new understanding of context-dependencies of warming effects” unclear. What exactly does this mean, and what is new in further applying the results? Something seems missing in understanding what the authors imply and how they used the information/data to move to the following research step. I think the information is in lines 318-325.

Lines 328-329 – I am not convinced about this statement because of the challenges and limitations in the data analysis. The interpretation is likely, but the results are not 100% convincing. It is more likely that non-linearities are the rule rather than a straightforward linear increase, as speculated by the authors. This is discussed in lines 334-342.

Line 344 – this is an important statement, but the challenge is identifying the sites/areas.

Line 354-356 – I fully agree with this statement, but I feel it is a weak way to end the manuscript as it is not a novel call or conclusion.

Lines 38-43 – I assume these measurements were with “dark chambers.” Please clarify.

Referee #2 (Remarks to the Author):

This paper reports on critically important and novel results from a large number of arctic and alpine experimental tundra sites covering a very wide geographic area. Quantifying the responses of soil respiration in the tundra is of great interest to a wide audience, and using detailed and geographically extensive experimental data to do so, as is done here, is vastly superior to other approaches. The results here are therefore of the highest interest and importance, and certainly have the potential to be Nature publishable. Enormous amounts of data were collected, and the authors have analyzed the results thoughtfully. The use of Hedges' g , weighted by precision, and of accepted meta-analysis protocols, are an excellent foundation for the analyses. Nevertheless, I think more can be extracted from this impressive data set. It was difficult not to get lost in the plethora of data points, things measured, and analyses, and it is very easy (and likely) for readers to misinterpret the results. I have a number of questions and suggestions in this regard that may improve the sharpness and accuracy of the analyses.

The overarching question here is, how did soil respiration in tundra respond to experimental increases in temperature, and the answer is, it increased a great deal, but how much depends on the

details. Fair enough, but I think the answer can be sharpened quantitatively with somewhat different methods, and a different graphical approach. The answer, “by a mean difference of 30%” is likely to be extrapolated and misinterpreted, and is, in fact, a very vague generalization.

As the authors emphasize, the magnitude of the response is heterogeneous, and depends on context and various soil and other covariates. I did not understand how the within-area heterogeneity was distinguished statistically from the among-area heterogeneity, or how the “hot spots” of sensitivity and enhanced responses were identified; clearly this is important. How was it determined that Arctic responses were larger than alpine responses, for instance, or that some areas of Siberia were particularly sensitive to warming? How was this distinguished from ambient warming that is also occurring?

Apparently the 30% is the mean across all years and sites (extended Figure 2), which to me did not make sense. There are several reasons I am not convinced this is the right way to summarize the results. First, the controls are warming during the duration of the experiment, by different amounts based on location, because especially Arctic tundra is warming rapidly. So, it might make more sense to use the controls from the earlier studies as a baseline, if that is possible (it may not be, I understand, depending on what data is available for which sites). In any case, the simultaneous warming of the controls should be considered. Second, there is a mention of using years within sites in the meta-regression, which is good, but these are trends (that is, the data are sequential), so it doesn't really make complete sense. If soil respiration is not an immediate response (it isn't), but rather the response is cumulative, somehow describing the shape(s) of that response over time is critical; just describing the duration of warming isn't adequate, either, because the actual year is also important (all responses in, say, 2000, or 2010, share some similarities, as warming, CO₂, rainfall, etc. change over time ambiently). Despite the fact that this is an enormous data set, a lot of information one would want isn't available and I recognize that this can't be modeled precisely, but somehow a sharper picture should be possible to extract, and a much clearer graphical presentation than ext Fig 2 should be created and included right up front in the paper (this to me is the most important data). Perhaps a timeline with each data set shown as a line graph would be clearer? Or a multi-figure graph could be drawn with one graph for each site with responses for each year shown with a single time line, and the graphs for the sites stacked, plus one line averaged by date across all available data across sites for that date? I wanted to see how the responses changed over time (the shapes, by years) for each area (averaged across the plots in that area). How to handle that statistically is admittedly challenging, but at least showing it graphically would be valuable.

It wasn't completely clear to me why Hedges' *d* was used instead of response ratio (as the results are reported as a ratio or percent), or how the “ratio of means” differs from a response ratio, or why for some estimates the “ratio of means” was used and for others Hedges' *g* was used (response ratios and Hedges' *g* are both perfectly acceptable, just different).

Using geographically estimated values for some parameters seems appropriate, but where there were other missing values, multiple imputation would provide a more accurate analysis than would omitting the sites that have missing data (it wasn't clear how missing data were handled). Multiple imputation is a relatively new approach in meta-analysis, but appears to be a substantial improvement with respect to missingness. The approaches taken to dealing with the mass of data

were courageous, but the one-at-a-time approach used in places is an older one that is lacking in important ways. One better approach for dealing with the complications of direct and indirect effects that is so apparent here is SEM meta-regression, pioneered by the social science statistician Mike Cheung (Singapore Univ.). This would be a more precise and elegant approach. In some cases, using multivariate meta-analysis (not meta-regression, but using something like NMDS, PCA, or the like) would help to reduce the dimensionality of the undoubtedly correlated factors instead of the one-at-a-time approach. Heterogeneity is included in places, but I^2 and Q should be reported more consistently and the method used for calculating τ^2 for the random effects models should be reported; clearly these data are highly heterogeneous. In fact, that heterogeneity is an important part of the story being told here, because it is biologically reasonable and interpretable (at least in part). Showing how accounting for the biologically meaningful differences in response reduces that statistical heterogeneity would be very valuable.

Referee #3 (Remarks to the Author):

Review by Ivan Janssens

The submitted manuscript analyzes an impressive dataset on passive warming experiments by OTCs, focusing on the environmental drivers explaining the variation of the ecosystem respiration (ER) response to the warming. The rationale for the study is that more insight is needed in the response of respiration to warming to obtain better estimates of the land carbon-climate feedback. The key finding is that across all experiments and independent of warming duration, there is a mean increase of ER of 30% for a mean warming of 1.4 °C. Experiments with lower nitrogen concentrations in the mineral soil exhibited a greater stimulation of ER by the placement of OTCs. An upscaling approach using global maps of the key drivers then revealed Western and Eastern Siberia and the Canadian Archipelago as hotspots of warming sensitivity.

While this analysis of the response of ER to the OTC placement across a large number of experiments is of course interesting and novel, there are multiple issues with the current version, through which I cannot assess whether or not this study is truly relevant.

I will list these issues first and thereafter provide a list of questions/comments that arose while reading the text.

Major issues:

1. This manuscript deals with warming responses, using the increase in air temperature induced by OTC placement. However, air temperature would only affect aboveground respiration (roughly 20-40% of ER) and not the part that is key to the land carbon-climate feedback. How did soil temperatures change? I was very surprised that Figure 1b, nor any of the Tables give this soil warming data. The majority of experiment placed the OTCs only in the growing season. Because of the passive warming and the vegetation cover in summer, I doubt that substantial soil warming was realized, at least not in the first month. By removing the chambers every August/September, any difference in soil temperature would be undone every Autumn to Spring period, in contrast to the

experiments where the OTCs remained in place. Surely this must have affected the soil warming intensity? By averaging out all RE measurements over the growing season, independent of the amount of soil warming, and not taking this into account in the analysis, substantial error may have been introduced. The manuscript also fails to report on the differences in snow cover and wind speed, that have huge impacts on vegetation and thus on ER as well.

2. This brings me to a major issue. The manuscript only reports ER fluxes; I guess also GPP data will be available but being spared for a follow-up paper. The problem is that GPP is a more important determinant for ER than air temperature and even soil temperature: Plants can only respire more if GPP increases to sustain the higher respiration; microbes that respire more deplete their food source, which results in a shrinking microbial population and eventually reduces microbial respiration to the level of soil carbon inputs under the warmed conditions. Thus: long-term sustained increases in ER must be accompanied by increased GPP and NPP. In this study, there are clear indications that the 30% increase in ER is more driven by a warming-induced productivity increase than directly by warming. For example, there is a clear tendency towards reduced bulk density in both organic and mineral layers, for which the most likely reason is increased soil organic matter inputs and associated microbial- and faunal activity. Bulk density may also have decreased because the OTCs provide warmer and wind-free shelters for soil animals. Total C stocks are not given, but C concentration did on average not decrease, which one would expect if stimulation of ER would be driven by the warming and not by increased GPP. Given that I believe that increased productivity is driving increased ER, plus that soil carbon stocks were not affected by the OTCs, I cannot see the relevance of the 30% increase in ER for the land carbon-climate feedback, which is how the authors currently sell the story.

3. The global scaling depends on the model found across these warming experiments, but does not extrapolate the uncertainty of the model. Then, this model is multiplied with an ISRIC dataset providing the spatial data for the model variables, which has gigantic uncertainties for the Arctic region, which could have been quantified given the available site data, but are also fully ignored. In addition, one of the drivers is not even available in the ISRIC dataset but is approximated. Also this proxy was not evaluated using the available empirical data and its uncertainty not extrapolated in the final product. Best remove this entire part of the manuscript or support it with a thorough uncertainty analysis.

More detailed comments:

-The duration effect should be studied within ecosystems, not across all experiments, to avoid other drivers confounding the analysis.

-Soil carbon and changes therein are crucial, but little to no methods were given and changes not reported. Was it sampled uniformly?

-Fig 1: show the soil T response too

-Fig 2: N and C concentrations are given but soils have become lighter by the OTC treatment. Did you account for this also in the sampling? Sampling at the same depth is not appropriate if they have loosened.

-Fig 2: in the graphs you give multiple ER data for single driver observations. Please only show one ER per driver observation.

-Line 260: this hypothesis could be tested by comparing the change in SOC stock between N-poor and N-rich sites...

-Figure 3: was there an interaction between vegetation type and soil warming? Did soil warming

differ between growing season-only and full year applied infrastructures?

-Line 294: surely, the depth from the surface where the mineral soil starts must have influenced the importance of mineral soil N and C/N. Was this analyzed? Also, the role of N turnover is hypothesized, probably correctly, but why wasn't a meta-analysis conducted on the N turnover directly? I assume these data were available at many sites, given that N turnover is well known to be a key determinant of the ecosystem response to warming.

Fig 5: the text states that the ER response extends to >100%, then best expand the scale to 100%; although as stated above I have little confidence that this analysis bears any meaning

Line 327: this is not true. It depends on the persistence of an imbalance between C inputs and respiratory losses, not on the persistence of stimulated ER.

Line 334: explaining why some of the long-term running sites show increased ER while others show declining ER to me would be a very informative analysis

Line 345: providing modelers with a more robust understanding would require you to show the responses of GPP and NPP as well.

SI Line 27: the differences in OTC height is not slight; the effect on wind penetrating to the soil surface must be huge.

Line 41: this description does not suffice. Imagine that the OTCs were placed on day X and measurements started on day X+1, then likely vegetation profited from the reduced cooling at night, but the soil would not have changed. Warming the soil, especially the underlying mineral soil, with passive warming as applied here, must take many weeks/months. Removing the OTCs every autumn undoes the warming, such that even after 10 years this artefact remains. Details should be given on this.

Unit for NPP is wrong; define CAVM; define how soil environmental variables were measured (what depth?); methods should allow the reader to reproduce the analysis, but apart from explaining SMD, I have no idea what exactly was done. I assume you used the difference in seasonal mean ER from an experiment? This gives the effect size, but how did you deal with the differences in warming throughout the season, across soil depths, and across years?

I did not fully understand the statistical approach, so some of the comments above may be unfair.

Response letter to the manuscript
*‘Environmental drivers of persistently increased ecosystem
respiration with warming in the tundra’*

Referee #1

1. This manuscript analyzes a new dataset of warming experiments across tundra ecosystems. This important effort compiled information from 56 open-top-chamber in situ warming experiments. The authors performed meta-analysis techniques and proposed that a rise in air temperature (~1.4oC) causes an increase in growing-season ecosystem respiration by 30% that persisted for at least 25 years. These results build on previous studies that have analyzed a selected number of experiments; therefore, this manuscript expands those analyses and updates and improves the estimates and observed ecosystem responses. The authors also identified potential relationships of ER with other soil physical factors and vegetation community composition. Then the authors performed an upscaling approach to present the global sensitivity of warming in tundra ecosystems. The authors compiled an important dataset, and the results are provocative. That said, there are some weaknesses in the strength of the results, and there is missing information on the underlying assumptions for some calculations. Therefore, the authors should revise the use of statistics and how uncertainties are propagated, especially for the upscaling approach. My main concern is that the statement of persistency of high ER needs to be better supported by the analysis and thus requires revision. Here I include a series of comments to improve the manuscript.

- We thank Reviewer #1 for the constructive comments. We appreciate their enthusiasm and the thoroughness of the review. We believe our revised manuscript addresses the remarks that were raised, as outlined below. Overall, we addressed the reviewer’s comments as follows:
 - ‘there is missing information on the underlying assumptions for some calculations’;
‘should revise the use of statistics and how uncertainties are propagated, especially for the upscaling approach’
 - We thoroughly revised our analyses, including the Methods, Extended Data, and Supplementary Information, to explain the methodologies used throughout our manuscript better. We added extra tables and figures to address the uncertainties mentioned by the reviewer. Specifically, for the **meta-regression models** evaluating relationships between environmental drivers and ER response rates, we performed a sensitivity analysis to address the uncertainty regarding the appropriate sample size to use (**Supp. Discussion #4, Supp. Fig. 4**). For the upscaling, we fully revised the work flow and performed a detailed uncertainty analysis as described in the **Methods** and **Supp. Methods #3**. See our reply to your comment #5 for details.
 - For the upscaling, we also specifically explored how well the distribution of our dataset compares to the distribution of gridded soil data used for upscaling. To do that, we created a scatter plot for TN and C:N-ratio where we overlaid both the observed (from our dataset) and gridded values (gridded soil dataset used for upscaling (new **Supplementary Fig. 9**)). It shows that the observed

values from our dataset are slightly biased towards low TN values, but otherwise the observed values cover the main concentration of TN-C:N-ratio distribution rather well.

- Based on the above changes, we believe we now sufficiently describe and take into account the uncertainties throughout our different analyses, and trust these results to be robust.
- ‘statement of persistency of high ER needs to be better supported by the analysis and thus requires revision’
 - We agree that the evaluation of temporal patterns needed improvements, incl. a streamlined approach throughout the manuscript to assess the different aspects of temporal patterns. We therefore thoroughly revised how we analyze the ER response over time (i.e. how it changes with experimental warming duration). We believe we now adequately demonstrate the persistency of the ER response with the new **Ext. Table 5a** and new **Figure 3**. Additionally, addressing similar concerns by Reviewer #2, we looked at within-experiment temporal patterns, reporting these results in **Supp. Discussion #6** and **Supp. Fig. #5**. We do not present the within-experiment patterns in the main text however, and we think they should be interpreted with caution, because of the reasons explained in **Supp. Discussion #6** (in short, low sample sizes within age classes, and lack of sufficiently long within-experiment time series across age classes).

Major Comments

2. Line 148 clearly explains the overall objective of the manuscript. That said, it would be important to include testable hypotheses. Lines 103-146 provide a clear introduction to the problem and the knowledge gaps. Then, what are the hypotheses associated with the overall objective based on our current understanding? Are the results consistent with current expectations, or are they surprisingly different from what was expected?
 - We agree with the reviewer that adding testable hypotheses improves the storyline of our manuscript. We included the following specific hypotheses now on Lines (L.) 153 in the section *Objectives and study design*:
 - *We hypothesize that (i) there is an overall positive ER response to warming (both R_a and R_h), but the response magnitude varies across time and space. We further hypothesize that differences in the magnitude are driven (ii) indirectly by variation in warming-induced changes in local environmental conditions, as well as by (iii) context-dependent variation in environmental conditions.”*
3. Lines 197-200 – It is unclear where the value of 30% more ER comes from Fig. 2. This figure has six panels with positive and negative slopes. Clarification is needed to better explain where this number comes from, as it is a primary result of the manuscript.
 - The referenced figure on the mean 30% change of ER with warming was former **Ext. Figure 2** (see below this comment), where we show the ER Hedges Standardized Mean Difference (SMD) response and 95% confidence intervals, averaged across all 136 datasets (left: MEAN),

and for each individual dataset. We suspect that the reviewer was instead looking at former **Figure 2** in the main text (see below), when referring to the six panels with positive and negative slopes.

- We agree with the reviewer however that more clarification was needed on where the 30% change in ER came from. This number was calculated from the log-transformed Ratio Of Means (ROM) of 0.26, i.e. the secondary effect size used for our meta-analysis. Since the figure showed the results based on the primary effect size (Hedges SMD), the percentage change could not be derived from it. We have clarified this now as follows:
 - The Methods (section *Meta-analysis*) includes an explanation of how and why we calculate a mean percentage change value from the mean ROM as secondary effect size. L. 49 reads: “We used *Hedges’ g Standardized Mean Difference [SMD]* as *primary effect size across all models*, and the log-transformed *Ratio of Means [ROM]* as *additional effect size to quantify percentage change in ER with warming*. ... The *ROM* was calculated as [mean growing season average ER in warmed plots / mean growing season average ER in control plots] which we transformed to a mean percentage increase as $100*(\exp(\text{ROM})-1)$, i.e. a more intuitive measure of changes in ER.”. Please note that ROM and $\ln\text{RR}$ (log response ratio) are used interchangeably in the literature and refer to the same effect size.
 - The new **Ext. Table 4** explains how mean percentage change is calculated from the ROM: “Percentage change = $100*(\exp(\text{ROM})-1)$ ”. When referring to the mean 30% change in the main text, we refer to this table to avoid confusion. **Ext. Table 1** now also provides all ROM and mean percentage change values per dataset, in addition to the Hedges SMD.
 - We include what was formerly Extended Figure 2 as **Fig. 2** in the main text to demonstrate our mean meta-analysis estimate (i.e. ER Hedges SMD), as well as the variability in the ER response across experiments and datasets. This demonstrates the answer to hypothesis number (i), i.e. we find an overall increase in ER with warming (mean Hedges = 0.57 ~ mean ROM = 0.26 ~ mean percentage change = 30%, **Ext. Table 4**), but the ER response varies across space and time. We believe **Fig. 2** also helps familiarize readers with the structure of the data, i.e. the different studies and years, that were used throughout our analyses.

Former Extended Figure 2

46 **Figure 2:** Effects of experimental open-top-chamber (OTC) warming on ecosystem respiration (ER) response rates.

47 Experimental warming increased ecosystem respiration across the tundra biome but there was significant heterogeneity across the datasets.

48 Effect of OTC warming on ecosystem respiration (ER) response rates (Hedges standardized mean differences or SMD calculated as [mean ER of the warmed plots - mean ER of the control plots]

49 / pooled standard deviation) across the 136 datasets (i.e. experiment x flux year combinations). With a black diamond, the Mean estimate ('est') of the response rate (pooled Hedges SMD) and

50 95% confidence intervals (CIs, error bars) across the 136 datasets are provided as well as the Q-value ('Q') for heterogeneity, and p-value ('p-val') from the meta-analysis. Black dots represent

Former Figure 2

7 Figure 2: Warming-induced changes in soil conditions and vegetation community drive respiration response.

4. Line 201 - Q10 value needs to include uncertainty; ideally as 95%CI
 - We included an uncertainty interval on L. 212. We also included our calculations for the Q10 values in **Supp. Methods #2**. Please note that we discovered an error in the formula used of Q10, so the results have changed in comparison to the previous version of the manuscript. The new Q10 values did not change anything in the interpretation of the results.

5. Lines 204-209 – I think the authors need to be careful about interpreting the p-value of this trend. There has been a long history of discussing how the scientific community uses this information (i.e., p-values) and problems with “p-hacking.” The challenge with this study is that the authors interpret a “marginally significant” slope for ER response to warming across all 25 years as a weak positive trend. This positive trend is then interpreted and discussed in the manuscript as an essential effect despite a p-value of 0.049. In contrast, the supplementary materials show that the authors made arbitrary decisions to exclude potential confounding variables with “marginally significant” effects (e.g., flux measurement methodology; or lines 149-152 in Supplementary Discussion). Furthermore, supplementary figure 5 shows the classical example that a combination of different slopes (e.g., negative and positive trends) is not the same as when data are plotted together. One can envision different fits (e.g., polynomial), piecewise, or arguably no trend. Finally, Figure 3e shows that the CI of the 3 data points for experiments of 25-30 years do not show clear evidence of increasing ER in these long-term experiments, although this could be an artifact of the low sample size. Thus, the implications of these results depend on how data is fitted and interpreted.

The authors must consider the scientific and ethical interpretation of a marginally significant effect for a potential warming trend, especially for a publication under review in Nature. Is it too early to provide strong evidence for this trend? Is the interpretation scientifically sound, and is it ethically correct to move forward with such an interpretation? Lines 209-212 state that despite this weak trend, the authors speculate that ER responses to experimental warming will not decrease in the longer term (i.e., multi-decadal duration). This is a statement that the authors and Nature editors need to evaluate, considering the evidence provided and how “marginal results” are interpreted throughout the manuscript.

- As the reviewer suggests, we acknowledge the need to consider a scientific, ethical, and at the same time consistent interpretation of effect ‘sizes’ (~p-values) across our regression models. We agree that there was a lack of consistency in our initial manuscript regarding how we

interpreted the various models results, e.g. regression models on environmental drivers vs. temporal patterns vs. methodological drivers. For instance, we had indeed not taken into account the methodological marginally significant driver ($p < 0.1$) but *did* report and interpret a marginally significant effect of warming duration on the ER response. The revisions of the manuscript address these points now by consistently interpreting significant results as $p < 0.05$. We respond to the different points raised by the reviewer here as follows:

- A) ‘Lines 204-209 – I think the authors need to be careful about interpreting the p-value of this trend. There has been a long history of discussing how the scientific community uses this information (i.e., p-values) and problems with “p-hacking.” The challenge with this study is that the authors interpret a “marginally significant” slope for ER response to warming across all 25 years as a weak positive trend. This positive trend is then interpreted and discussed in the manuscript as an essential effect despite a p-value of 0.049. In contrast, the supplementary materials show that the authors made arbitrary decisions to exclude potential confounding variables with “marginally significant” effects (e.g., flux measurement methodology; or lines 149-152 in Supplementary Discussion).’
- We acknowledge the concerns in terms of consistency in statistical interpretation of the meta-regression effects throughout the manuscript. In our revised version, we streamlined the interpretation of significant effects, specifically mentioning on L. 188 in the Methods that “*We considered and interpreted results as significant in our study if $p < 0.05$, as is commonly done in meta-analyses*” As mentioned, this threshold is commonly used in meta-analyses¹⁻³ and deemed appropriate for our purposes. We no longer interpret marginally significant effects throughout the manuscript (**Fig. 3-5, Ext. Table 3-5**). Hence, our interpretation and conclusion in the main text is only based on effects with p-value < 0.05 . Moreover, our sensitivity analysis using different inclusion criteria for environmental data showed that the results were robust (see **Supplementary Discussion #4**).
 - During the revisions, we addressed the specific examples of (marginally) significant effects mentioned by the reviewer as follows:
 - **Duration:** we thoroughly revised our analysis of temporal patterns, using meta-regression instead of linear regression models. This led to the ‘*weak positive trend in ER response across all 25 years ($p < 0.049$)*’ disappearing, and instead we find no significant trend across all years (*across age classes, Ext. Table 5a*). Further, the temporal patterns *within* age classes now clearly show a non-linear trend over time. Taking together all our results from the assessment of temporal patterns, we believe our findings now appropriately justify our statement on persistency of the ER increase with warming over time (L. 230 of main text, **Ext. Table 5a, Fig. 3**). After these revisions, our earlier conclusion of a persistent positive effect of ER with warming and no indication of waning in the long term, still remains solid.
 - **Machine Type:** we no longer interpret, but do report the ‘marginally significant’ (p-value $0.05 < p < 0.1$) methodological drivers of ER responses as *trends* in **Supp. Fig. 3**. We explain the reasoning for, and methodology behind this supplementary analysis in **Supp. Discussion #2** in detail.

- B) ‘Furthermore, supplementary figure 5 shows the classical example that a combination of different slopes (e.g., negative and positive trends) is not the same as when data are plotted together. One can envision different fits (e.g., polynomial), piecewise, or arguably no trend. Finally, Figure 3e shows that the CI of the 3 data points for experiments of 25-30 years do not show clear evidence of increasing ER in these long-term experiments, although this could be an artifact of the low sample size. Thus, the implications of these results depend on how data is fitted and interpreted.’
- We acknowledge that these results on temporal patterns were not clear enough yet, and therefore thoroughly revised these. We clarify and visualize the new analyses and results on L. 222-233 in the main text, in **Fig. 3** and **Ext. Table 5a**. Based on these revisions, we believe our statement on persistence over time still holds. Our new approach uses meta-regression models investigating effects of warming duration across and within age classes. The specific findings that support the persistency of the ER response over time are as follows (highlighted in **bold**):
“The mean ER response remained positive and did not differ significantly across the four age classes (Fig. 3a), hence the positive warming effect persisted over time. Investigating the temporal patterns further showed indications for non-linear, positive responses over time (Fig. 3a, b, Ext. Table 5a): a decrease in the magnitude of the positive ER response during 5-9 years of warming ($Q_m=63$, $p<0.001$, $N=28$, Fig. 3b) was followed by an increasing magnitude during 10-14 years of warming ($Q_m=5.6$, $p<0.05$, $N=15$, Fig. 3b). Combined with the lack of change in the magnitude of the ER response over time during 0-5 years and >15 years of warming, the overall slope across all years was non-significant, indicating the persistency of the positive ER response to warming (Fig. 3a, Ext. Table 5a). While the ER response to experimental warming may thus falter around the end of the first decade of warming, our data do not provide evidence that ER responses to experimental warming wane in the longer term (i.e., multi-decadal duration).”
 - The separation into four age classes and investigating effects of duration within/across age classes (**Fig. 3, Ext. Table 5a**) proved a suitable approach to look at possible non-linear temporal patterns. It allowed us to deal with the high errors (large CI) for longer-term experiments (as the reviewer points out). Namely, by grouping all datasets with duration of warming greater than 15 years, we have a sufficiently balanced number of datasets across each age class, for our model results to be robust (**Fig. 3**: $N=70$, 28, 15, 23 for age class 1, 2, 3, 4).
 - We assume the reviewer also referred to possible differences in trends when plotting/modelling across vs. within experiments. Please see our response to comment #5 from Reviewer #2 for this, which we interpreted as a similar comment.
- C) ‘The authors must consider the scientific and ethical interpretation of a marginally significant effect for a potential warming trend, especially for a publication under review in Nature. Is it too early to provide strong evidence for this trend? Is the interpretation scientifically sound, and is it ethically correct to move forward with such an interpretation? Lines 209-212 state that despite this weak trend, the authors speculate that ER responses to experimental warming will not decrease in the longer term (i.e., multi-decadal duration). This is a statement that the authors and Nature editors need to evaluate, considering the evidence provided and how “marginal results” are interpreted throughout the manuscript.’

- See our response to part B) for a detailed explanation on this. We now improved the temporal analysis and thank the reviewer for pointing this out. We trust that our revised analyses and results now clearly demonstrate persistency in the response of ER to warming. That is, all age classes have a mean positive ER response. However, we observed differences in the relation between age and ER response in each age class. We made this more clear in the main text (L. 224-231). An apparent faltering in the response between 5-9 years of warming, followed by an increase in the response magnitude between 10-14 years. See **Fig. 3, Ext. Table 5a** and interpretation of these results on L. 377-393.
6. Upscaling of ER due to increase in temperature. This is an add-on to the manuscript with multiple underlying assumptions that must be explicitly listed in the Methods. Furthermore, there are issues of uncertainty derived from the meta-analysis (i.e., uncertainty in the metaregression) and with the datasets used as covariates (i.e., uncertainty in the original estimation of the covariates, and then the uncertainty in upscaling from the native resolution of these datasets [e.g., from 1x1 km resolution to 5x5 km resolution]). I see the appeal of doing this exercise. Still, at this moment, there are many missing details for this analysis which arguably results in a “back of the envelope” calculation as is currently presented. The authors must clearly explain the Methods and the assumptions and think about error propagation in the input variables and model. Finally, think about uncertainty beyond that calculated from the metaregression (which I think is presented in supplementary figure 7).
- We fully revised the upscaling workflow, addressing not only the uncertainty from the meta-regression model but also from the ISRIC dataset and from using a proxy for C:N-ratio. The revised method is explained in detail in the Methods (*Spatial upscaling*) and summarised in a flow diagram in **Supp. Methods #3**. The main changes to the method are:
 - We now use the native resolution (250 m) soil data, for which ISRIC provides uncertainty estimates (which are not yet available for other resolution (i.e. 1 km and 5 km products). This allowed us to estimate the uncertainty originating from the ISRIC data. Due to the large study area (10 million km²), we still needed to aggregate the soil data to 1 km resolution as even with this resolution we needed a supercomputer to run the uncertainty analysis.
 - We used the reported C:N-ratio and reported TN and TC from our database (**Supp. Fig 8**) to estimate the uncertainty originating of the proxy using TN and SOC to derive the C:N-ratio, and incorporated this uncertainty in our Monte Carlo uncertainty analysis.
 - In the Monte Carlo uncertainty analysis, we created 100 pairs of TN and C:N-ratio values for each 1 km² grid cell using a truncated multivariate random normal distribution, using the mean and standard deviation of these variables as provided by ISRIC.
 - We provide information about the degree of extrapolation of the variables in our upscaling model (**Supp. Fig. 9**).
 - We then used the resulting 100 pairs of TN and C:N-ratios as input for our ROM model for each 1 km² grid cell, resulting in 100 predicted %ER changes with associated standard errors. These were then combined to get the mean predicted percentage change in ER induced by 1.4°C warming and a combined measure of standard deviation.
 - We have revised our upscaling, so that it is more nuanced, and is meant to give an idea of the magnitude of ER respiration increases across different tundra areas. On L. 348, it now reads: “*Although these numbers are crude estimates, they suggest an order of*

magnitude of ER respiration increase that can be expected for the tundra as a whole, and clearly show spatial differences in sensitivity.”

7. ‘The website <http://arcticflux.org/> describes the goal, the team, and the locations. I strongly support the commitment by the authors to make the data and scripts available on Figshare or other data FAIR repositories.
 - We agree on the importance of sharing data and scripts on global, common repositories. As mentioned at the time of submission, we plan to make all data, code, and figures available on Figshare upon publication. During the review process, they can be accessed through the link to our shared folder. See the *Data and Code availability* statements for details.
 - Please note that we considerably revised our database website, including also a section on specific projects performed with the database, where we plan to describe where the data, code, and figures from each project publication can be found. The updated website location is <http://www.tundrafluxdatabase.com>.

Detailed comments

8. Supplementary Figure 4 – Are all these slopes statistically significant? One plot seems to have only 3 points.
 - These figures (now thoroughly updated) showed a selection of effect plots from our meta-regression models testing the influence of indirect warming effects and context-dependencies. With these figures, we aimed to show a specific aspect of our data, namely that for some environmental drivers (soil, vegetation, microbial community), the same data (same colour) could be linked to more than one ER data point from different years. We realize this was not clear, and not sufficiently accounted for in our previous analyses. Please see **Supp. Discussion #4** where we describe this data aspect in more detail now, and explain how we assessed the potential implications of this for our results.
 - To respond to the specific question raised by the reviewer: not all these slopes were statistically significant, and the regression lines and significance levels themselves were actually not relevant for what we wanted to show. That is, that there was no apparent influence on the meta-regression results due to the fact that measurements for several environmental drivers may have (i) differed in exact years from the year in which the ER measurements were taken, and (ii) been replicated for multiple ER data points. We believe that our inclusion of **Supp. Discussion #4**, describing these aspects in detail, combined with a thorough sensitivity analysis of which results are shown in **Supp. Figure 4**, present this better now in the revised manuscript. In sum, we believe that results of our sensitivity analysis show that the significant results based on the full dataset model are robust and valid to present in our main text.
9. Line 231- arbuscular mycorrhizae is a type of mycorrhizae. Do you mean ectomycorrhizae?
 - Since we decided not to interpret results that had a p-value between 0.05-0.1 in the revised manuscript, this statement was removed (**Ext. Table 5b**). However, to respond to the reviewer’s question, we were indeed highlighting the specific type of mycorrhizae i.e. arbuscular mycorrhizae or AM, which typically associate with graminoids.

10. Line 254 – The effect of total nitrogen is also “marginally significant”? Then how does this borderline effect influence the interpretation in lines 257-262?
- In our revised manuscript, additional gap-filling based on suggestions from Reviewer #3 led to the effect of TN becoming clearly significant with p -value = 0.01*, making this comment no longer an issue. See the specific changes in the results below.
 - submitted results: Ext. Table 4: regression estimate = -0.63; [-1.24, -0.01]; p -val = 0.046*; $Q_m=4.0$; number of datasets=36
 - revised results: Ext. Table 5c, Fig. 5a: regression estimate = -0.59; [-1.06, -0.13]; **p -val = 0.01***; $Q_m=6.3$; number of datasets=43
 - Thus, we remain convinced about our statement on L. 283 reading “*Greater ER increases with warming in nutrient-poor sites may be linked to changes in stimulated belowground C allocation by the plants and subsequent soil priming by root leachates.*”
11. Lines 290-292 – This is not a surprising result as it is well-known that local biophysical factors influence the responses of ecosystems. If this is a novel result, the authors must prove that previous knowledge contradicts this. Or rephrase the statement by saying that the meta-analysis confirms expectations of how local soil conditions influence ecosystem responses.
- We revised our statement on L. 318 so that it reads as a confirmation of our hypothesis (ii) and (iii), specifically for soil conditions, instead of a result that was unsurprising or already known.
12. Lines 297-301 – These statements are interesting, but the authors should consider the strength of the results, especially in light of Total Nitrogen.
- See our response to your comment #9 for this. The strength of the result for total nitrogen increased because of gap filling for the revised manuscript, so we believe these statements are still, and even more so, valid now.
13. Lines 302 – I find the statement “our new understanding of context-dependencies of warming effects” unclear. What exactly does this mean, and what is new in further applying the results? Something seems missing in understanding what the authors imply and how they used the information/data to move to the following research step. I think the information is in lines 318-325.
- We rephrased the statement on L. 336 to clarify which context-dependencies we refer to and how they were used to do our spatial upscaling.
14. Lines 328-329 – I am not convinced about this statement because of the challenges and limitations in the data analysis. The interpretation is likely, but the results are not 100% convincing. It is more likely that non-linearities are the rule rather than a straightforward linear increase, as speculated by the authors. This is discussed in lines 334-342.
- We believe our revised statement that “*we found sustained ER increases across 25 years of experimental warming*” is correct. When we say ‘sustained’, we are not speculating that the trend over time is linearly increasing, although the phrasing from the initial manuscript: ‘weak positive trend’ might have been what the reviewer pointed at here. We rephrased this section (L. 222-233), and highlight in the revised manuscript that:

- (i) ER responses were significantly positive (above zero) for each of the four duration age classes (**Ext. Table 5a, Fig. 3a**), implying that an increased ER due to warming is persistent across 25 years of warming duration.
- (ii) There is no significant linear effect of experimental warming duration across the 25 years (**Ext. Table 5a**).
- (iii) A decreasing trend in ER response rate to warming between 5-9 years of warming is compensated by an increasing trend between 10-14 years (**Ext. Table 5b, Fig. 3b**). This third result relates to the specific point on non-linearities in the temporal trend mentioned by the reviewer.
- → Overall, our data suggests that the ER response to warming does not diminish in the longer term across at least 2 decades of warming. We believe we clearly demonstrate now the effects of duration on ER response in a clear and consistent way (**Ext. Table 5b and Fig. 3b**).

15. Line 344 – this is an important statement, but the challenge is identifying the sites/areas.

- We revised this statement as a whole on L. 367, whilst revising the final paragraphs of the manuscript based on our updated results. The initial statement did not sufficiently bring forward that we actually attempted to increase this mechanistic understanding with our own results, i.e. the meta-regression results and spatial upscaling. With meta-regression, we were able to investigate the causes for spatial variability in the ER response to warming, and with the spatial upscaling, we were able to assess, albeit roughly, which sites would be more prone to warming-induced ER increases than others. The reviewer is right in stating that the challenge is in identifying the sites/areas more prone than others, but this is exactly what we aimed to demonstrate with our upscaling results. Namely, experimental field data was used to improve our mechanistic understanding on spatial sensitivity. This understanding can be used to improve regional climate models for the tundra biome.

16. Line 354-356 – I fully agree with this statement, but I feel it is a weak way to end the manuscript as it is not a novel call or conclusion.

- As part of reworking the final paragraph, this statement is now moved upwards on L. 363, such that the final paragraph is stronger and highlights the impact of our findings.

17. Lines 38-43 – I assume these measurements were with “dark chambers.” Please clarify.

- Indeed, all experiments except for one used dark or opaque chambers to measure ER. For experiment ‘ALA_1’, which had automated chambers, all measurements were done with clear chambers, since switching chambers on an automated system would require a lot more infrastructure. ALA_1 then distinguished between ER and NEE measurements based on photosynthetically active radiation (PAR) measurements. Flux measurements when PAR was smaller than $5 \mu\text{mol m}^{-2} \text{s}^{-1}$ were considered nighttime or ER measurements. We explain this now on L. 43-46 in the Methods. See also **Supp. Discussion #2** for how we checked whether this difference in chamber measurements (i.e. different ‘timing’ of ER measurements) influenced our results. We found no apparent influence of the timing on the ER response.

Referee #2

1. This paper reports on critically important and novel results from a large number of arctic and alpine experimental tundra sites covering a very wide geographic area. Quantifying the responses of soil respiration in the tundra is of great interest to a wide audience, and using detailed and geographically extensive experimental data to do so, as is done here, is vastly superior to other approaches. The results here are therefore of the highest interest and importance, and certainly have the potential to be Nature publishable. Enormous amounts of data were collected, and the authors have analyzed the results thoughtfully. The use of Hedges' g , weighted by precision, and of accepted meta-analysis protocols, are an excellent foundation for the analyses. Nevertheless, I think more can be extracted from this impressive data set. It was difficult not to get lost in the plethora of data points, things measured, and analyses, and it is very easy (and likely) for readers to misinterpret the results. I have a number of questions and suggestions in this regard that may improve the sharpness and accuracy of the analyses.
 - We appreciate the positive feedback on our work, the valuation of our data collection effort and approach to analysing the data. We are equally enthusiastic about the dataset and apparent robustness of the statistical approach. We believe that the additional sensitivity analyses that we have done demonstrate this robustness of the Methods and results. We further hope that we have elucidated our drivers/measurements and the different steps in analysis in this revised version, such that results would not be misinterpreted but instead better understood.
 - We do want to stress that our study evaluates ecosystem respiration or ER instead of soil/microbial respiration. Soil respiration, referred in the manuscript as microbial or heterotrophic respiration is only a component of ER, while the other component that makes up ER is plant or autotrophic respiration. We clarify this key distinction in the main text on L. 110, and believe it should be clearer as well by including supplementary meta-analysis results on these two separate components in response to warming, which were done in response to Reviewer #3's comments (**Ext. Table 4b, Supp. Discussion #1**).
2. The overarching question here is, how did soil respiration in tundra respond to experimental increases in temperature, and the answer is, it increased a great deal, but how much depends on the details. Fair enough, but I think the answer can be sharpened quantitatively with somewhat different Methods, and a different graphical approach. The answer, "by a mean difference of 30%" is likely to be extrapolated and misinterpreted, and is, in fact, a very vague generalization.
 - Indeed, we agree that the 30% ER increase should be more nuanced by highlighting at the same time the significant heterogeneity in the response. We have nuanced this by (i) always including the confidence interval in addition to the mean response, and (ii) including **Fig. 2** now in the main text, which clearly shows the **variability** in the response alongside the **mean** response. (iii) mentioning that our data was based on growing season measurements. Our analyses and manuscript aim exactly at quantifying and attributing this heterogeneity in ER response to warming to carefully selected environmental drivers, so that this variation can be taken into account in e.g. climate models.
 - Thus, we believe our mean 30% increase in ER with warming can still be brought forward as a key result, as it gives an idea of the impact and strong, positive overall response across all datasets, which supports our first hypothesis (*Objectives and study design*, **Ext. Table 4a**). Still, we agree that caution is needed with such 'impact' numbers, which could be used in the wrong

way, and therefore specifically include the following statement on L. 220: “*While the ER response to warming was overall positive and strong, there was significant heterogeneity across the datasets (Q -value 731, $p < 0.001$, Fig. 2, Ext. Table 4), implying that ER responses to warming vary across time and/or space*”.

3. As the authors emphasize, the magnitude of the response is heterogeneous, and depends on context and various soil and other covariates. I did not understand how the within-area heterogeneity was distinguished statistically from the among-area heterogeneity, or how the “hot spots” of sensitivity and enhanced responses were identified; clearly this is important. How was it determined that arctic responses were larger than alpine responses, for instance, or that some areas of Siberia were particularly sensitive to warming? How was this distinguished from ambient warming that is also occurring?

- We respond to the different questions of the reviewer point per point, as they relate to different steps in our analyses. In each response, we highlight how we clarified these methodological aspects in our revised version.
- A) Distinction between Within-site heterogeneity vs. Among-area heterogeneity
 - We used within-site and among-area heterogeneity in a somewhat different way than that the reviewer assumes. We checked for consistency in wording to prevent confusion about this. In the OTC warming experiments, and subsequently in our upscaling analyses, ‘**within-site**’ (experiment) heterogeneity quantifies how warming has affected environmental drivers (e.g. local soil moisture) per experiment. This heterogeneity was quantified as effect sizes of ecological factors in warmed vs. control plots per experiment, and is what we refer to as ‘indirect warming effects’ throughout the manuscript. See **Ext. Table 5b** for an overview of these factors (SMD climate, soil, vegetation, microbial) and a revised explanation in the Methods (L. 77 and following) and **Supp. Methods #1**. Importantly, the within-site heterogeneity (as in variation between replicates) is taken into account when calculating effect sizes.
 - In contrast, the ‘**among-area**’ (across-experiments) heterogeneity reflects the variability in environmental drivers among each experiment or (larger) site, depending on the specific driver. Most of these measurements of among-area environmental conditions, were quantified based on field measurements that we obtained from the control plots. We have clarified the different levels at which these specific environmental drivers were collected in more detail now in **Supp. Methods #1**.
- B) Hotspots of sensitivity and enhanced responses. How was it determined that arctic responses were larger than alpine responses, for instance, or that some areas of Siberia were particularly sensitive to warming?
 - This question showed that the upscaling approach was not entirely clear. We hope that by extending the methodology of our upscaling (which brought forward the results the reviewer mentions), and by adding an additional flowchart that summarizes the method (**Supp. Methods #3**) better explains how these results were obtained.
- B) Influence of ambient warming
 - Please see the next comment for a detailed response on this.

4. Apparently the 30% is the mean across all years and sites (extended Figure 2), which to me did not make sense. There are several reasons I am not convinced this is the right way to summarize the results. First, the controls are warming during the duration of the experiment, by different amounts based on location, because especially arctic tundra is warming rapidly. So, it might make more sense to use the controls from the earlier studies as a baseline, if that is possible (it may not be, I understand, depending on what data is available for which sites). In any case, the simultaneous warming of the controls should be considered.
- Please see our response to your comment #3 on the 30% value.
 - In response to the ambient/background warming effects, we do not believe this confounded our results because of the following.
 - First, we use ER Hedges SMD values, i.e. mean differences between ER in the warmed and control plots (divided by pooled standard deviation) for each dataset, hence our response reflects the *additional* ER for each experiment and year exceeding any potential ambient warming effect that may have occurred. We argue that, by focusing on ER response rather than absolute ER values, we overcame the variation that ambient warming potentially could have caused.
 - Second, calendar year itself did not affect the ER response (meta-regression: p-val=0.6 (non-sign), $Q_m=0.4$).
 - Third, the reviewer's suggestion to use 'controls from the earlier studies as a baseline' is interesting, but in our opinion not feasible because baseline data is lacking in most studies, and cannot be transferred across studies (**Ext. Table 1-2**). We believe an analysis of changes in baseline ER would be outside the scope of this manuscript.
5. Second, there is a mention of using years within sites in the meta-regression, which is good, but these are trends (that is, the data are sequential), so it doesn't really make complete sense. If soil respiration is not an immediate response (it isn't), but rather the response is cumulative, somehow describing the shape(s) of that response over time is critical; just describing the duration of warming isn't adequate, either, because the actual year is also important (all responses in, say, 2000, or 2010, share some similarities, as warming, CO₂, rainfall, etc. change over time ambiently). Despite the fact that this is an enormous data set, a lot of information one would want isn't available and I recognize that this can't be modeled precisely, but somehow a sharper picture should be possible to extract, and a much clearer graphical presentation than ext Fig 2 should be created and included right up front in the paper (this to me is the most important data). Perhaps a timeline with each data set shown as a line graph would be clearer? Or a multi-figure graph could be drawn with one graph for each site with responses for each year shown with a single time line, and the graphs for the sites stacked, plus one line averaged by date across all available data across sites for that date? I wanted to see how the responses changed over time (the shapes, by years) for each area (averaged across the plots in that area). How to handle that statistically is admittedly challenging, but at least showing it graphically would be valuable.
- We agree, and have streamlined and clarified the temporal analysis. In short, we specifically include **Fig. 3** (based on **Ext. Table 5a**) which presents our main results for the temporal analysis *across* experiments and include **Supp. Discussion #6** and **Supp. Fig. 5** to visualize *within*-experiment responses as well, as suggested by the reviewer.

- We want to stress, however, that we believe our main strength of our dataset and analysis, are to show *across*-experiment effects of duration/age class, which we do in **Figure 3**, in a similar way as we look at the effect size *across*-experiments in the main meta-analysis. That is, looking at the *within*-experiment effects of duration are less informative in our opinion. This is because of (i) uneven distribution of repeated measurements per experiment (see **Ext. Table 2**: Nr ER years, and the overview table below*), and (ii) lack of experiments stretching the whole duration range that we cover with the across-experiment data (0-25 years).

*Overview table:

A) Overview of the variability in time series length of ER years, i.e. Nr of ER measurement years within an experiment. Note that, although there are 2 long time series of 11 and 13 years of subsequent ER data (ALA_1, GRE_6 – Ext. Table 1-2), other experiments only provided data from between 1-4 years. Note as well that repeated measurements did not always include successive years (Ext. Table 2).	Nr of experiments	Nr of ER years
	21	1
	16	2
	9	3
	8	4
	1	11
B) Overview of the range in warming duration age class covered by the different datasets.	Nr of datasets	Warming duration age class (years)
	70	class 1 [0-5)
	28	class 2 [5-10)
	15	class 3 [10-15)
	23	class 4 (≥ 15)

- Further, we believe that looking at across-experiment variation to analyze temporal patterns is appropriate here, and is similar to what we do with our other meta-regression models (**Ext. Table 5b-c**). This is, to our knowledge, the strength of a meta-analysis and -regression approach: to overcome variation across different studies (in our case experiments). We also wish to stress that we do not evaluate absolute ER values in our manuscript, but respiration *responses to warming (effect sizes)*, which we believe is less problematic to evaluate across different experiments than it would be for absolute ER values. The latter would, we believe, indeed be much more influenced by specific measurement times or climatic conditions than the response to warming would be.
- We do not fully agree with the following statements on supposed missing information (*‘a lot of information one would want isn’t available’*), but believe this was related to lack of clarity in the previous manuscript. We attempted to clarify these things as follows:
 - ‘If soil respiration is not an immediate response (it isn’t), but rather the response is cumulative, somehow describing the shape(s) of that response over time is critical; just describing the duration of warming isn’t adequate, either, because the actual year is also important’.
 - Indeed, describing the shape(s) of the ER response over time is highly relevant, and this is exactly what we aimed to do with our temporal analysis (**Ext. Table 5a, Fig. 3**).

- We agree that the actual calendar year can also be important, as the climate is warming over the duration of these experiments, and climatic differences will exist among the calendar years. However, we believe the actual calendar year will not have influenced the majority of our results here, because of our focus on ER response to warming, and not absolute ER over time. Nevertheless, the temporal patterns examined in the manuscript might have been partly obscured by the underlying calendar year differences, as suggested by the reviewer. Because the experiments used in the analysis cover many different starting times and measurement years, however, we believe that the calendar year will only tend to obscure rather than actually contribute to the impacts of warming duration that we have assessed.
 - ‘Perhaps a timeline with each data set shown as a line graph would be clearer? Or a multi-figure graph could be drawn with one graph for each site with responses for each year shown with a single time line, and the graphs for the sites stacked, plus one line averaged by date across all available data across sites for that date? I wanted to see how the responses changed over time (the shapes, by years) for each area (averaged across the plots in that area).’
 - We included something like this in the previous **Supp. Fig. 5**, where we showed linear regression results for individual experiments per age class, combined with a mean regression line per age class. However, it was clear from the reviewer’s comment that we needed to clarify our temporal analysis throughout the manuscript, and hope that the combination of **Fig. 3**, **Supp. Fig. 5** and **Supp. Discussion #6** clarifies the temporal results. Specifically, we looked at both *across*- and *within*-experiment temporal patterns now but believe the across-experiment trends are most informative and therefore report these in our main text.
 - Due to the uneven distribution of ER measurements across experimental age (see overview table above), combining the different datasets proved the best way to test the effect of warming duration on ER response *across* experiments (i.e. space-for-time substitution). Here lies, in our opinion, the greater strength of our datasets to analyze temporal patterns, and we highlight this, as well as the distinction between the two types of temporal responses (across vs. within experiments) now in **Supp. Discussion #6**.
6. It wasn’t completely clear to me why Hedges’ *d* was used instead of response ratio (as the results are reported as a ratio or percent), or how the “ratio of means” differs from a response ratio, or why for some estimates the “ratio of means” was used and for others Hedges’ *g* was used (response ratios and Hedges’ *g* are both perfectly acceptable, just different).
- We focus on Hedges Standardized Mean Differences (SMD) as primary effect size for the meta-analysis and meta-regression models (**Ext. Table 4-5**) and interpretation of results, because Hedges SMD takes into account the precision of the measurements by including the pooled standard deviation in the calculation. The other effect size, log-transformed Ratio of Means (ROM), does not take into account variability in the measurements, but only the means, which is less preferred because of the variability in sampling design and methodology across the datasets (**Supp. Discussion #2**, **Supp. Fig. 2**). We included ROM as a secondary effect size for the main meta-analysis (**Ext. Table 4**) and upscaling, because it allows calculating a percentage change value. Percentage change allowed a more intuitive interpretation of the results, and was

essential to do a spatial upscaling (this could not be done with Hedges SMD). See L. 49-64 of our revised Methods for explanations on these different effect sizes and why and when each of them was used throughout the analyses.

- Importantly, in evaluating the warming-induced changes in microclimate (e.g. soil temperature), soil conditions, and microbial community composition, we also assessed a third effect size, i.e. raw mean difference or RMD, besides Hedges SMD. We do this to get a better feel of how big or small the environmental and vegetation/microbial community changes were caused by the warming treatment (**Ext. Table 3**, Methods L. 139). Since this effect size does not take into account the measurement precision, we always focus on the Hedges SMD instead of RMD in the main text.
7. Using geographically estimated values for some parameters seems appropriate, but where there were other missing values, multiple imputation would provide a more accurate analysis than would omitting the sites that have missing data (it wasn't clear how missing data were handled). Multiple imputation is a relatively new approach in meta-analysis, but appears to be a substantial improvement with respect to missingness. The approaches taken to dealing with the mass of data were courageous, but the one-at-a-time approach used in places is an older one that is lacking in important ways.
- It was unclear for us whether the reviewer suggested to do imputation of the ER response or of the environmental drivers here. We decided not to apply any statistical imputation for either because of the following reasons.
 - First, to our knowledge, imputation techniques are often necessary in meta-analysis in case of missing standard deviation or sample size to compute the sampling variance of the effect size. The lack of standard deviation or sample size is commonly due to primary studies not reporting such information. Yet, in our case, the dataset was compiled from the raw data, i.e. individual ER measurements, which implies that all necessary statistics to calculate our effect sizes and sampling variance calculation were available.
 - Second, imputation of predictors (environmental driver data) used in meta-regression was not performed either because of too large gaps in (some of) the predictor data, which would, in our opinion, not yield realistic results for imputed predictors. See **Supp. Table 2** for a sample size overview for each predictor. We chose to apply a more strict and parsimonious criterion for the predictors by focusing exclusively on what was measured in the field by the data contributors, or what could be collected from databases.
 - Importantly, there was some form of 'gap filling' in our data collection already, since data contributors could submit predictor data (environmental drivers characterizing soil conditions, vegetation, and microbial community) from a certain year to link with (sometimes multiple) ER datasets from another year. Please see explanations on that, and a detailed sensitivity analysis we performed to check that this did not influence the results in **Supp. Discussion #4** and **Supp. Fig 4**.
 - For our reply to the 'one-by-one approach' for meta-regression results, please see our response to your next comment #8.

Indirect warming effect drivers:

C Vegetation community

D Microbial community

E Context-dependency drivers

9. Heterogeneity is included in places, but I^2 and Q should be reported more consistently and the method used for calculating τ^2 for the random effects models should be reported; clearly these data are highly heterogeneous. In fact, that heterogeneity is an important part of the story being told here, because it is biologically reasonable and interpretable (at least in part). Showing how accounting for the biologically meaningful differences in response reduces that statistical heterogeneity would be very valuable.
- We agree and reported the strength and the quality of the models in more detail (Ext. Table 4-5). Previously, we only reported regression estimates, Q_m and our calculated R^2 values for meta-regression models. Since we used multi-level meta-analysis and -regression models, including both a random component structure as well as an autocorrelation structure, the estimated heterogeneity in our models is partitioned across these different components³⁻⁸. Therefore, we expanded our tables with model results to also include (i) the Q -values for heterogeneity, (ii) the variance partitioning of the three-level component across σ^2_1 (for between-experiments heterogeneity) and σ^2_2 (for between observation-within-experiment

heterogeneity); and (iii) τ^2 and ρ values reflecting the autoregressive component in our meta-models: **Ext. Table 4-5**. We also updated the method description on how we account for heterogeneity (L. 174).

Referee #3 Ivan Janssens

1. The submitted manuscript analyzes an impressive dataset on passive warming experiments by OTCs, focusing on the environmental drivers explaining the variation of the ecosystem respiration (ER) response to the warming. The rationale for the study is that more insight is needed in the response of respiration to warming to obtain better estimates of the land carbon-climate feedback. The key finding is that across all experiments and independent of warming duration, there is a mean increase of ER of 30% for a mean warming of 1.4 °C. Experiments with lower nitrogen concentrations in the mineral soil exhibited a greater stimulation of ER by the placement of OTCs. An upscaling approach using global maps of the key drivers then revealed Western and Eastern Siberia and the Canadian Archipelago as hotspots of warming sensitivity. While this analysis of the response of ER to the OTC placement across a large number of experiments is of course interesting and novel, there are multiple issues with the current version, through which I cannot assess whether or not this study is truly relevant. I will list these issues first and thereafter provide a list of questions/comments that arose while reading the text.

- We appreciate the reviewer's positive remarks and addressed the outspoken concerns with care below. We do want to stress that our aim was written as "*we urgently need a better quantification and greater mechanistic understanding of how warming influences ER across the tundra biome*" (L. 148). This differs from the aim to obtain *better estimates of the carbon-climate feedback* the reviewer mentioned. This is an important distinction, related as well to the reviewer's comment #3 on lack of GPP data since it was not available. With our dataset on environmental drivers of ER in response to warming, the land carbon-climate feedback is hard to access, but we believe we *can* assess one of the key components of that feedback, i.e. respiration by plants and microbes. We believe this is novel because no synthesis study of respiration changes with warming, and based on field data, has been done for the tundra biome at this scale before. We do describe potentially strong *implications for* (but do not claim estimations of) the land-carbon climate feedback. However, including NPP and GPP is outside the scope of our work here.

Major comments

2. This manuscript deals with warming responses, using the increase in air temperature induced by OTC placement. However, air temperature would only affect aboveground respiration (roughly 20-40% of ER) and not the part that is key to the land carbon-climate feedback. How did soil temperatures change? I was very surprised that Figure 1b, nor any of the Tables give this soil warming data. The majority of experiment placed the OTCs only in the growing season. Because of the passive warming and the vegetation cover in summer, I doubt that substantial soil warming was realized, at least not in the first month. By removing the chambers every August/September, any difference in soil temperature would be undone every Autumn to Spring period, in contrast to the experiments where the OTCs remained in place. Surely this must have affected the soil warming intensity? By averaging out all RE measurements over the growing season, independent of the amount of soil warming, and not taking this into account in the analysis, substantial error may have been introduced. The manuscript also fails to report on the differences in snow cover and wind speed, that have huge impacts on vegetation and thus on ER as well.
- We did find an overall soil warming which is described on L. 199: "*Warming with OTCs led to a mean 1.4 °C [95% CI 0.9-2.0 °C] (p<0.001; N=77) increase in growing season air*

temperatures, a mean 0.4 °C [CI 0.2-0.7 °C] ($p < 0.001$; $N = 118$) increase in growing season soil temperature, and a mean 1.6% [CI 0.8-2.4%] ($p < 0.001$; $N = 111$) decrease in growing season soil moisture compared to ambient conditions (based on meta-analysis models, *Ext. Table 3*).” We include the effects on soil temperature and moisture in the Visual Abstract now as well:

- Regarding: ‘we did not take into account the amount of soil warming in our analysis, and therefore may have introduced substantial error’. → As part of our main meta-regression models investigating indirect warming effects, we specifically assess the relationships between the ER response to warming (Hedges SMD) and the microclimatic warming effects (Hedges SMD of air temperature, soil temperature, and soil moisture, **Ext. Table 5b**) which is equivalent to the amount of warming the reviewer refers to. Yet, we did not find any significant relationships between the ER effect sizes and the microclimatic effect sizes as described in (L. 250).
 - Finally, regarding: A) ‘I doubt that substantial soil warming was realized, at least not in the first month’. → Soil warming *was* realized throughout the growing season. We found a mean estimated Hedges SMD of 0.18 [CI 0.08-0.29]***, corresponding to a mean estimated increase of soil temperatures with 0.44°C [CI 0.24-0.65 °C]*** (**Ext. Table 3**). Further, we now checked seasonal variation in the effect size of soil warming, by running a meta-regression model testing the effect of the measurement ‘Day of the year’ on daily ER Hedges SMD. This showed that soil warming became stronger over the season (green box; table below), but was already there at the beginning of the growing season (day 152 – 1st of June). However, we do not believe this has confounded our results because of the following reasons. Second, when we run the meta-regression model that tests for an effect of soil warming (Hedges SMD soil temperature) on ER Hedges SMD separately for the early half vs. late half of the growing season, we find positive ER responses in both parts of the growing season (output with red boxes below).

■ *Results of meta-regression model testing the effect of the measurement ‘Day’ on daily ER Hedges SMD per experiment, and statistics and visualization of the result in a scatter plot:*

	estimate <chr>	se <chr>	zval <chr>	pval <chr>	ci.lb <chr>	ci.ub <chr>	<S3: noquote>
intrcpt	0.0819	0.2030	0.4033	0.6867	-0.3160	0.4798	
Day	0.0018	0.0007	2.4948	0.0126	0.0004	0.0032	*

- *Results of meta-regression models testing the effect of soil warming (Hedges SMD soil temperature) on ER Hedges MSD separately for the early half vs. late half of the growing season:*

Early season [Julian Day 152-197]

	estimate <chr>	se <chr>	zval <chr>	pval <chr>	ci.lb <chr>	ci.ub <chr>	<S3: noquote> ***
intrcpt	0.6852	0.0725	9.4500	<.0001	0.5431	0.8273	
SMD_ST	-0.0176	0.0257	-0.6861	0.4927	-0.0679	0.0327	

Late season [Julian Day 198-244]

	estimate <chr>	se <chr>	zval <chr>	pval <chr>	ci.lb <chr>	ci.ub <chr>	<S3: noquote> ***
intrcpt	0.6440	0.0675	9.5463	<.0001	0.5117	0.7762	
SMD_ST	0.0822	0.0756	1.0882	0.2765	-0.0659	0.2304	

- B) In response to ‘By removing the chambers every August/September, any difference in soil temperature would be undone every Autumn to Spring period, in contrast to the experiments where the OTCs remained in place. Surely this must have affected the soil warming intensity?’ → Although our analyses focused on the growing season (June-August), when all OTC chambers would have been in place, removing the OTCs in winter could still affect the intensity of the microclimate effects during the growing season due to lagged effects. To assess this, we checked with an additional meta-regression model whether microclimatic effect sizes, i.e. Hedges SMD of air temperature, soil temperature, and soil moisture, were influenced by OTC removal or not. See the distribution of datasets where OTCs were removed or not, as well as the model output below.

- *Air temperature*: no significant effect of OTC removal on our Hedges SMD of air temperature in the growing season.

```
##           estimate      se    zval    pval    ci.lb    ci.ub
## intrcpt          0.2050  0.1214  1.6884  0.0913  -0.0330  0.4431 .
## OTC_RemovalYes  0.1397  0.1435  0.9731  0.3305  -0.1417  0.4210
##
## ---
## Signif. codes:  0 '***' 0.001 '**' 0.01 '*' 0.05 '.' 0.1 ' ' 1
```

- *Soil temperature*: no significant effect of OTC removal on our Hedges SMD of soil temperature in the growing season.

```
##           estimate      se    zval    pval    ci.lb    ci.ub
## intrcpt          0.1737  0.0825  2.1064  0.0352  0.0121  0.3353 *
## OTC_RemovalYes  0.0444  0.1117  0.3975  0.6910  -0.1745  0.2634
##
```

- *Soil moisture*: significant effect of OTC removal on our Hedges SMD of soil moisture in the growing season, with larger SMD values (smaller decreases) when OTCs were removed in winter.

- In sum, there was no effect of removing the OTCs in winter on the increase in air or soil temperature due to warming. However, removal of OTCs in winter did affect the decrease in soil moisture due to warming, with removal of OTCs leading to a smaller decrease in soil moisture with warming in the growing season. Yet, we do not think this confounded our results of the lack of relationship between the ER response and the soil moisture response to warming (**Ext. Table 5b**). Namely, we do not find an interaction between OTC removal and the SMD of soil moisture on the SMD of ER (see below), indicating that relationships between SMD ER and soil moisture are similar for datasets where OTCs were removed, as for datasets where they were not removed.

	estimate	se	zval	pval	ci.lb	ci.ub	<S3: noquote>
intcpt	0.6011	0.1159	5.1852	<.0001	0.3739	0.8284	***
SMD_SM	-0.0505	0.0714	-0.7069	0.4796	-0.1904	0.0894	
OTC_RemovalYes	0.0142	0.1646	0.0864	0.9312	-0.3083	0.3367	
SMD_SM.OTC_RemovalYes	0.0567	0.1147	0.4938	0.6214	-0.1682	0.2815	

- C) ‘The manuscript also fails to report on the differences in snow cover and wind speed, that have huge impacts on vegetation and thus on ER as well’.
 - Although we agree that snow cover and wind speed confound OTC effects, we believe that this is minimal, because (i) we do not find a difference in the ER response rates with warming between experiments that kept their OTCs in the winter versus not; (ii) although these aspects can indeed affect vegetation structure/height, we do not find that changes or differences in vegetation structure (e.g. SMD shrub cover; SMD mean height) influenced the ER response rates (**Ext. Table 5b**, L. 250); and (iii) our analysis focuses on the growing season, a time where snow cover and high wind speeds are minimal, hence they could have influenced the ER response rates only indirectly through vegetation changes, which we assessed (see (ii)).
3. This brings me to a major issue. The manuscript only reports ER fluxes; I guess also GPP data will be available but being spared for a follow-up paper. The problem is that GPP is a more important determinant for ER than air temperature and even soil temperature: Plants can only respire more if

GPP increases to sustain the higher respiration; microbes that respire more deplete their food source, which results in a shrinking microbial population and eventually reduces microbial respiration to the level of soil carbon inputs under the warmed conditions. Thus: long-term sustained increases in ER must be accompanied by increased GPP and NPP. In this study, there are clear indications that the 30% increase in ER is more driven by a warming-induced productivity increase than directly by warming. For example, there is a clear tendency towards reduced bulk density in both organic and mineral layers, for which the most likely reason is increased soil organic matter inputs and associated microbial- and faunal activity. Bulk density may also have decreased because the OTCs provide warmer and wind-free shelters for soil animals. Total C stocks are not given, but C concentration did on average not decrease, which one would expect if stimulation of ER would be driven by the warming and not by increased GPP. Given that I believe that increased productivity is driving increased ER, plus that soil carbon stocks were not affected by the OTCs, I cannot see the relevance of the 30% increase in ER for the land carbon-climate feedback, which is how the authors currently sell the story.

- We appreciate this comment that led us to thoroughly rephrase our objectives and to collect additional data on ecosystem respiration partitioning to further strengthen our findings and storyline. Below, we reply in detail to the different subsections and statements from the reviewer's comments:

A) 'The manuscript only reports ER fluxes; I guess also GPP data will be available but being spared for a follow-up paper.'

- Although we strongly agree that adding GPP data to our database would allow to explore climate feedbacks and C budgets in detail, it would, at the same time, take our current story away from what it is truly about: mechanistic drivers of changes in ecosystem respiration due to warming. We believe that investigating changes in, and drivers of, increased ER across the tundra is highly relevant and novel to assess on its own because of ER being a key component in the C balance. Adding GPP to the story would not allow us to explore the drivers of RE in this much depth since the two processes (GPP vs. ER) are driven by fundamentally different drivers. Understanding ER is therefore an important aim, and understanding GPP and its drivers could indeed be the next step.
- If we would collect additional **GPP** data, we should also collect **NEE** data to properly assess the C budget/net effects, which stretches beyond our objectives, and would require extensive additional sampling. Therefore, instead of extending our story to focus on net C budgets (NEE) driven by changes in GPP vs. ER, by collecting additional flux responses, we decided to strengthen our story on the ER partitioning (L. 214, 258, 402), since the reviewer suggested that increased ER would mainly be driven by increased R_a and not by increased R_h as well. This showed that both R_a and R_h increased by warming.

B) 'Plants can only respire more if GPP increases to sustain the higher respiration; microbes that respire more deplete their food source, which results in a shrinking microbial population and eventually reduces microbial respiration to the level of soil carbon inputs under the warmed conditions. Thus: long-term sustained increases in ER must be accompanied by increased GPP and NPP.'

- Although studies have shown indeed that climate warming may lead to increased (heterotrophic) respiration first, quickly followed by no significant increase in respiration after 5-10 years, possibly due to a combination of **microbial acclimation and substrate depletion**^{11,12}, later studies have actually shown that these assumptions may not be correct for

all types of ecosystems. In Dorrepaal et al. (2009)¹³, for instance, ER response in a tundra site continued for at least 8 years, with the main part of this increase coming from heterotrophic decomposition and from **deeper soil layers**. Similarly, long-term consistent increases in respiration with warming and permafrost thaw have been found by Schuur et al. (2009)¹⁴. At first, it was speculated that these more persistent increases in ER in tundra sites might be a response typical of their organic rich, poorly decomposed soils: the decomposition per g soil C might be too low (still being quite cold) and the total C pool too large to deplete the substrate fast enough. However, even the classic hardwood forest story which proposed the accelerated mineralization and increased respiration response to warming to be short-lived when evaluating the first decade of warming¹¹, turned out to be more complex than a simple carbon depletion story when they continued the measurements until 25 years¹⁵. Although of course, in the longer term, many more ecosystem-mediated responses start to affect the respiration response. Interestingly, our results demonstrate a similar non-linear trend with cumulative warming as for the hardwood forest: first a dip in responsiveness between 5-10 years of warming (i.e. the ER remains increased with warming, but decreases in magnitude, see L. 225 and **Fig. 3b**), then a recovery in responsiveness, i.e. increase of the response magnitude, which ultimately leads to no significant change in the long run (**Ext. Table 4a**).

- In addition to microbes delving into SOC pools in deeper soil layers, several studies have also shown that warming can lead to increases in respiration from **older soil carbon**, instead of just plant-related respiration and young soil C^{10,16}. These studies showed that soil microbes do not get starved by depletion of soil C very quickly, and that ecosystem-mediated changes in respiration pathways in the longer term may cause breakdown of even older soil C.

C) ‘In this study, there are clear indications that the 30% increase in ER is more driven by a warming-induced productivity increase than directly by warming’.

- Prompted by the reviewer’s hypothesis that the increase in ER is more driven by a warming-induced productivity increase than directly by warming, we collected additional data, i.e. the warming response of the autotrophic vs. heterotrophic components of ER, or the plant-related vs. microbial respiration, for a subset of 9 datasets used in the manuscript. With these new data (see overview below and **Ext. Table 4b, Supp. Discussion #1**), we demonstrate clearly that the 30% increase in ER is likely not driven only by a warming-induced productivity increase leading to increased plant-related respiration, but equally so by a warming-induced increase in microbial respiration. We present these results in the revised manuscript on L. 214, in **Ext. Table 4b**, and explain and visualize them in **Supp. Discussion #1**.
 - Plant-related respiration Ra (N=9)
 - Mean Hedges SMD ↑ 0.44 [0.08, 0.80] ***
 - Mean percentage increase ↑ 56.9 [6.9, 130.2] % ***
 - Heterotrophic respiration Rh (N=9)
 - Mean Hedges SMD ↑ 0.92 [0.36, 1.48] ***
 - Mean percentage increase ↑ 54.5 [31.8, 81.2] % ***
 - ER of the partitioning subset (N=9)
 - Mean Hedges SMD ↑ 0.72 [0.42, 1.02] *** (N=9)
 - Mean percentage increase ↑ 39.2 [19.6, 62.1] %
 - ER (N=136) – full dataset: to demonstrate that ER (N=9) is representative of full dataset trend
 - Mean Hedges SMD ↑ 0.57 [0.44, 0.70] ***

- Mean percentage increase \uparrow 29.8 [22.3, 37.8] % ***

D) ‘For example, there is a clear tendency towards reduced bulk density in both organic and mineral layers, for which the most likely reason is increased soil organic matter inputs and associated microbial- and faunal activity. Bulk density may also have decreased because the OTCs provide warmer and wind-free shelters for soil animals. Total C stocks are not given, but C concentration did on average not decrease, which one would expect if stimulation of ER would be driven by the warming and not by increased GPP.’

- As part of our revision, we have also gone through the entire data used in the manuscript. With this, a few outliers were detected, and removed from the analysis after discussion with the data contributors. Due to this update, the effects of warming on local soil conditions were updated, see **Ext. Table 3**. Hence, the reduced bulk density is no longer significant, and no longer interpreted as a trend in the updated manuscript. Further, we only have total C stock data at experiment level and not in response to the warming treatment (**Ext. Table 2**), so we could not test whether this was affected by the warming treatment. The effects of warming on total C stock are also heavily debated, and could be a paper on its own^{17,18}. Finally, many of our experiments are located on sites with organic soils with very high C content, so detecting the impact of warming on soil C stocks would be difficult here, due to a lack of sufficient spatial variation in soil C content.

E) ‘Given that I believe that increased productivity is driving increased ER, plus that soil carbon stocks were not affected by the OTCs, I cannot see the relevance of the 30% increase in ER for the land carbon-climate feedback, which is how the authors currently sell the story.’

- We show with our additional analyses (see our response to part C) that the reviewer’s hypothesis that ‘increased productivity is (solely) driving increased ER’ is not supported by our data, where increased ER arose from both increased plant *and* microbial respiration. Further, we do not know whether soil C stocks are affected by the OTCs, given that we could not calculate treatment-level stocks (as explained in more detail in comment #10). Based on this evidence, and the reasons explained above, we believe our results are highly valuable and implicational for the land carbon-climate feedback, which we include in more detail now on L. 404. Namely, “... Namely, while **increased autotrophic respiration might be met by an equally strong increase in productivity, a combination of increased autotrophic and heterotrophic respiration with warming is likely to free up more carbon than can be compensated for by increased plant productivity alone**. While our analyses pertain to the growing season ER fluxes only, changes in non-growing season fluxes are likely to aggravate future C losses. Our findings therefore **imply a potentially strong, persistent positive climate feedback coming from the tundra, even under modest warming during the coming decades, which may accelerate, rather than mitigate the impact of global warming**”.

4. The global scaling depends on the model found across these warming experiments, but does not extrapolate the uncertainty of the model. Then, this model is multiplied with an ISRIC dataset providing the spatial data for the model variables, which has gigantic uncertainties for the arctic region, which could have been quantified given the available site data, but are also fully ignored. In addition, one of the drivers is not even available in the ISRIC dataset but is approximated. Also this proxy was not evaluated using the available empirical data and its uncertainty not extrapolated in the final product. Best remove this entire part of the manuscript or support it with a thorough uncertainty analysis.

- We have now fully revised the upscaling workflow, addressed the uncertainty not only from the meta-regression model but also from the ISRIC dataset and from using a proxy for C:N-ratio. The revised method is explained in detail in the Methods (*Spatial upscaling*) and summarised in a flow diagram in **Supp. Methods #3**. The main changes to the method are:
 - We now use the native resolution (250 m) soil data, for which ISRIC provides uncertainty estimates (which are not yet available for other resolution (i.e. 1 km and 5 km products). This allowed us to estimate the uncertainty originating from the ISRIC data. Due to the large study area (10 million km²), we still needed to aggregate the soil data to 1 km resolution as even with this resolution we needed a supercomputer to run the uncertainty analysis.
 - We used the reported C:N-ratio and reported TN and TC from our database (**Supp. Fig 8**) to estimate the uncertainty originating of the proxy using TN and SOC to derive the C:N-ratio, and incorporated this uncertainty in our Monte Carlo uncertainty analysis.
 - In the Monte Carlo uncertainty analysis, we created 100 pairs of TN and C:N-ratio values for each 1 km² grid cell using a truncated multivariate random normal distribution generator, using the mean and standard deviation of these variables.
 - We then used the resulting 100 pairs of TN and C:N-ratios as input for our ROM model for each 1 km² grid cell, resulting in 100 estimations for mean and standard deviation. These were then combined to get the mean percentage change in ER induced by 1.4°C warming and its combined standard deviation.

Detailed comments

5. The duration effect should be studied within ecosystems, not across all experiments, to avoid other drivers confounding the analysis.
 - Please see our response to comment #5 from Reviewer 2 for this, as these comments were similar. In short: we *have* investigated both *within*-and *across*-experiment effects of duration, the former with linear regression models and the latter with meta-regression models which account for methodological differences across the experiments (**Fig. 3**, **Supp. Fig. 5** respectively). However, we think the across-experiments results are most informative, looking at the uneven length of the repeated measurements of ER within experiments we have, and the limited range of duration that most experiments cover, while covering a range of 0-25 years across all experiments. See **Supp. Discussion #6** as well for details.
6. Soil carbon and changes therein are crucial, but little to no Methods were given and changes not reported. Was it sampled uniformly?
 - Please see our response to your comment #2 for this. We *did* analyze changes in soil C concentration (SOC and SOM mg/g soil) due to the warming treatment, and report these in **Ext. Table 3**. We looked at total concentrations (TC, TN, C:N) in the mineral and organic layer separately, where the mineral vs. organic layer was determined by data contributors based on their site and ecosystem knowledge. These data was not sampled across a standardized measurement protocol, but this is very common in meta-analyses. We only included measurements using CNS element analyzers (the majority of measurements) and expressed all values as mg/g soil. We describe this in the Methods (L. 103, see also **Supp. Methods #1**). For neither of the soil layers did we detect a significant trend in C concentrations with warming in the study sites (based on the mean Hedges SMD results, **Ext. Table 3**).

7. Fig 1: show the soil T response too
 - See our response to your comment #2. We now added the soil temperature (and air temperature and soil moisture) effect resulting from experimental warming to the Visual Abstract and in the main text (L. 200).
8. Fig 2: N and C concentrations are given but soils have become lighter by the OTC treatment. Did you account for this also in the sampling? Sampling at the same depth is not appropriate if they have loosened.
 - We looked carefully at the data and we detected a mistake in the way BD was provided to us and corrected it. This changed the relation to not significant (Hedges SMD bulk density of mineral layer: -0.24 [CI -0.70, 0.22]).
 - Standardized sampling in such multi-participants studies is challenging. We standardized by separating samples from mineral and organic layers. Most contributors are part of the ITEX network which makes use of common protocols.
9. Fig 2: in the graphs you give multiple ER data for single driver observations. Please only show one ER per driver observation.
 - We think the reviewer refers here to the fact that in several experiments, replicated environmental driver data from the same measurement year is used, relating this data to distinct, unique ER data points. We clarified this methodological aspect of our driver data collection and how we tested for the possible influence on our results with a sensitivity analysis in **Supp. Discussion #4** and **Supp. Fig. 5**. This showed that, even when we allowed less repetition of environmental driver data, the main results did not change. Please see our response to comment #8 of Reviewer #1 as well.
10. Line 260: this hypothesis could be tested by comparing the change in SOC stock between N-poor and N-rich sites...
 - The reviewer refers to the hypothesis that ‘warming might stimulate plant root production and root exudation rates to access nutrients and prime microbial communities to mineralize nutrients (i.e. rhizosphere priming), thus enhancing root and microbial respiration and thus ER response rates.’ We believe the reviewer means we could test whether the change in SOC stock (e.g. Hedges SMD effect size) was significantly affected by the TN content. However, we do not have SOC stock values at treatment level (for both control and warmed plots), only at experiment (site) level, so unfortunately cannot test this.
 - Please note that, particularly because one should be very cautious about the specific sampling method for accurate SOC stock quantification, we did not find it appropriate to calculate SOC stock values at the treatment level based on the datasets where we have both bulk density and total C values available. That is, we cannot ascertain the similarity in Methods for the two drivers across experiments, e.g. sampled where in the plots, at which depth, for us to properly assess SOC stock as a driver¹⁹⁻²³. Through our meta-regression models, we do account for this sampling variation, but this is based on the original data (e.g. TC, BD) and not derived data (e.g. SOC stock based on TC and BD). Further, the number of datasets that have both bulk density and TC available, are limited and the majority of those come from the same site, i.e. 6 out of the 9 datasets for mineral layer data from Latnjajaure (experiments SWE_1 to 5), and 9

out of the 15 datasets for organic layer. Even if we did run a meta-regression based on derived SOC stock values, the information to extract from that would not be very informative for our overall dataset.

11. Figure 3: A) was there an interaction between vegetation type and soil warming? B) Did soil warming differ between growing season-only and full year applied infrastructures?

- We checked these questions through additional models and plots and report the results point per point below:

A) Was there an interaction between vegetation type and soil warming?

- We interpret this question as ‘whether the amount of soil warming differed significantly between vegetation classes’. If so, we believe the reviewer thinks that this might have confounded our result that vegetation class did not affect ER Hedges SMD (**Ext. Table 5c**). We tested this by running a meta-regression model evaluating effect of vegetation class on the Hedges SMD of soil temperature. There is no significant effect of vegetation class on these Hedges SMD values for soil temperature ($Q_m=6.24$, $p\text{-val} = 0.18$, see plot below). Because it does not appear like a potential interaction could have confounded our results, we did not add this to our manuscript or story.

B) Did soil warming differ between growing season-only and full year applied infrastructures?

- See our response to your comment #2 for this. To test whether soil warming differed between growing-season only and full-year applied infrastructures, we ran a similar meta-regression model as for A), but testing the effect of OTC removal as predictor. We do not find an effect of removal of OTCs (i.e. growing-season only application) vs. no removal of OTCs (full-year application) on the amount of soil warming (Hedges SMD soil temperature; $Q_m=0.16$, $p\text{-val} = 0.69$, see plot below). Since we find no apparent confounding, we did not add this to our manuscript.

12. Line 294:

A) Surely, the depth from the surface where the mineral soil starts must have influenced the importance of mineral soil N and C/N. Was this analyzed?

B) Also, the role of N turnover is hypothesized, probably correctly, but why wasn't a meta-analysis conducted on the N turnover directly? I assume these data were available at many sites, given that N turnover is well known to be a key determinant of the ecosystem response to warming.

- To your first question A), as earlier mentioned, soil environmental driver data was obtained from past measurements, and represented samples from the mineral or organic soil layer, without specifications of which depth the N / C:N samples were taken at. Slight differences in Methods are common in meta-analysis that gain power by numbers despite sometimes lacking sampling specific information. Moreover, as most of the sites belonged to the ITEX network sampling Methods were most likely relatively standardized compared to other meta-analyses.
- To your second question B): because it would be sensible to collect N mineralization or turnover data according to a standardized protocol²⁴, and at this stage the majority of experiments likely do not have this data yet, we did not collect/analyze this here.

13. Fig 5: the text states that the ER response extends to >100%, then best expand the scale to 100%; although as stated above I have little confidence that this analysis bears any meaning

- The scale has been updated accordingly in **Figure 6**. See our response to your comment #4 for details on how we have revised our upscaling analysis, including adding an uncertainty analysis to it.

14. Line 327: this is not true. It depends on the persistency of an imbalance between C inputs and respiratory losses, not on the persistency of stimulated ER.

- Agreed, and we believe we now improved the discussion on the C balance. We have revised the sentence on L. 375. It now reads: *“The magnitude of the long-term land carbon-climate feedback critically depends on the balance between C inputs through changes in photosynthetic plant uptake and C losses through ecosystem respiration. In this regard, the magnitude and persistency of C losses through ER is a crucial component in evaluating this feedback and how it may change with future warming.”*

15. Line 334: explaining why some of the long-term running sites show increased ER while others show declining ER to me would be a very informative analysis

- Importantly, we believe the reviewer means to say ‘increased ER *responses* to warming while others show declining *responses*’. This is an important difference, as we are not analyzing temporal patterns in absolute values of ER here, but trends in how *warming* affects ER over time (quantified as ER Hedges SMD or effect sizes). We agree that the occurrence and drivers of potentially different temporal patterns between experiments would be a very interesting question to address in a future meta-analysis. However, currently, the temporal resolution, replication and total duration of the available ER and driver data *within* experiments, as well as total replication and standardization of ER measurements *across* experiments are unfortunately not sufficient for this (yet).
- Please see our **Supp. Discussion #6** and **Supp. Fig. 5** though for detailed output of the within-experiment temporal patterns, and a discussion on why we decided that analysing baseline changes of ER were out of the scope for this manuscript in comment #4 of Reviewer #2.

16. Line 345: providing modelers with a more robust understanding would require you to show the responses of GPP and NPP as well.

- Please see our response to your comment #3 for this. We further believe that our partitioning results are equally interesting to modelers as NPP and GPP, since they are the underlying processes and thus give us more mechanistic insight.

17. SI Line 27: the differences in OTC height is not slight; the effect on wind penetrating to the soil surface must be huge.

- The reviewer probably refers here to the previous Methods Line 27 instead of Supp. Info Lines 27. In the Methods, we stated previously that ‘experiments differed slightly in OTC size (0.3-0.7m height),...’. This information, and how we assessed whether it influenced the ER response rates was also included in Supplementary Discussion section 1.

- We agree that a range of 40 cm height difference should not be considered ‘slight’ in these experiments, so we removed this word on L. 27 in the Methods. However, almost all studies had a height of 0.3-0.5 which is a much smaller range (see figure above).
- Since **OTC height** differences showed no significant effect on the ER Hedges SMD (**Supp. Fig. 3: $Q_m = 3.1$, $p = 0.08$**), we disregarded it as an important driver. Further, when running an explorative backward selection model on Methodological drivers, OTC height was not selected

in the final model (see **Supp. Discussion #2**). Neither did OTC height significantly affect ER Hedges SMD in a single-factor model.

- Wind circulation effects may not only depend on height. To address the reviewers concern, we also tested whether **OTC plot size** might have influenced our ER Hedges SMD, since the plot size reflects differences in diameters, and this may also affect wind circulation in addition to OTC height. Plot size ranged from 0.25 to 4.3 m² (see histogram below). Plot size did not affect our ER Hedges SMD, therefore we believe this reaffirms our arguments that chamber dimensions did not affect our main findings. See **Supp. Discussion #2** for a detailed explanation on how we checked for, and report on, possible methodological bias in our results.

	estimate <chr>	se <chr>	zval <chr>	pval <chr>	ci.lb <chr>	ci.ub <chr>	<S3: noquote>
intrcpt	0.6510	0.1086	5.9955	<.0001	0.4382	0.8639	****
Plot_Size	-0.0529	0.0557	-0.9498	0.3422	-0.1620	0.0562	

18. Line 41: this description does not suffice. Imagine that the OTCs were placed on day X and measurements started on day X+1, then likely vegetation profited from the reduced cooling at night, but the soil would not have changed. Warming the soil, especially the underlying mineral soil, with passive warming as applied here, must take many weeks/months. Removing the OTCs every autumn undoes the warming, such that even after 10 years this artefact remains. Details should be given on this.

- We are unsure whether the reviewer's comment really relates to Line 41 in the initial main text since this was about how we calculated ER effect sizes. However, we can reply based on the content of the comment, namely:

A) 'Warming the soil, especially the underlying mineral soil, with passive warming as applied here, must take many weeks/months.'

- See our response to your comment #2 for this. Passive warming in the experiments did lead to significant soil warming even at the start of the growing season (Julian day 152), so we believe that a potential delay in, or lack of, soil warming was not an issue in our analyses. Further, we checked whether there was seasonal variation in the soil warming response and how this might have influenced the results. Please see our response to your comment #2 for that.

B) 'Removing the OTCs every autumn undoes the warming, such that even after 10 years this artefact remains. Details should be given on this.'

- See our response to your comment #2 and 11 for this. We checked whether OTC removal in winter influenced 1) the effect size of soil warming, which it did not. Neither did it influence

2) our ER response, so we believe we do not need to account for removal of OTCs in our analysis based on this.

19. Unit for NPP is wrong

- We added year⁻¹ to correct the unit for NPP: kg C m⁻² year⁻¹ in the Methods and **Supp. Methods #1**.

20. Define CAVM

- We defined CAVM in the Methods and **Supp. Methods #1**.

21. Define how soil environmental variables were measured (what depth?)

- Soil temperature is measured for every *ER measurement* level with climate loggers, and hence were measured at a relevant depth for the ER measurement. i.e. every ER measurement had coinciding soil temperature and soil moisture values (if these data were available for the experiment). The ER measurements, and hence temperature measurements were done at different depths. We provide the overview of the depths, and how many datasets and experiments measured at different depths, below and in **Supp. Methods #1** now. Since the **majority of the measurements were done at 5 cm**, we selected the moisture or temperature data from this depth in case multiple measurements were available.

Soil moisture			Soil temperature		
Depth (cm)	Number of datasets	Number of experiments	Depth (cm)	Number of datasets	Number of experiments
2.5	5	5	2	7	6
3	11	5	2.5	1	1
3.5	9	5	3	2	1
3.75	3	1	5	78	39
5	40	17	7.5	8	4
6	13	1	10	22	5
7.5	11	5	total:	118	56
10	19	5			
total:	111	44			

- The other soil environmental drivers were measured at *plot* level, i.e. soil organic matter (SOM, %), total C concentration (TC, %), total N concentration (TN, %), C:N-ratio, bulk density (BD, g cm⁻³), and pH from the mineral or organic layer; as well as organic layer depth (OLdepth, cm). For these drivers, we requested data contributors to provide their data as either from mineral or organic layer, so there are no specified depths at which samples were taken. We trust that the data contributors know their site well enough to correctly define their organic and mineral layers best, and that they provide samples that are representative of that layer.
- Important to note, it is common to have different measurement methodologies across studies in meta-analyses. This is also why we perform our analysis with the meta-analysis and -regression approach, since this statistical technique takes sampling variation into account.

22. Methods should allow the reader to reproduce the analysis, but apart from explaining SMD, I have no idea what exactly was done. I assume you used the difference in seasonal mean ER from an

experiment? This gives the effect size, but how did you deal with the differences in warming throughout the season, across soil depths, and across years?

- We have revised the Methods to better explain what we have done. To respond to your specific questions: we indeed used the difference in growing season mean ER from control and warmed plots for every dataset to calculate ER response (SMD) for that dataset. A ‘dataset’ represented here a unique ER measurement year x experiment combination as we often had repeated measurements of ER from the same experiment. This was used as our response variable (primary effect size) for the multivariate meta-analysis and meta-regression models (‘rma.mv’ from metafor package) and is explained on L. 50 in the Methods.
- Further, we performed multivariate meta-regression models to analyze precisely what the reviewer hints at, e.g. to investigate which drivers influenced variation in the mean growing season ER. The meta-regression models, which are explained on L. 142 in the Methods and results shown in **Fig. 3-5** and **Ext. Table 5** are comparable with linear mixed effect models, in that they can account for random effect structure, as well as for temporal autocorrelation structures. But at the same time they increase precision of the estimated effects by weighting each dataset by the inverse of its estimated sampling variance. To answer your specific queries, see below: ‘How did you deal with...’

A) the differences in warming throughout the season?

- Please see our response to your comment #2 on seasonal variation.

B) the differences in warming across soil depths?

- In our response to your comment #21, we show that soil temperature data was measured at a range of soil depths, i.e. between 2.5 and 10 cm deep, but with the majority of experiments measuring soil temperatures at 5 cm deep (17 out of the 44 experiments; 40 out of the 111 datasets). These soil depths are relevant for plant growth and microbial activity, and hence for ER. Soil depth did not influence the Hedges SMD of soil warming (see results below). Detailed studies of how warming at different depths would affect ER is, given our results that deeper soil layers may be important for ER, an interesting avenue for further studies.

	estimate <chr>	se <chr>	zval <chr>	pval <chr>	ci.lb <chr>	ci.ub <chr>
intrcpt	0.1419	0.1515	0.9365	0.3490	-0.1551	0.4389
ST_Depth	0.0103	0.0259	0.3998	0.6893	-0.0404	0.0611

C) the difference in warming across years?

- As part of our meta-regression models (**Ext. Table 5b**: indirect warming effects), we tested whether the Hedges SMD of air temperature, soil temperature (i.e. reflecting the air and soil warming magnitude and direction per dataset) influenced the ER response. They did not, which implies for us that differences in achieved warming across years and experiments does not drive variation in ER response rates.

23. I did not fully understand the statistical approach, so some of the comments above may be unfair.

- We appreciate the honesty of the reviewer, and hope we have adequately addressed the concerns and unclarities to make our manuscript and the analyses we have done more comprehensible.

References

1. Jenkins, D. G. *et al.* A meta-analysis of isolation by distance: Relic or reference standard for landscape genetics? *Ecography (Cop.)*. **33**, 315–320 (2010).
2. Vaessen, T. *et al.* The association between self-reported stress and cardiovascular measures in daily life: A systematic review. *PLoS One* **16**, 1–28 (2021).
3. Signorini, M., Midolo, G., Cesco, S., Mimmo, T. & Borruso, L. A Matter of Metals: Copper but Not Cadmium Affects the Microbial Alpha-Diversity of Soils and Sediments — a Meta-analysis. *Microb. Ecol.* (2022) doi:10.1007/s00248-022-02115-4.
4. Konstantopoulos, S. Fixed effects and variance components estimation in three-level meta-analysis. *Res. Synth. Methods* **2**, 61–76 (2011).
5. Trikalinos, T. A. & Olkin, I. Meta-analysis of effect sizes reported at multiple time points: A multivariate approach. *Clin. Trials* **9**, 610–620 (2012).
6. Ishak, K. J., Platt, R. W., Joseph, L., Hanley, J. A. & Caro, J. J. Meta-analysis of longitudinal studies. *Clin. Trials* **4**, 525–539 (2007).
7. Higgins, J. P. T. & Thompson, S. G. Quantifying heterogeneity in a meta-analysis. *Stat. Med.* **21**, 1539–1558 (2002).
8. Nakagawa, S. & Santos, E. S. A. Methodological issues and advances in biological meta-analysis. *Evol. Ecol.* **26**, 1253–1274 (2012).
9. Epron, D. Separating autotrophic and heterotrophic components of soil respiration: Lessons learned from trenching and related root-exclusion experiments. *Soil Carbon Dyn. An Integr. Methodol.* 157–168 (2010) doi:10.1017/CBO9780511711794.009.
10. Hicks Pries, C. E. *et al.* Decadal warming causes a consistent and persistent shift from heterotrophic to autotrophic respiration in contrasting permafrost ecosystems. *Glob. Chang. Biol.* **21**, 4508–4519 (2015).
11. Melillo, J. M. *et al.* Soil warming and carbon-cycle feedbacks to the climate system. *Science (80-)*. **298**, 2173–2176 (2002).
12. Luo, Y., Wan, S., Hui, D. & Wallace, L. L. Acclimatization of soil respiration to warming in a tall grass prairie. *Nature* **413**, 4–7 (2001).
13. Dorrepaal, E. *et al.* Carbon respiration from subsurface peat accelerated by climate warming in the subarctic. *Nature* **460**, 616–619 (2009).
14. Schuur, E. A. G. *et al.* The effect of permafrost thaw on old carbon release and net carbon exchange from tundra. *Nature* **459**, 556–559 (2009).
15. Melillo, J. M. *et al.* Long-term pattern and magnitude of soil carbon feedback to the climate system in a warming world. *Science (80-)*. **358**, 101–105 (2017).
16. Hicks Pries, C. E., Schuur, E. A. G., Natali, S. M. & Crummer, K. G. Old soil carbon losses increase with ecosystem respiration in experimentally thawed tundra. *Nat. Clim. Chang.* **6**, 214–218 (2016).
17. Crowther, T. W. *et al.* Quantifying global soil carbon losses in response to warming. *Nature* **540**, 104–110 (2016).
18. van Gestel, N. *et al.* Predicting soil carbon loss with warming. *Nat. Publ. Gr.* **554**, (2018).
19. Nayak, A. K. *et al.* Current and emerging methodologies for estimating carbon sequestration in agricultural soils: A review. *Sci. Total Environ.* **665**, 890–912 (2019).
20. Poelplau, C., Vos, C. & Don, A. Soil organic carbon stocks are systematically overestimated by misuse of the parameters bulk density and rock fragment content. *Soil* **3**, 61–66 (2017).
21. Goidts, E., Van Wesemael, B. & Crucifix, M. Magnitude and sources of uncertainties in soil organic carbon (SOC) stock assessments at various scales. *Eur. J. Soil Sci.* **60**, 723–739 (2009).
22. Tadiello, T., Perego, A., Valkama, E., Schillaci, C. & Acutis, M. Computation of total soil organic carbon stock and its standard deviation from layered soils. *MethodsX* **9**, 101662 (2022).
23. Zhi, J. *et al.* Estimating soil organic carbon stocks and spatial patterns with statistical and GIS-based Methods. *PLoS One* **9**, 30–33 (2014).
24. Rustad, L. E. *et al.* A meta-analysis of the response of soil respiration, net nitrogen mineralization, and aboveground plant growth to experimental ecosystem warming. *Oecologia* **126**, 543–562 (2001).

Reviewer Reports on the First Revision:

Referees' comments:

Referee #1 (Remarks to the Author):

This submission contains a revised manuscript that showcases the results of a meta-analysis utilizing a unique ecosystem respiration database from tundra warming experiments. The topic is captivating, and the authors deserve recognition for their diligent work in compiling this data. The manuscript provides a comprehensive overview of the meta-analysis findings and supplements them with an upscaling approach. However, while the results are interesting, I noticed issues that could lead to an overinterpretation of the findings. As a result, I have reservations about publishing this manuscript in its present form.

Major concerns

-Rebuttal letter: The authors provided an extensive response to reviewers' comments clarifying multiple points and explaining the rationale for several aspects of the study. That said, I have a major concern regarding the main goal of the manuscript and their response to Reviewer #3. The authors clarify that the current story is "truly about: mechanistic drivers of changes in ecosystem respiration due to warming" (rebuttal page 24)..."Adding GPP to the story would not allow us to explore the drivers of RE in this much depth since the two processes (GPP vs. ER) are driven by fundamentally different drivers." (rebuttal page 24)

This argument is concerning because the authors deny that GPP and ER are processes that are inherently related and multiple drivers (such as temperature and moisture) influence both processes. It is widely known that GPP and ER share influence from similar drivers (e.g., temperature and moisture), and therefore separating the direct and indirect influence on these drivers is complicated and results in confounding effects. Arguably, many of the interpretations and analyses provided in this study show the challenges of not accounting for confounding effects. The response to Reviewer #3 shows that the authors consider GPP a nuisance for explaining how increases in temperature influence ER and claim that by ignoring GPP, they can "explore the drivers of ER in much depth".

By neglecting GPP, the authors explore a reductionist approach for the mechanisms that control ER. Neglecting information on GPP in this study is supported by a meta-analysis using only 9 sites where partitioning of ER was performed. The authors observed that warming (in 9 sites) resulted in an increase of ~50% for Ra and Rh, claiming that there was an equal warming-induced increase in plant- and microbial-respiration. This analysis has the limitation that it was done in 9 out of 136 sites (<10% of the sites) and challenges its implication for representation of the global tundra. In addition, if >50% of the response of ER is potentially driven by Ra, then why not include the effect of GPP in the overall study?

In summary, the authors take a reductionist approach for this analysis, neglecting the influence of GPP on ER, and do not explore the drivers of ER in much depth as proposed in their objective. I understand the challenges of partition ER and the overall effort for developing synthesis studies, but I encourage the authors to consider the limitations of the analysis and their subsequent interpretations for their manuscript.

- The authors have rephrased the hypotheses, but the expectation of a positive ER response to warming in these ecosystems is not novel (H1). What would be more interesting is to quantify the relative influence of Ra or Rh with respect to warming, but the authors are analyzing the overall ER due to methodological challenges for partitioning into Ra and Rh and a limited sample size for this analysis. Also, the general expectation that the response of ER magnitude varies across time and space is not novel. Hypothesis 2 (H2) is more mechanistic, but it is also framed in a very general way that follows current expectations that local conditions and context-dependent variation influence how ER responds to warming. Reviewer 3 has provided an interesting and important discussion about other processes (i.e., gross primary productivity [GPP]) that are important to understanding the effect of warming on ER but were neglected in the current study. Ultimately, the method for data analysis is an empirical approach that is influenced by the input variables. Removing or adding variables will likely influence results, and missing important variables known to be important biological drivers (e.g., GPP) may bias the mechanistic interpretation of the results.

- The Q10 value of 6.8 [CI 5.6-10.9] is very high and arguably suggests confounding effects. Why would the authors expect such a large Q10 value, and how is it biophysically possible? Could it be an empirical error based on the nature of the measurements? Reporting such a high Q10 value in Nature and potentially used in models have a large ethical and scientific responsibility. This large Q10 value is a result of the deterministic nature of the way Q10 is calculated and is likely an overestimation of this value (Subke and Bahn 2010). Alternatively, this could be a calculation error on how Q10 was derived (see Supp Methods 2) or the way this apparent Q10 should be interpreted as it is derived from a response ratio. In summary, the Q10 value of 6.8 [CI 5.6-10.9] is very high and requires verification in calculations and interpretation.

- Interpretation of ER responses to warming across time. The authors show that, overall, warming increases ER across 136 sites (Figure 3a and elsewhere). However, the demonstration and explanation that ER increases with time due to warming is not supported. The authors showed that there is no temporal relationship when all information is combined. Thus, there is no clear evidence that increased warming will constantly increase ER through time. This message is not clear in the current version of the manuscript (lines 223-233) and gives the impression that there is a positive ER response through time. The changes in the effect direction over time are unclear and show substantial variation across the years in the experiments. Hence, the title's wording of "persistently increased" is questionable because the overall effect of warming is not consistent over time as there are instances of negative or no relationship of this effect over time.

- Reviewer #2 brought up an interesting point about using a SEM approach, and the authors explained that this was not possible because of the structure of the dataset (e.g., different sample sizes, and different variables at different sites). The authors responded that when they applied a multi-model-regression approach (albeit with a limited sample size), they did not find any significant relationships between a selected set of drivers and Er response. This opens the discussion for the interpretation of the results where the authors test the "independent" influence of a variable on the response (e.g., ER Hedges SMD, Figure 4). Is it possible that the individualistic empirical relationship is apparent because it neglects confounding (indirect and direct) or additive effects? Why would these relationships disappear when multiple variables are tested together? Some of the

relationships were tested with a substantially smaller sample size, so how robust are these results? Therefore, the current interpretation of the mechanisms (by testing them independently) may be biased, and the authors should consider these limitations. The authors expressed that multi-factor analyses were not significant in the response to the reviewers, but this statement was omitted in the methods section (page 31, lines 167-172).

- The upscaling approach may need to be revisited or deleted from the study as suggested by previous reviewers. This analysis comes as an add-on, and the authors imply in the main text that this is not a robust calculation but is an interesting exercise. The methods show that the model is parameterized with only 37 points (page 35, lines 264-265), which is very limited for predicting to such an extent region. Furthermore, this approach assumes a constant increase in ROM based on TN and C:N and overemphasizes the dependency on TN (as TN was used to calculate C:N) in the model. This imposed dependency (because of the fixed selection of variables) likely influences the results and, therefore, the spatial patterns and magnitudes reported in Figure 6. There is also a temporal mismatch between the baseline ER (years 1960-2011) and plant respiration (year 2015). Finally, Figure 6 should have the predictions and the associated uncertainty from Supplementary figure 7 to show how large the errors are in the predictions. The authors claim that some areas will have >75% change in ER, but their model also predicts a relative standard deviation for the change in ER of >175%. By hiding this results in the supplementary materials the authors are not being transparent with their results in the main text (Figure 6). Ultimately what these results show is that although some areas could have an increase of >75% the authors are not sure if this could happen because the model has an error of >175%. I strongly suggest removing the upscaling approach from this manuscript to avoid potential over and mis interpretation of the results.

Comments in detail

Line 90-91 – The fact that increase in ER is both by plant-related (i.e., autotrophic respiration) and microbial respiration (i.e., heterotrophic respiration) is not novel enough as these two processes influence ER by definition. What would be more interesting is to know which are the contributions of the increase in ER due to autotrophic or heterotrophic components.

Lines 95-96 – In the response to the reviewers, the authors acknowledge that it is known that local soil conditions are important for ER. So, I wonder how relevant and novel is this statement as is widely known that these conditions are important for ER. Arguably, what is important is to identify how increase in warming influence local soil conditions and consequently other processes responsible for ER.

Lines 101-104 – I fully agree with this statement, and multiple manuscripts authored by the co-authors of this study have reported similar results. That said, the second statement that ER will be higher than plant productivity (lines 104-105) was not tested in this manuscript despite one reviewer acknowledging the important link between ER and GPP. This may have been a missing opportunity and arguably a missing link to explain the net effect of raising temperature on the carbon balance of these ecosystems.

Lines 169 – Is assigning higher importance to datasets with lower variance a good approach? Arguably context-dependent conditions are important, and within-site variability is the norm for soil processes, so giving more weight to sites that have lower spatial variability may include bias. Would it have been better to add more weight to experiments with longer time records?

Figure 3b – Remove the lines for no significant slopes. Including these lines gives an incorrect perspective.

Line 257-259 – Results show that both RH and RA increase with warming, but the authors place emphasis on RH and neglect the drivers of RA or the potential interactions between RA and RH (e.g., substrate supply).

Line 322-327 – This conclusion is influenced by a small sample size, and the authors should consider confounding effects that may influence this result, as discussed earlier. The authors attribute that changes originated in the deeper mineral layer, but not all possible drivers (mainly for RA) were considered in their calculations. Is this a biased result based on the variables selected and how the empirical analysis was performed? Do adding different variables or interactions would result in a similar conclusion? Is this an overinterpretation of the study?

Lines 364-367 – As discussed earlier, the analysis neglects several variables of interest and arguably does not fully evaluate the mechanisms involved. I think this section can be toned down to avoid over interpretation.

Lines 375-377 – Agreed, but this was not considered in this study.

Lines 386-387 – The graph shows a decrease in the response, then an increase and then no change. Overall, the effects are positive but the data does not support that “persistently increase” over time. For example, see wording in line 389, indicating a “nonlinear pattern of ER increase over time”, which is not supported by the data. The results only say that overall, warming increases 30% ER throughout the experiments, but there is no clear trend over time. This is a matter of semantics, but the wording needs to be clarified throughout the manuscript.

Line 404-407 – Overall, this statement seems realistic, but the authors can test this if GPP was included in the analysis as suggested by Reviewer #3.

Lines 409-411 – Again, the authors imply increased changes with time. This was not supported, so I think the statement needs to be revisited to clarify that the warming effect is sustained through time, but there is no evidence of increases over time.

Page 25 lines 37-46 – How often were the measurements performed at sites? Do the authors account for sampling intervals? There is evidence that manual sporadic measurements may not properly represent climate change effects in field experiments (Vicca et al. 2014). More information is needed to understand data collection and data aggregation in this section. Finally, lines 44-46 show that these measurements may not be ER as they do not include RA during day.

Supplementary Figure 7 – This should be Figure 6 in the main text to show the large uncertainty associated with the approach.

Supplementary Figure 9 – This is an example that the few data points used for parameterizing the model do not represent the statistical space for the predictions. Therefore, the upscaling is based on a very limited sample size that does not represent the statistical parameter space for predictions.

References

Subke, Jens-Arne, and Michael Bahn. 2010. "On the 'Temperature Sensitivity' of Soil Respiration: Can We Use the Immeasurable to Predict the Unknown?" *Soil Biology & Biochemistry* 42 (9): 1653–56.

Vicca et al. 2014. "Can Current Moisture Responses Predict Soil CO₂ Efflux under Altered Precipitation Regimes? A Synthesis of Manipulation Experiments." *Biogeosciences* 11 (11): 2991–3013.

Referee #2 (Remarks to the Author):

The authors have addressed my comments in a thorough and very thoughtful manner. The analyses are carefully justified, robust and carried out and documented fully. The study not only provides a very important summary of critical ecosystem responses to warming using an enormous data set, but also is a model for how to do complex ecological meta-analyses. My one remaining comment is to suggest that in Figure 2 the grand mean be made to stand out more, with a larger symbol and a different color. That little diamond is almost lost in that sea of data!

I should have signed my original review (I was Referee 2). It is a bit overwhelming and enormously satisfying to see how far you have taken these analyses beyond what I introduced 22 years ago (Rustad et al. 2001). --Jessica Gurevitch

Referee #3 (Remarks to the Author):

Review by Ivan Janssens

Having read the response letter and the new version, I now have a better understanding of the study. Most of my methodological issues have been well addressed, but I do have two remaining issues.

The first is the extrapolation exercise, which still does not convince me and is in my view also not necessary. The hotspot areas where the largest warming-induced increases in RE are expected are vastly understudied, rendering the global driver data that underly the map hugely uncertain in those areas, and there are also hardly any empirical data available to corroborate these results. I would suggest to remove the map or at least state more explicitly that the driver maps are highly

uncertain in most areas and that empirical evidence is now needed to verify the existence of the modelled hotspots.

The second issue relates to the message that the end of the abstract is conveying. '... our findings imply a potentially strong, positive climate feedback from the tundra during the coming decades, even under modest warming scenarios. Since persistently enhanced respiration with warming is likely to free up more carbon than can be compensated for by increased plant productivity, the tundra biome may accelerate, rather than mitigate the impact of global warming'.

In the response letter the authors state that they are aware that this study does not provide evidence for the climate-carbon feedback, but with the abstract ending like this and the knowledge that most journalists will not thoroughly read the manuscript and will ignore the 'potentially' and 'may', this is exactly how it will be received by the public. I suggest to add a clause like: 'if not accompanied by increased carbon inputs by vegetation' or 'further research is needed to study whether the increased respiration is actually resulting in carbon loss from these tundra ecosystems'. This is much clearer than the conditional phrasing in the current abstract.

Overall, I congratulate the authors with this interesting study.

Response letter 2 - September 2023

Referee #1

1. This submission contains a revised manuscript that showcases the results of a meta-analysis utilizing a unique ecosystem respiration database from tundra warming experiments. The topic is captivating, and the authors deserve recognition for their diligent work in compiling this data. The manuscript provides a comprehensive overview of the meta-analysis findings and supplements them with an upscaling approach. However, while the results are interesting, I noticed issues that could lead to an overinterpretation of the findings. As a result, I have reservations about publishing this manuscript in its present form.

- We thank reviewer #1 for the constructive comments and feedback. The authors believe that some of these reservations are due to unclear communication about which data is available and the specific novelty of our study.

We therefore tried to highlight in the revised ms that:

- a. Our analyses provide the **most comprehensive overview of ecosystem respiration measurements in OTC warming experiments in the tundra** that currently exists to our knowledge (136 datasets from 56 experiments),
- b. By focusing on ecosystem respiration (ER) alone as a response here instead of the whole carbon cycle (also GPP and NEE), we aimed to provide **valuable insights into the respiration component**, which is relatively less influenced by light conditions, while still recognizing the importance of GPP indirectly through inclusion of NPP and plant biomass.
- c. The respiration partitioning ‘smaller’ meta-analysis (9 datasets) added during the first revision round as response to reviewer #3 comments were **not a subsetted dataset by choice**, but instead **reflects the available partitioning data from the 136 experiments** included in the larger meta-analysis on ecosystem respiration.

Major concerns

2. Rebuttal letter: The authors provided an extensive response to reviewers’ comments clarifying multiple points and explaining the rationale for several aspects of the study. That said, I have a major concern regarding the main goal of the manuscript and their response to Reviewer #3.

(1) The authors clarify that the current story is “*truly about: mechanistic drivers of changes in ecosystem respiration due to warming*” (rebuttal page 24)...”*Adding GPP to the story would not allow us to explore the drivers of RE in this much depth since the two processes (GPP vs. ER) are driven by fundamentally different drivers.*” (rebuttal page 24)

This argument is concerning because the authors deny that GPP and ER are processes that are inherently related and multiple drivers (such as temperature and moisture) influence both processes. It is widely known that GPP and ER share influence from similar drivers (e.g., temperature and moisture), and therefore separating the direct and indirect influence on these drivers is complicated and results in confounding effects. Arguably, many of the interpretations and analyses provided in this study show the challenges of not accounting for confounding effects. The response to Reviewer #3 shows that the authors consider GPP a nuisance for explaining how increases in temperature influence ER and claim that by ignoring GPP, they can “explore the drivers of ER in much depth”.

By neglecting GPP, the authors explore a reductionist approach for the mechanisms that control ER.

- The authors believe that concentrating on ER alone offers in-depth insights and warrants our current approach, which we explain better in the revised ms (L 184-189 in Main Text and L102-112 in Methods). That is:
 - a. We incorporated variables related to plant abundance and growth such as Net Primary Productivity (NPP) and aboveground plant biomass measurements in our analyses (**Fig. 1, Ext. Fig. 2, Ext. Table 5c**). We also analysed the partitioning on all the available data after checking with the data contributors. Thus, our study *does* take the contribution of autotrophic processes for ER into account. NPP represents the net amount of carbon assimilated by plants, which contributes to both plant growth and influence both plant and soil respiration. Plant biomass, however, accounts for the stored carbon in plant tissues, which can later be subject to decomposition. By including these variables, we indirectly considered the influence of GPP without directly measuring it.
 - b. Our intention was not to suggest that GPP was irrelevant or a nuisance in explaining the influence of temperature on ER, or to use it as an excuse for excluding it from the analysis. Yet, our database does not include GPP data, because it is difficult to consistently incorporate GPP data into ER datasets from dark chambers due to the light-dependency of photosynthesis and the associated high spatio-temporal variability of GPP measurements (i.e. reflecting different light conditions). This would make accurate inclusion and comparison of GPP data across time and across multiple experiments challenging. However, since both processes (GPP, ER) are partly driven by the activity of the vegetation, we argue that it is more meaningful to incorporate the role of the response of plant growth to warming for ecosystem respiration here to increase our mechanistic understanding of the underlying ecological drivers of ER response to warming. We therefore included (changes in) plant biomass and community composition, as well as net primary productivity (NPP) as potential drivers of ER (**Ext. Table 5b-c**).
 - c. In sum, by focusing on ER, we provide **valuable insights into the respiration component, which is less influenced by light conditions, while still recognizing the importance of GPP indirectly through NPP and plant biomass.**

(2) Neglecting information on GPP in this study is supported by a meta-analysis using only 9 sites where partitioning of ER was performed. The authors observed that warming (in 9 sites) resulted in an increase of ~50% for R_a and R_h , claiming that there was an equal warming-induced increase in plant- and microbial-respiration. This analysis has the limitation that it was done in 9 out of 136 sites (<10% of the sites) and challenges its implication for representation of the global tundra. In addition, if >50% of the response of ER is potentially driven by R_a , then why not include the effect of GPP in the overall study?

In summary, the authors take a reductionist approach for this analysis, neglecting the influence of GPP on ER, and do not explore the drivers of ER in much depth as proposed in their objective. I understand the challenges of partition ER and the overall effort for developing synthesis studies, but I encourage the authors to consider the limitations of the analysis and their subsequent interpretations for their manuscript.

- Even though 9 is only 7% of the 136 datasets included in our study, it represents all the available information at the moment. This less frequent measuring of ER partitioning is likely due to methodological challenges, i.e. the *isotopic labeling* partitioning method is very resource intensive and challenging to use over extended periods, and the *vegetation clipping* method is labor intensive and disruptive to the ecosystem being studied. This smaller sample size for partitioning vs. total ER data actually highlights an important gap in ER data collection, which only further underscores the value of our sub-meta-analysis.

- Further, 9 replicates is still a relatively high replication for an ecological meta-analysis. These nine studies also proved to be a good representation of all studies as outlined in **Supplementary Discussion 1**.
- In sum, our additional sub-meta-analysis represents **the most comprehensive compilation of ER partitioning from the same tundra warming experiments, and provides novel and valuable insights on the magnitude of the ER partitioning**.

3. Hypotheses:

(1) H1: The authors have rephrased the hypotheses, but the expectation of a positive ER response to warming in these ecosystems is not novel (H1). What would be more interesting is to quantify the relative influence of Ra or Rh with respect to warming, but the authors are analyzing the overall ER due to methodological challenges for partitioning into Ra and Rh and a limited sample size for this analysis. H2: Also, the general expectation that the response of ER magnitude varies across time and space is not novel. Hypothesis 2 (H2) is more mechanistic, but it is also framed in a very general way that follows current expectations that local conditions and context-dependent variation influence how ER responds to warming.

- Based on previous studies from individual or multiple sites, we indeed expected an overall positive effect of warming. The novelty of our study therefore does not lie with identifying *whether* the effect is positive, but what determines the magnitude of the mean effect across the large spatiotemporal gradient that our dataset covers. Our study is unique and novel because **no single study has assessed the ER response to warming yet across a large spatiotemporal environmental gradient in the tundra**, using a standardized experimental setup while also including an extensive set of environmental drivers. To clarify the novelty of our study even better, we have revised our hypotheses on L 154-158, and detailed why our database is novel and unique on L 161-164.
- As explained in response to your comment #2, we extensively reached out to all data contributors to gather respiration partitioning data and found that 9 studies had suitable respiration partitioning data. We stress now on L 226 that this is *all* the available partitioning data ‘out there’ from the 136 studies that measured total ER.

(2) Reviewer 3 has provided an interesting and important discussion about other processes (i.e., gross primary productivity [GPP]) that are important to understanding the effect of warming on ER but were neglected in the current study. Ultimately, the method for data analysis is an empirical approach that is influenced by the input variables. Removing or adding variables will likely influence results, and missing important variables known to be important biological drivers (e.g., GPP) may bias the mechanistic interpretation of the results.

- Please see our response to your comment #2, and our response to reviewer 3 in our previous response letter for justification for the lack of GPP data in our study.

4. The Q10 value of 6.8 [CI 5.6-10.9] is very high and arguably suggests confounding effects. Why would the authors expect such a large Q10 value, and how is it biophysically possible? Could it be an empirical error based on the nature of the measurements? Reporting such a high Q10 value in Nature and potentially used in models have a large ethical and scientific responsibility. This large Q10 value is a result of the deterministic nature of the way Q10 is calculated and is likely an overestimation of this value (Subke and Bahn 2010). Alternatively, this could be a calculation error on how Q10 was derived (see Supp Methods 2) or the way this apparent Q10 should be interpreted as it is derived from a response ratio. In summary, the Q10 value of 6.8 [CI 5.6-10.9] is very high and requires verification in calculations and interpretation.

- We have had similar concerns ourselves, and therefore refer to our calculations as ‘*apparent* Q10’ which differs from a Q10 calculated from laboratory settings or enzymatic rates¹⁻⁴. We further stress that (i) we believe the Q10 value is correctly calculated, (ii) it is not a measure of how ER would increase in absolute numbers with 10°C of warming, but how the response ratio would increase, and (iii) we still find it a valuable number to illustrate temperature sensitivity across studies. We have added this nuance in the interpretation of Q10 to the **Main Text** (L 222-224) and **Supp. Methods 2** now (L 133-139).
- Yet, we do not wish to mislead or confuse readers and therefore would agree with removing it from the manuscript upon the editor’s request as well. This is because (i) removal of Q10 would not impact the main conclusion or novelty of our study and (ii) there is in general discussion about the use of Q10² to quantify temperature sensitivity of ecological processes such as decomposition or respiration.

5. Interpretation of ER responses to warming across time. The authors show that, overall, warming increases ER across 136 sites (Figure 3a and elsewhere). However, the demonstration and explanation that ER increases with time due to warming is not supported. The authors showed that there is no temporal relationship when all information is combined. Thus, there is no clear evidence that increased warming will constantly increase ER through time. This message is not clear in the current version of the manuscript (L 223-233) and gives the impression that there is a positive ER response through time. The changes in the effect direction over time are unclear and show substantial variation across the years in the experiments. Hence, the title's wording of "persistently increased" is questionable because the overall effect of warming is not consistent over time as there are instances of negative or no relationship of this effect over time.

- We do not agree with the following statements by the reviewer and explain below why:
“there is no clear evidence that increased warming will constantly increase ER through time”
“... gives the impression that there is a positive ER response through time. The changes in the effect direction over time are unclear and show substantial variation across the years in the experiments.”
“the title's wording of "persistently increased" is questionable because the overall effect of warming is not consistent over time as there are instances of negative or no relationship of this effect over time.”
- Our model results provide evidence that increased warming will constantly increase ER through time, given that none of the effect sizes within a warming duration age class, whether presented as mean points (**Fig. 3a**) or as temporal trends (**Fig. 3b**) dip below zero. This continued positive effect size, representing the change in ER with warming, implies continued increased ER with warming across the time window within which we have data (warming between 0-25 years).
- The reviewer is perhaps hinting here at the raw data behind our model results, which indeed (expectedly) show variation in the effect size across the years, including some experiments showing a respiration decrease with warming in some years (**Fig. 3: Hedges SMD < 0**). We believe this then becomes a discussion/question on what one should interpret in the ms: the raw data points or the model results. We focused on interpreting the model results throughout our manuscript because we trust the accuracy and robustness of our meta-analysis and -regression approach to comprehensively analyze our data. However, we acknowledge that an ideal scenario would be to assess long-term temporal trends within the same sites (cfr. Melillo experiment^{5,6}). Hence, we eliminated the word ‘persistent(ly)’ from our ms to avoid confusion, pointing to the exact findings in our models (increased ER with warming over time) and not to our interpretation of them (persistence in the increased ER).

- Overall, this comment reiterated how vital correct interpretation of the figures and main effect size is that we use in the ms. The recurring effect size visualized on the y-axes in **Figs. 2-5**, ER Hedges SMD, represents the *change* in respiration with experimental warming, and not the absolute value of respiration. To elucidate this, we now add the interpretations of this effect size, i.e. ER \uparrow with warming above x-axis, ER \downarrow with warming below x-axis, to all these figures.

6. Reviewer #2 brought up an interesting point about using a **SEM approach**, and the authors explained that this was not possible because of the structure of the dataset (e.g., different sample sizes, and different variables at different sites). The authors responded that when they applied a multi-model-regression approach (albeit with a limited sample size), they did not find any significant relationships between a selected set of drivers and Er response. This opens the discussion for the interpretation of the results where the authors test the “independent” influence of a variable on the response (e.g., ER Hedges SMD, Figure 4). Is it possible that the individualistic empirical relationship is apparent because it neglects confounding (indirect and direct) or additive effects? Why would these relationships disappear when multiple variables are tested together? Some of the relationships were tested with a substantially smaller sample size, so how robust are these results? Therefore, the current interpretation of the mechanisms (by testing them independently) may be biased, and the authors should consider these limitations. The authors expressed that multi-factor analyses were not significant in the response to the reviewers, but this statement was omitted in the methods section (page 31, L 167-172).

- We think the reviewer refers to our response to comment #8 from reviewer #2 in the initial response letter, but that this was misinterpreted. We had said:

*“Besides, a limited SEM using drivers for which all datasets had an observation would be relatively comparable to a multi-model meta-regression approach. However, we do not mention the result of this in the main text because (i) it was only possible for a rather limited, random selection of drivers (Zone, soil C stock, pH class, vegetation class and NPP), and (ii) we did not find any significant relationships between these drivers and the ER response with the multi-factor model either (cfr. single-factor model results, **Ext. Table 5c**).”*

We indeed applied a multi-model regression approach where we thought it relevant, i.e. for those environmental drivers where the majority of experiments had a data point, and sample size variation was small (between 131-136, **Supp. Table 2**). These drivers were either derived from global maps (Zone, Permafrost probability, Net primary productivity, Soil C stock), or submitted by data contributors as expert knowledge about the soil and vegetation conditions in their experiment (Soil pH class, Vegetation class, Soil moisture class). **We did not find a significant relationship between these drivers and the ER Hedges SMD, neither with a multi-factor model, nor with the single-factor models** of each of these drivers separately. Because this was the only combination of drivers relevant to add to a multi-factor model without losing power due to unevenly distributed sample size (**Supp. Table 2**), we report in our ms only the single-factor model results to avoid confusion among readers that we used different approaches, which then also differed depending on the specific driver. **Since our results for those drivers where we ran both types of models (multi and single factor) did not differ between the model approach, we think it is sensible to only report the single-factor results in the ms.**

- As explained above, we do not agree with the reviewer that “*these relationships disappear when multiple variables are tested together*”. but think that this is a misinterpretation of our response letter from July 2023. There were no cases where there were significant relationships between drivers and the ER Hedges SMD in a multi-factor model that turned non-significant when running the same drivers in single-factor models.
- In sum, in an ideal scenario, we would have environmental driver data available for all 136 datasets, and based on field measurements. In such a case, multi-factor models and/or an SEM approaches

would be suitable to dive deeper into the relationships between environmental factors and the respiration response. Such ‘ideal dataset’ is non-existent at the moment, however, and we attempted with our current database to provide the most comprehensive overview of the tundra respiration response to warming. We include environmental drivers that might drive this response, using currently available data from a mix of data sources (global datasets, site expert knowledge, field measurements). We chose single-factor models to have as much power in the models as possible, and multi-factor models would severely restrict this because the sample size of all drivers ranges between 16-136 (**Supp. Table 2**). Finally, though the reviewer rightly points out that our formulation may have been somewhat unclear in the rebuttal letter, we assume this was clear enough in the main text (which is most important for the readers) as the reviewer does not raise this point there.

7. The upscaling approach may need to be revisited or deleted from the study as suggested by previous reviewers. This analysis comes as an add-on, and the authors imply in the main text that this is not a robust calculation but is an interesting exercise. The methods show that the model is parameterized with only 37 points (page 35, L 264-265), which is very limited for predicting to such an extent region. Furthermore, this approach assumes a constant increase in ROM based on TN and C:N and overemphasizes the dependency on TN (as TN was used to calculate C:N) in the model. This imposed dependency (because of the fixed selection of variables) likely influences the results and, therefore, the spatial patterns and magnitudes reported in Figure 6. There is also a temporal mismatch between the baseline ER (years 1960-2011) and plant respiration (year 2015). Finally, Figure 6 should have the predictions and the associated uncertainty from Supplementary figure 7 to show how large the errors are in the predictions. The authors claim that some areas will have >75% change in ER, but their model also predicts a relative standard deviation for the change in ER of >175%. By hiding this results in the supplementary materials the authors are not being transparent with their results in the main text (Figure 6). Ultimately what these results show is that although some areas could have an increase of >75% the authors are not sure if this could happen because the model has an error of >175%. I strongly suggest removing the upscaling approach from this manuscript to avoid potential over and misinterpretation of the results.

- We acknowledge the concern of the reviewer, and similarly of reviewer #3 (comment #2). We decided not to remove the upscaling exercise from the main text, because we believe it provides insight in the overall impact and spatial sensitivity of the tundra ER response to warming, and readers can relate to this by looking at the location that they are working in. We do acknowledge the large uncertainty in our upscaling, originating largely due to high uncertainty in input data (see below), and argue that it is important to show that with the current limited data availability, the uncertainty will be high for arctic upscaling.
- To meet the concerns of both reviewers, we...
 - Now include the uncertainty map (formerly in **Supp. Fig. 7**) in the figure in the main text, which highlights the high uncertainty behind the upscaling calculations. We also did an extra analysis where we estimated the proportion of the uncertainty originating from the input data (gridded soil data) compared to the uncertainty originating from the meta-regression upscaling model (see revised **Fig. 6**). In that we show that in almost entire study area, most of the uncertainty originates from the input data – in average around 63% over entire study area. We mention on L 366-367 that this high uncertainty implies for us that spatial sampling gaps across the tundra biome should urgently be addressed in follow-up studies⁷ (see also our response to your comment 22). In mentioning this, we immediately show the high

uncertainty of the upscaling, as well as explain what the implications might be, and how our exercise has brought this forward.

- Rephrase the goal of our upscaling from identifying hotspots in expected ER increase with warming to ‘visualizing spatial differences in the sensitivity of the response’ combined with facilitating a ‘crude estimate of how much additional C would be respired by 1.4 °C warming compared to present temperatures’.

Revised Fig 6.

Comments in detail

8. Line 90-91 – The fact that increase in ER is both by plant-related (i.e., autotrophic respiration) and microbial respiration (i.e., heterotrophic respiration) is not novel enough as these two processes influence ER by definition. What would be more interesting is to know which are the contributions of the increase in ER due to autotrophic or heterotrophic components

- Given the limited data available on ER partitioning, we believe that this type of analysis can only represent a preliminary result and should be confirmed or refuted by larger, more representative datasets in the future. We would like to emphasize here that for the data used in this study, we have already collected all available data on ER partitioning, and we did not have more data at hand (see our detailed response to your comment #2).

9. L 95-96 – In the response to the reviewers, the authors acknowledge that it is known that local soil conditions are important for ER. So, I wonder how relevant and novel is this statement as is widely known that these conditions are important for ER. Arguably, what is important is to identify how

Environmental drivers of ~~persistently~~ increased ecosystem respiration with warming in the tundra

increase in warming influence local soil conditions and consequently other processes responsible for ER.

- The reviewer refers to our previous statement in the abstract:

“These results highlight the importance of local soil conditions, and warming-induced changes therein, for accurate estimations of future climatic impacts on respiration.”

Though we agree it is known that local soil conditions are important for ER, we reiterate that the novelty in our study arises from the large spatiotemporal gradient covered with our data, which makes the specific findings highly relevant to be used in future studies, e.g. for benchmarking climate models.

We agree though that this statement jumped too quickly from the finding (i.e. soil conditions influence ER response to warming over large spatiotemporal scale) to the implication (i.e. these data are useful to benchmark climate models), and therefore revised it as follows (L 95-97):

“These results highlight the importance of local soil conditions, and warming-induced changes therein, for future climatic impacts on respiration, and can be used to improve model estimations.”

10. L 101-104 – I fully agree with this statement, and multiple manuscripts authored by the co-authors of this study have reported similar results. That said, the second statement that ER will be higher than plant productivity (L 104-105) was not tested in this manuscript despite one reviewer acknowledging the important link between ER and GPP. This may have been a missing opportunity and arguably a missing link to explain the net effect of raising temperature on the carbon balance of these ecosystems

- Please see our response to your comment #2 on the ‘missed opportunity’ of not including GPP.
- On the statement that ‘ER will be higher than plant productivity’: we have adjusted this statement on L 101-103 and 432-433 to clarify our focus is on ER, and that statements/findings on the carbon balance are outside the scope of our manuscript.

11. L 169 – Is assigning higher importance to datasets with lower variance a good approach? Arguably context-dependent conditions are important, and within-site variability is the norm for soil processes, so giving more weight to sites that have lower spatial variability may include bias. Would it have been better to add more weight to experiments with longer time records?

- Assigning higher importance or ‘weights’ to datasets with higher precision (i.e., lower variance) is a standard approach in meta-analysis. For clarification, we add the weight values for each dataset to **Supp. Table 1**. We do not think we should give ‘more weight to experiments with longer time records’ in our statistical approach, as we take into account this temporal aspect (i.e. the fact that some experiments have multiple years of ER data while others do not) by including a spatial autocorrelation structure in all meta-models (CAR-structure, see Methods L 190-198). Further, we reiterate that the meta-analysis with or without the two sites with longer time series (ALA_1, GRE_6) did not show different results (**Supp. Discussion 5**). This implies that the weight approach used here likely does not impact the results.

12. Figure 3b – Remove the Lines for no significant slopes. Including these Lines gives an incorrect perspective.

- Agreed, we have adjusted this.

13. Line 257-259 – Results show that both RH and RA increase with warming, but the authors place emphasis on RH and neglect the drivers of RA or the potential interactions between RA and RH (e.g., substrate supply).

Environmental drivers of ~~persistently~~ increased ecosystem respiration with warming in the tundra

- We revised the sentence on L 271-274, it now reads “*Combined with our ER partitioning results showing increased heterotrophic as well as autotrophic respiration with warming, stimulation of both microbial and plant activity might lead to the observed increase in ER with warming (Ext. Table 4b, Supp. Discussion 1).*”

14. Line 322-327 – This conclusion is influenced by a small sample size, and the authors should consider confounding effects that may influence this result, as discussed earlier. The authors attribute that changes originated in the deeper mineral layer, but not all possible drivers (mainly for RA) were considered in their calculations. Is this a biased result based on the variables selected and how the empirical analysis was performed? Do adding different variables or interactions would result in a similar conclusion? Is this an overinterpretation of the study?

- On the ‘*possible confounding effects and not considering all possible drivers in our calculations*’, see our response to your comment #6 (we believe this is due to a misinterpretation).
- The reviewer also wonders if the sample size behind the models demonstrating significant relationships between environmental drivers and the ER Hedges SMD could bias our results. The significant drivers comprised the *warming-induced change in mineral soil total N* (42 datasets) & in soil pH (27), see **Fig. 4**; and the *soil total N* (43) & *CN-ratio* (39), see **Fig. 5**. Though we recognize that our single-factor approach does not allow to investigate feedbacks/interactions between the significant drivers, **investigating these main effects is, we believe, already a key first step in understanding what drives increased respiration in the tundra with warming.** The database we gathered is to our knowledge the largest field-based ER dataset gathered from any biome up to now. **We hope that our study will provide a momentum for follow-up studies aiming for more standardized field measurements across all experiments, so that a next step can be to investigate interactive effects and feedbacks.** For now, we trust that these results are robust as our additional sensitivity analyses demonstrates that the results do not change under different sample size restriction scenarios, for the significant environmental drivers (**Supp. Discussion 4, Supp. Fig. 4**).

15. L 364-367 – As discussed earlier, the analysis neglects several variables of interest and arguably does not fully evaluate the mechanisms involved. I think this section can be toned down to avoid over interpretation.

- We assume the reviewer implies here that our approach does not fully evaluate the mechanisms involved because we do not study feedbacks or interactions among environmental drivers. Please see our response to your previous comment #14 for that. We decided to leave the paragraph as is.

16. L 375-377 – Agreed, but this was not considered in this study.

- The reviewer refers here to the statement on the land carbon-climate feedback, i.e. “*The magnitude of the long-term land carbon-climate feedback critically depends on the balance between C uptake through changes in photosynthesis activity and C release through changes in ER*”. We agree that statements referring to the feedback can mislead readers into thinking we have studied also the effects of warming on C uptake through increased photosynthesis. We acknowledge on L 185 (Main Text) and L 102 (Methods) in the revised ms that we do not focus on C uptake, but solely on changes in C release (ER), and also clarified it in the statement on L 401.

17. L 386-387 – The graph shows a decrease in the response, then an increase and then no change. Overall, the effects are positive but the data does not support that “persistently increase” over time. For

Environmental drivers of ~~persistently~~ increased ecosystem respiration with warming in the tundra

example, see wording in line 389, indicating a “nonlinear pattern of ER increase over time”, which is not supported by the data. The results only say that overall, warming increases 30% ER throughout the experiments, but there is no clear trend over time. This is a matter of semantics, but the wording needs to be clarified throughout the manuscript.

- Please see our response to your comment #5 for this. We only partially agree that there is ‘*no clear trend over time*’. While we find no significant trend between 0-25 years of cumulative warming, we do find a significant decrease followed by a significant increase over time when looking at the data in age classes (facilitating nonlinear trend detection) (**Fig. 3b**). We think this finding is valuable and should not be ignored. It is also in line with the single (non-tundra) study that addressed this experimentally⁵.

18. Line 404-407 – Overall, this statement seems realistic, but the authors can test this if GPP was included in the analysis as suggested by Reviewer #3.

- Please see our response to your comment #2 on the lack of GPP inclusion.

19. L 409-411 – Again, the authors imply increased changes with time. This was not supported, so I think the statement needs to be revisited to clarify that the warming effect is sustained through time, but there is no evidence of increases over time.

- Please see our response to your comment #5 on the temporal trends in the ER response to warming.

20. Page 25 L 37-46 – (1) How often were the measurements performed at sites? Do the authors account for sampling intervals? There is evidence that manual sporadic measurements may not properly represent climate change effects in field experiments (Vicca et al. 2014). S More information is needed to understand data collection and data aggregation in this section.

- We refer the reviewer to **Supp. Fig. 2** as well as **Fig. 1** and **Supp. Table 1** where we already presented the frequency and number of observations for every dataset. We specifically refer to these figures and table in the section that the reviewer refers to (Methods L 43-44). We have also added the number of day-observations used to calculate the mean growing season ER Hedges SMD per dataset now to **Supp. Table 1** (reflecting number of ‘dots’ or data points per dataset in **Supp. Fig. 2**).
- We did not take into account clustering of ER measurements throughout the growing season in our models, because we believe **our meta-analysis approach sufficiently accounts for such methodological differences across the different studies** (i.e. by including a crossed and nested random effects structure and by including a temporal autocorrelation structure, L 185-198 in Methods).

(2) Finally, L 44-46 show that these measurements may not be ER as they do not include RA during day.

- The reviewer refers here to the specific case of ALA_1, the only experiment where automated chamber measurements are used. This experiment uses clear or ‘opaque’ chambers for NEE and then partitions those data into ER and GPP values by using nighttime ER values (defined as PAR<5) and extrapolating those to daytime conditions⁸⁻¹⁰. We decided to only use the nighttime ER values in our study to avoid using extrapolated or ‘modelled’ ER data from one experiment vs. actual measured ER data from all other experiments. Importantly, ALA_1 data contributors have investigated the difference in respiration for nighttime vs. daytime dark chamber measurements, and found the expected relationship between night and day ER.
- Further, in a robustness analysis, our meta-analysis results remained consistent whether ALA_1 was included or excluded. Therefore, we believe that the specific PAR cutoff used by ALA_1 to assess measured fluxes as ER did not influence our findings.

21. Supplementary Figure 7 – This should be Figure 6 in the main text to show the large uncertainty associated with the approach.

- Agreed, that we should show the uncertainty in Figure 6. We revised Figure 6 and show now the uncertainty map in it. Please see our response to your comment #7 for more details.

22. Supplementary Figure 9 – This is an example that the few data points used for parameterizing the model do not represent the statistical space for the predictions. Therefore, the upscaling is based on a very limited sample size that does not represent the statistical parameter space for predictions.

- We agree with the reviewer that observations do not cover the entire heterogeneity of the arctic (I.e. statistical parameter space for predictions). We now also updated this figure by adding a convex hull bounding the range of soil properties observed in our meta-analysis database, showing that this covers ca 64% of the corresponding variation in the soil database for the study area. This shows that while the observations cover most of the variability in soil conditions, the measurements should be expanded to areas to cover better the heterogeneity in the arctic – particularly areas with low C:N-ratio and high TN as well as low C:N ratio and low TN.

Revised **Supp. Fig. 9**

References (Referee #1)

1. Subke, Jens-Arne, and Michael Bahn. 2010. "On the 'Temperature Sensitivity' of Soil Respiration: Can We Use the Immeasurable to Predict the Unknown?" *Soil Biology & Biochemistry* 42 (9): 1653–56.
2. Vicca et al. 2014. "Can Current Moisture Responses Predict Soil CO₂ Efflux under Altered Precipitation Regimes? A Synthesis of Manipulation Experiments." *Biogeosciences* 11 (11): 2991–3013.

Referee #2 Jessica Gurevitch

1. The authors have addressed my comments in a thorough and very thoughtful manner. The analyses are carefully justified, robust and carried out and documented fully. The study not only provides a very important summary of critical ecosystem responses to warming using an enormous data set, but also is a model for how to do complex ecological meta-analyses. My one remaining comment is to suggest that in Figure 2 the grand mean be made to stand out more, with a larger symbol and a different color. That little diamond is almost lost in that sea of data! I should have signed my original review (I was Referee 2). It is a bit overwhelming and enormously satisfying to see how far you have taken these analyses beyond what I introduced 22 years ago (Rustad et al. 2001).

- We thank the reviewer for their positive feedback. We are happy to hear that the reviewer evaluates our analyses as '*carefully justified and robust*'.
- We adjusted the size and color of the grand mean in **Figure 2** in the Main text, to stand out better against the individual dataset effect sizes to the right of it.

Referee #3 Ivan Janssens

1. Having read the response letter and the new version, I now have a better understanding of the study. Most of my methodological issues have been well addressed, but I do have two remaining issues.

- We appreciate this comment, and are happy to hear that the (first) revisions made our study more clear to the reviewer. We hope we adequately addressed these remaining concerns, which we agree with overall, in our revised ms.

2. The first is the extrapolation exercise, which still does not convince me and is in my view also not necessary. The hotspot areas where the largest warming-induced increases in RE are expected are vastly understudied, rendering the global driver data that underly the map hugely uncertain in those areas, and there are also hardly any empirical data available to corroborate these results. I would suggest to remove the map or at least state more explicitly that the driver maps are highly uncertain in most areas and that empirical evidence is now needed to verify the existence of the modelled hotspots.

- We acknowledge the concern of the reviewer, and similarly of reviewer #1 (comment #7). We decided not to remove the upscaling exercise from the main text, because we believe it provides insight in the overall impact and spatial sensitivity of the tundra ER response to warming, and readers can relate to this by looking at the location that they are working in. We do acknowledge the large uncertainty in our upscaling, originating largely due to high uncertainty in input data (see below), and argue that it is important to show that with the current data availability the uncertainty is high in the arctic. We now include the uncertainty map to **Fig 6** in the main text, that highlights the high uncertainty behind the upscaling calculations.
- To address the reviewer's comment on the uncertainty originating from the input data, we performed an extra analysis to estimate this where we estimated the proportion of the uncertainty originating from the input data (gridded soil data) compared to the uncertainty originating from the meta-regression upscaling model (see revised **Fig. 6**). We show that in almost the entire study area, most of the uncertainty originates from the input data (revised **Figure 6c**)– in average around 63% over entire study area. Finally, we mention on L 366-367 that this high uncertainty reflects for us that spatial sampling gaps across the tundra biome should urgently be addressed in follow-up studies. In mentioning this, we immediately show the high uncertainty of the upscaling, as well as explain what the implications might be, and how our exercise has brought this forward.

3. The second issue relates to the message that the end of the abstract is conveying. '... our findings imply a potentially strong, positive climate feedback from the tundra during the coming decades, even under modest warming scenarios. Since persistently enhanced respiration with warming is likely to free up more carbon than can be compensated for by increased plant productivity, the tundra biome may accelerate, rather than mitigate the impact of global warming'. In the response letter the authors state that they are aware that this study does not provide evidence for the climate-carbon feedback, but with the abstract ending like this and the knowledge that most journalists will not thoroughly read the manuscript and will ignore the 'potentially' and 'may', this is exactly how it will be received by the public. I suggest to add a clause like: 'if not accompanied by increased carbon inputs by vegetation' or 'further research is needed to study whether the increased respiration is actually resulting in carbon loss from these tundra ecosystems'. This is much clearer than the conditional phrasing in the current abstract.

- Thank you for these thoughts, and the textual suggestion. We have rephrased the end of the abstract accordingly, and explicitly acknowledge in our revised ms that we only focus on ER (L 101, 185, 401, 432), so cannot state implications for the whole carbon cycle.

Response letter #2022-11-18995

Environmental drivers of ~~persistently~~ increased ecosystem respiration with warming in the tundra

4. Overall, I congratulate the authors with this interesting study.

- Many thanks for this nice comment.

References

1. Luan, J., Liu, S., Wang, J. & Zhu, X. Factors Affecting Spatial Variation of Annual Apparent Q10 of Soil Respiration in Two Warm Temperate Forests. *PLoS One* 8, 1–8 (2013).
 2. Wu, Q., Ye, R., Bridgham, S. D. & Jin, Q. Limitations of the Q10 Coefficient for Quantifying Temperature Sensitivity of Anaerobic Organic Matter Decomposition: A Modeling Based Assessment. *J. Geophys. Res. Biogeosciences* 126, 1–18 (2021).
 3. Davidson, E. A. & Janssens, I. A. Temperature sensitivity of soil carbon decomposition and feedbacks to climate change. *Nature* 440, 165–173 (2006).
 4. Perkins, D. M. et al. Consistent temperature dependence of respiration across ecosystems contrasting in thermal history. *Glob. Chang. Biol.* 18, 1300–1311 (2012).
 5. Melillo, J. M. et al. Long-term pattern and magnitude of soil carbon feedback to the climate system in a warming world. *Science* (80-.). 358, 101–105 (2017).
 6. Melillo, J. M. et al. Soil warming and carbon-cycle feedbacks to the climate system. *Science* (80-.). 298, 2173–2176 (2002).
 7. Metcalfe, D. B. et al. Patchy field sampling biases understanding of climate change impacts across the Arctic. *Nat. Ecol. Evol.* 2, 1443–1448 (2018).
 8. Mauritz, M. et al. Nonlinear CO₂ flux response to 7 years of experimentally induced permafrost thaw. *Glob. Chang. Biol.* 23, 3646–3666 (2017).
 9. Rodenhizer, H. et al. Abrupt permafrost thaw drives spatially heterogeneous soil moisture and carbon dioxide fluxes in upland tundra. *Glob. Chang. Biol.* 6286–6302 (2023) doi:10.1111/gcb.16936.
 10. Natali, S. M. et al. Effects of experimental warming of air, soil and permafrost on carbon balance in Alaskan tundra. *Glob. Chang. Biol.* 17, 1394–1407 (2011).
-

Response letter 1 - July 2023

Referee #1

1. This manuscript analyzes a new dataset of warming experiments across tundra ecosystems. This important effort compiled information from 56 open-top-chamber in situ warming experiments. The authors performed meta-analysis techniques and proposed that a rise in air temperature (~1.4oC) causes an increase in growing-season ecosystem respiration by 30% that persisted for at least 25 years. These results build on previous studies that have analyzed a selected number of experiments; therefore, this manuscript expands those analyses and updates and improves the estimates and observed ecosystem responses. The authors also identified potential relationships of ER with other soil physical factors and vegetation community composition. Then the authors performed an upscaling approach to present the global sensitivity of warming in tundra ecosystems. The authors compiled an important dataset, and the results are provocative. That said, there are some weaknesses in the strength of the results, and there is missing information on the underlying assumptions for some calculations. Therefore, the authors should revise the use of statistics and how uncertainties are propagated, especially for the upscaling approach. My main concern is that the statement of persistency of high ER needs to be better supported by the analysis and thus requires revision. Here I include a series of comments to improve the manuscript.

- We thank Reviewer #1 for the constructive comments. We appreciate their enthusiasm and the thoroughness of the review. We believe our revised manuscript addresses the remarks that were raised, as outlined below. Overall, we addressed the reviewer's comments as follows:
 - 'there is missing information on the underlying assumptions for some calculations'; 'should revise the use of statistics and how uncertainties are propagated, especially for the upscaling approach'
 - We thoroughly revised our analyses, including the Methods, Extended Data, and Supplementary Information, to explain the methodologies used throughout our manuscript better. We added extra tables and figures to address the uncertainties mentioned by the reviewer. Specifically, for the **meta-regression models** evaluating relationships between environmental drivers and ER response rates, we performed a sensitivity analysis to address the uncertainty regarding the appropriate sample size to use (**Supp. Discussion #4, Supp. Fig. 4**). For the upscaling, we fully revised the work flow and performed a detailed uncertainty analysis as described in the **Methods** and **Supp. Methods #3**. See our reply to your comment #5 for details.
 - For the upscaling, we also specifically explored how well the distribution of our dataset compares to the distribution of gridded soil data used for upscaling. To do that, we created a scatter plot for TN and C:N-ratio where we overlaid both the observed (from our dataset) and gridded values (gridded soil dataset used for upscaling (new **Supplementary Fig. 9**). It shows that the observed values from our dataset are slightly biased towards low TN values, but otherwise the observed values cover the main concentration of TN-C:N-ratio distribution rather well.

- Based on the above changes, we believe we now sufficiently describe and take into account the uncertainties throughout our different analyses, and trust these results to be robust.
- ‘statement of persistency of high ER needs to be better supported by the analysis and thus requires revision’
 - We agree that the evaluation of temporal patterns needed improvements, incl. a streamlined approach throughout the manuscript to assess the different aspects of temporal patterns. We therefore thoroughly revised how we analyze the ER response over time (i.e. how it changes with experimental warming duration). We believe we now adequately demonstrate the persistency of the ER response with the new **Ext. Table 5a** and new **Figure 3**. Additionally, addressing similar concerns by Reviewer #2, we looked at within-experiment temporal patterns, reporting these results in **Supp. Discussion #6** and **Supp. Fig. #5**. We do not present the within-experiment patterns in the main text however, and we think they should be interpreted with caution, because of the reasons explained in **Supp. Discussion #6** (in short, low sample sizes within age classes, and lack of sufficiently long within-experiment time series across age classes).

Major Comments

2. Line 148 clearly explains the overall objective of the manuscript. That said, it would be important to include testable hypotheses. L 103-146 provide a clear introduction to the problem and the knowledge gaps. Then, what are the hypotheses associated with the overall objective based on our current understanding? Are the results consistent with current expectations, or are they surprisingly different from what was expected?
 - We agree with the reviewer that adding testable hypotheses improves the storyline of our manuscript. We included the following specific hypotheses now on L (L.) 153 in the section *Objectives and study design*:
 - *We hypothesize that (i) there is an overall positive ER response to warming (both R_a and R_h), but the response magnitude varies across time and space. We further hypothesize that differences in the magnitude are driven (ii) indirectly by variation in warming-induced changes in local environmental conditions, as well as by (iii) context-dependent variation in environmental conditions.*
3. L 197-200 – It is unclear where the value of 30% more ER comes from Fig. 2. This figure has six panels with positive and negative slopes. Clarification is needed to better explain where this number comes from, as it is a primary result of the manuscript.
 - The referenced figure on the mean 30% change of ER with warming was former **Ext. Figure 2** (see below this comment), where we show the ER Hedges Standardized Mean Difference (SMD) response and 95% confidence intervals, averaged across all 136 datasets (left: MEAN), and for each individual dataset. We suspect that the reviewer was instead looking at former **Figure 2** in the main text (see below), when referring to the six panels with positive and negative slopes.

Environmental drivers of ~~persistently~~ increased ecosystem respiration with warming in the tundra

- We agree with the reviewer however that more clarification was needed on where the 30% change in ER came from. This number was calculated from the log-transformed Ratio Of Means (ROM) of 0.26, i.e. the secondary effect size used for our meta-analysis. Since the figure showed the results based on the primary effect size (Hedges SMD), the percentage change could not be derived from it. We have clarified this now as follows:
 - The Methods (section *Meta-analysis*) includes an explanation of how and why we calculate a mean percentage change value from the mean ROM as secondary effect size. L. 49 reads: “We used *Hedges’ g Standardized Mean Difference [SMD] as primary effect size across all models, and the log-transformed Ratio of Means [ROM] as additional effect size to quantify percentage change in ER with warming. ... The ROM was calculated as [mean growing season average ER in warmed plots / mean growing season average ER in control plots] which we transformed to a mean percentage increase as $100*(exp(ROM)-1)$, i.e. a more intuitive measure of changes in ER.*”. Please note that ROM and $lnRR$ (log response ratio) are used interchangeably in the literature and refer to the same effect size.
 - The new **Ext. Table 4** explains how mean percentage change is calculated from the ROM: “*Percentage change = $100*(exp(ROM)-1)$* ”. When referring to the mean 30% change in the main text, we refer to this table to avoid confusion. **Ext. Table 1** now also provides all ROM and mean percentage change values per dataset, in addition to the Hedges SMD.
 - We include what was formerly Extended Figure 2 as **Fig. 2** in the main text to demonstrate our mean meta-analysis estimate (i.e. ER Hedges SMD), as well as the variability in the ER response across experiments and datasets. This demonstrates the answer to hypothesis number (i), i.e. we find an overall increase in ER with warming (mean Hedges = 0.57 ~ mean ROM = 0.26 ~ mean percentage change = 30%, **Ext. Table 4**), but the ER response varies across space and time. We believe **Fig. 2** also helps familiarize readers with the structure of the data, i.e. the different studies and years, that were used throughout our analyses.

Former Extended Figure 2

46 **Figure 2:** Effects of experimental open-top-chamber (OTC) warming on ecosystem respiration (ER) response rates.

47 Experimental warming increased ecosystem respiration across the tundra biome but there was significant heterogeneity across the datasets.
 48 Effect of OTC warming on ecosystem respiration (ER) response rates (Hedges standardized mean differences or SMD calculated as $[mean\ ER\ of\ the\ warmed\ plots - mean\ ER\ of\ the\ control\ plots] / pooled\ standard\ deviation$) across the 136 datasets (i.e. *experiment x flux year* combinations). With a black diamond, the Mean estimate ('est') of the response rate (pooled Hedges SMD) and
 49 / pooled standard deviation) across the 136 datasets (i.e. *experiment x flux year* combinations). With a black diamond, the Mean estimate ('est') of the response rate (pooled Hedges SMD) and
 50 95% confidence intervals (CIs, error bars) across the 136 datasets are provided as well as the Q-value ('Q') for heterogeneity, and p-value ('p-val') from the meta-analysis. Black dots represent

Former Figure 2

7 Figure 2: Warming-induced changes in soil conditions and vegetation community drive respiration response.

4. Line 201 - Q10 value needs to include uncertainty; ideally as 95%CI

- We included an uncertainty interval on L. 212. We also included our calculations for the Q10 values in **Supp. Methods #2**. Please note that we discovered an error in the formula used of Q10, so the results have changed in comparison to the previous version of the manuscript. The new Q10 values did not change anything in the interpretation of the results.

5. L 204-209 – I think the authors need to be careful about interpreting the p-value of this trend. There has been a long history of discussing how the scientific community uses this information (i.e., p-values) and problems with “p-hacking.” The challenge with this study is that the authors interpret a “marginally significant” slope for ER response to warming across all 25 years as a weak positive trend. This positive trend is then interpreted and discussed in the manuscript as an essential effect despite a p-value of 0.049. In contrast, the supplementary materials show that the authors made arbitrary decisions to exclude potential confounding variables with “marginally significant” effects (e.g., flux measurement methodology; or L 149-152 in Supplementary Discussion). Furthermore, supplementary figure 5 shows the classical example that a combination of different slopes (e.g., negative and positive trends) is not the same as when data are plotted together. One can envision different fits (e.g., polynomial), piecewise, or arguably no trend. Finally, Figure 3e shows that the CI of the 3 data points for experiments of 25-30 years do not show clear evidence of increasing ER in these long-term experiments, although this could be an artifact of the low sample size. Thus, the implications of these results depend on how data is fitted and interpreted.

The authors must consider the scientific and ethical interpretation of a marginally significant effect for a potential warming trend, especially for a publication under review in Nature. Is it too early to provide strong evidence for this trend? Is the interpretation scientifically sound, and is it ethically correct to move forward with such an interpretation? L 209-212 state that despite this weak trend, the authors speculate that ER responses to experimental warming will not decrease in the longer term (i.e., multi-decadal duration). This is a statement that the authors and Nature editors need to evaluate, considering the evidence provided and how “marginal results” are interpreted throughout the manuscript.

- As the reviewer suggests, we acknowledge the need to consider a scientific, ethical, and at the same time consistent interpretation of effect ‘sizes’ (~p-values) across our regression models. We agree that there was a lack of consistency in our initial manuscript regarding how we interpreted the various models results, e.g. regression models on environmental drivers vs. temporal patterns vs. methodological drivers. For instance, we had indeed not taken into

account the methodological marginally significant driver ($p < 0.1$) but *did* report and interpret a marginally significant effect of warming duration on the ER response. The revisions of the manuscript address these points now by consistently interpreting significant results as $p < 0.05$. We respond to the different points raised by the reviewer here as follows:

- A) ‘L 204-209 – I think the authors need to be careful about interpreting the p-value of this trend. There has been a long history of discussing how the scientific community uses this information (i.e., p-values) and problems with “p-hacking.” The challenge with this study is that the authors interpret a “marginally significant” slope for ER response to warming across all 25 years as a weak positive trend. This positive trend is then interpreted and discussed in the manuscript as an essential effect despite a p-value of 0.049. In contrast, the supplementary materials show that the authors made arbitrary decisions to exclude potential confounding variables with “marginally significant” effects (e.g., flux measurement methodology; or L 149-152 in Supplementary Discussion).’
- We acknowledge the concerns in terms of consistency in statistical interpretation of the meta-regression effects throughout the manuscript. In our revised version, we streamlined the interpretation of significant effects, specifically mentioning on L. 188 in the Methods that “*We considered and interpreted results as significant in our study if $p < 0.05$, as is commonly done in meta-analyses*” As mentioned, this threshold is commonly used in meta-analyses^{11–13} and deemed appropriate for our purposes. We no longer interpret marginally significant effects throughout the manuscript (**Fig. 3-5, Ext. Table 3-5**). Hence, our interpretation and conclusion in the main text is only based on effects with p-value < 0.05 . Moreover, our sensitivity analysis using different inclusion criteria for environmental data showed that the results were robust (see **Supplementary Discussion #4**).
 - During the revisions, we addressed the specific examples of (marginally) significant effects mentioned by the reviewer as follows:
 - Duration: we thoroughly revised our analysis of temporal patterns, using meta-regression instead of linear regression models. This led to the ‘*weak positive trend in ER response across all 25 years ($p < 0.049$)*’ disappearing, and instead we find no significant trend across all years (*across* age classes, **Ext. Table 5a**). Further, the temporal patterns *within* age classes now clearly show a non-linear trend over time. Taking together all our results from the assessment of temporal patterns, we believe our findings now appropriately justify our statement on persistency of the ER increase with warming over time (L. 230 of main text, **Ext. Table 5a, Fig. 3**). After these revisions, our earlier conclusion of a persistent positive effect of ER with warming and no indication of waning in the long term, still remains solid.
 - Machine Type: we no longer interpret, but do report the ‘marginally significant’ (p-value $0.05 < p < 0.1$) methodological drivers of ER responses as *trends* in **Supp. Fig. 3**. We explain the reasoning for, and methodology behind this supplementary analysis in **Supp. Discussion #2** in detail.
- B) ‘Furthermore, supplementary figure 5 shows the classical example that a combination of different slopes (e.g., negative and positive trends) is not the same as when data are plotted

together. One can envision different fits (e.g., polynomial), piecewise, or arguably no trend. Finally, Figure 3e shows that the CI of the 3 data points for experiments of 25-30 years do not show clear evidence of increasing ER in these long-term experiments, although this could be an artifact of the low sample size. Thus, the implications of these results depend on how data is fitted and interpreted.'

- We acknowledge that these results on temporal patterns were not clear enough yet, and therefore thoroughly revised these. We clarify and visualize the new analyses and results on L. 222-233 in the main text, in **Fig. 3** and **Ext. Table 5a**. Based on these revisions, we believe our statement on persistence over time still holds. Our new approach uses meta-regression models investigating effects of warming duration across and within age classes. The specific findings that support the persistency of the ER response over time are as follows (highlighted in **bold**):

“The mean ER response remained positive and did not differ significantly across the four age classes (Fig. 3a), hence the positive warming effect persisted over time. Investigating the temporal patterns further showed indications for non-linear, positive responses over time (Fig. 3a, b, Ext. Table 5a): a decrease in the magnitude of the positive ER response during 5-9 years of warming ($Q_m=63$, $p<0.001$, $N=28$, Fig. 3b) was followed by an increasing magnitude during 10-14 years of warming ($Q_m=5.6$, $p<0.05$, $N=15$, Fig. 3b). Combined with the lack of change in the magnitude of the ER response over time during 0-5 years and >15 years of warming, the overall slope across all years was non-significant, indicating the persistency of the positive ER response to warming (Fig. 3a, Ext. Table 5a). While the ER response to experimental warming may thus falter around the end of the first decade of warming, our data do not provide evidence that ER responses to experimental warming wane in the longer term (i.e., multi-decadal duration).”

- The separation into four age classes and investigating effects of duration within/across age classes (**Fig. 3, Ext. Table 5a**) proved a suitable approach to look at possible non-linear temporal patterns. It allowed us to deal with the high errors (large CI) for longer-term experiments (as the reviewer points out). Namely, by grouping all datasets with duration of warming greater than 15 years, we have a sufficiently balanced number of datasets across each age class, for our model results to be robust (**Fig. 3**: $N=70$, 28, 15, 23 for age class 1, 2, 3, 4).
- We assume the reviewer also referred to possible differences in trends when plotting/modelling across vs. within experiments. Please see our response to comment #5 from Reviewer #2 for this, which we interpreted as a similar comment.

C) ‘The authors must consider the scientific and ethical interpretation of a marginally significant effect for a potential warming trend, especially for a publication under review in Nature. Is it too early to provide strong evidence for this trend? Is the interpretation scientifically sound, and is it ethically correct to move forward with such an interpretation? L 209-212 state that despite this weak trend, the authors speculate that ER responses to experimental warming will not decrease in the longer term (i.e., multi-decadal duration). This is a statement that the authors and Nature editors need to evaluate, considering the evidence provided and how “marginal results” are interpreted throughout the manuscript.’

- See our response to part B) for a detailed explanation on this. We now improved the temporal analysis and thank the reviewer for pointing this out. We trust that our revised analyses and results now clearly demonstrate persistency in the response of ER to warming. That is, all age classes have a mean positive ER response. However, we observed differences in the relation between age and ER response in each age class. We made this more clear in the main text (L. 224-231). An apparent faltering in the response between 5-9 years of warming, followed by an increase in the response magnitude between 10-14 years. See **Fig. 3, Ext. Table 5a** and interpretation of these results on L. 377-393.
6. Upscaling of ER due to increase in temperature. This is an add-on to the manuscript with multiple underlying assumptions that must be explicitly listed in the Methods. Furthermore, there are issues of uncertainty derived from the meta-analysis (i.e., uncertainty in the metaregression) and with the datasets used as covariates (i.e., uncertainty in the original estimation of the covariates, and then the uncertainty in upscaling from the native resolution of these datasets [e.g., from 1x1 km resolution to 5x5 km resolution]). I see the appeal of doing this exercise. Still, at this moment, there are many missing details for this analysis which arguably results in a “back of the envelope” calculation as is currently presented. The authors must clearly explain the Methods and the assumptions and think about error propagation in the input variables and model. Finally, think about uncertainty beyond that calculated from the metaregression (which I think is presented in supplementary figure 7).
- We fully revised the upscaling workflow, addressing not only the uncertainty from the meta-regression model but also from the ISRIC dataset and from using a proxy for C:N-ratio. The revised method is explained in detail in the Methods (*Spatial upscaling*) and summarised in a flow diagram in **Supp. Methods #3**. The main changes to the method are:
 - We now use the native resolution (250 m) soil data, for which ISRIC provides uncertainty estimates (which are not yet available for other resolution (i.e. 1 km and 5 km products). This allowed us to estimate the uncertainty originating from the ISRIC data. Due to the large study area (10 million km²), we still needed to aggregate the soil data to 1 km resolution as even with this resolution we needed a supercomputer to run the uncertainty analysis.
 - We used the reported C:N-ratio and reported TN and TC from our database (**Supp. Fig 8**) to estimate the uncertainty originating of the proxy using TN and SOC to derive the C:N-ratio, and incorporated this uncertainty in our Monte Carlo uncertainty analysis.
 - In the Monte Carlo uncertainty analysis, we created 100 pairs of TN and C:N-ratio values for each 1 km² grid cell using a truncated multivariate random normal distribution, using the mean and standard deviation of these variables as provided by ISRIC.
 - We provide information about the degree of extrapolation of the variables in our upscaling model (**Supp. Fig. 9**).
 - We then used the resulting 100 pairs of TN and C:N-ratios as input for our ROM model for each 1 km² grid cell, resulting in 100 predicted %ER changes with associated standard errors. These were then combined to get the mean predicted percentage change in ER induced by 1.4°C warming and a combined measure of standard deviation.
 - We have revised our upscaling, so that it is more nuanced, and is meant to give an idea of the magnitude of ER respiration increases across different tundra areas. On L. 348, it now reads: “*Although these numbers are crude estimates, they suggest an order of*

magnitude of ER respiration increase that can be expected for the tundra as a whole, and clearly show spatial differences in sensitivity.”

7. ‘The website <http://arcticflux.org/> describes the goal, the team, and the locations. I strongly support the commitment by the authors to make the data and scripts available on Figshare or other data FAIR repositories.
 - We agree on the importance of sharing data and scripts on global, common repositories. As mentioned at the time of submission, we plan to make all data, code, and figures available on Figshare upon publication. During the review process, they can be accessed through the link to our shared folder. See the *Data and Code availability* statements for details.
 - Please note that we considerably revised our database website, including also a section on specific projects performed with the database, where we plan to describe where the data, code, and figures from each project publication can be found. The updated website location is <http://www.tundrafluxdatabase.com>.

Detailed comments

8. Supplementary Figure 4 – Are all these slopes statistically significant? One plot seems to have only 3 points.
 - These figures (now thoroughly updated) showed a selection of effect plots from our meta-regression models testing the influence of indirect warming effects and context-dependencies. With these figures, we aimed to show a specific aspect of our data, namely that for some environmental drivers (soil, vegetation, microbial community), the same data (same colour) could be linked to more than one ER data point from different years. We realize this was not clear, and not sufficiently accounted for in our previous analyses. Please see **Supp. Discussion #4** where we describe this data aspect in more detail now, and explain how we assessed the potential implications of this for our results.
 - To respond to the specific question raised by the reviewer: not all these slopes were statistically significant, and the regression L and significance levels themselves were actually not relevant for what we wanted to show. That is, that there was no apparent influence on the meta-regression results due to the fact that measurements for several environmental drivers may have (i) differed in exact years from the year in which the ER measurements were taken, and (ii) been replicated for multiple ER data points. We believe that our inclusion of **Supp. Discussion #4**, describing these aspects in detail, combined with a thorough sensitivity analysis of which results are shown in **Supp. Figure 4**, present this better now in the revised manuscript. In sum, we believe that results of our sensitivity analysis show that the significant results based on the full dataset model are robust and valid to present in our main text.
9. Line 231- arbuscular mycorrhizae is a type of mycorrhizae. Do you mean ectomycorrhizae?
 - Since we decided not to interpret results that had a p-value between 0.05-0.1 in the revised manuscript, this statement was removed (**Ext. Table 5b**). However, to respond to the reviewer’s question, we were indeed highlighting the specific type of mycorrhizae i.e. arbuscular mycorrhizae or AM, which typically associate with graminoids.

10. Line 254 – The effect of total nitrogen is also “marginally significant”? Then how does this borderline effect influence the interpretation in L 257-262?
- In our revised manuscript, additional gap-filling based on suggestions from Reviewer #3 led to the effect of TN becoming clearly significant with p -value = 0.01*, making this comment no longer an issue. See the specific changes in the results below.
 - submitted results: Ext. Table 4: regression estimate = -0.63; [-1.24, -0.01]; p -val = 0.046*; $Q_m=4.0$; number of datasets=36
 - revised results: Ext. Table 5c, Fig. 5a: regression estimate = -0.59; [-1.06, -0.13]; **p -val = 0.01***; $Q_m=6.3$; number of datasets=43
 - Thus, we remain convinced about our statement on L. 283 reading “*Greater ER increases with warming in nutrient-poor sites may be linked to changes in stimulated belowground C allocation by the plants and subsequent soil priming by root leachates.*”
11. L 290-292 – This is not a surprising result as it is well-known that local biophysical factors influence the responses of ecosystems. If this is a novel result, the authors must prove that previous knowledge contradicts this. Or rephrase the statement by saying that the meta-analysis confirms expectations of how local soil conditions influence ecosystem responses.
- We revised our statement on L. 318 so that it reads as a confirmation of our hypothesis (ii) and (iii), specifically for soil conditions, instead of a result that was unsurprising or already known.
12. L 297-301 – These statements are interesting, but the authors should consider the strength of the results, especially in light of Total Nitrogen.
- See our response to your comment #9 for this. The strength of the result for total nitrogen increased because of gap filling for the revised manuscript, so we believe these statements are still, and even more so, valid now.
13. L 302 – I find the statement “our new understanding of context-dependencies of warming effects” unclear. What exactly does this mean, and what is new in further applying the results? Something seems missing in understanding what the authors imply and how they used the information/data to move to the following research step. I think the information is in L 318-325.
- We rephrased the statement on L. 336 to clarify which context-dependencies we refer to and how they were used to do our spatial upscaling.
14. L 328-329 – I am not convinced about this statement because of the challenges and limitations in the data analysis. The interpretation is likely, but the results are not 100% convincing. It is more likely that non-linearities are the rule rather than a straightforward linear increase, as speculated by the authors. This is discussed in L 334-342.
- We believe our revised statement that “*we found sustained ER increases across 25 years of experimental warming*” is correct. When we say ‘sustained’, we are not speculating that the trend over time is linearly increasing, although the phrasing from the initial manuscript: ‘weak positive trend’ might have been what the reviewer pointed at here. We rephrased this section (L. 222-233), and highlight in the revised manuscript that:

Environmental drivers of ~~persistently~~ increased ecosystem respiration with warming in the tundra

- (i) ER responses were significantly positive (above zero) for each of the four duration age classes (**Ext. Table 5a, Fig. 3a**), implying that an increased ER due to warming is persistent across 25 years of warming duration.
- (ii) There is no significant linear effect of experimental warming duration across the 25 years (**Ext. Table 5a**).
- (iii) A decreasing trend in ER response rate to warming between 5-9 years of warming is compensated by an increasing trend between 10-14 years (**Ext. Table 5b, Fig. 3b**). This third result relates to the specific point on non-linearities in the temporal trend mentioned by the reviewer.
- Overall, our data suggests that the ER response to warming does not diminish in the longer term across at least 2 decades of warming. We believe we clearly demonstrate now the effects of duration on ER response in a clear and consistent way (**Ext. Table 5b and Fig. 3b**).

15. Line 344 – this is an important statement, but the challenge is identifying the sites/areas.

- We revised this statement as a whole on L. 367, whilst revising the final paragraphs of the manuscript based on our updated results. The initial statement did not sufficiently bring forward that we actually attempted to increase this mechanistic understanding with our own results, i.e. the meta-regression results and spatial upscaling. With meta-regression, we were able to investigate the causes for spatial variability in the ER response to warming, and with the spatial upscaling, we were able to assess, albeit roughly, which sites would be more prone to warming-induced ER increases than others. The reviewer is right in stating that the challenge is in identifying the sites/areas more prone than others, but this is exactly what we aimed to demonstrate with our upscaling results. Namely, experimental field data was used to improve our mechanistic understanding on spatial sensitivity. This understanding can be used to improve regional climate models for the tundra biome.

16. Line 354-356 – I fully agree with this statement, but I feel it is a weak way to end the manuscript as it is not a novel call or conclusion.

- As part of reworking the final paragraph, this statement is now moved upwards on L. 363, such that the final paragraph is stronger and highlights the impact of our findings.

17. L 38-43 – I assume these measurements were with “dark chambers.” Please clarify.

- Indeed, all experiments except for one used dark or opaque chambers to measure ER. For experiment ‘ALA_1’, which had automated chambers, all measurements were done with clear chambers, since switching chambers on an automated system would require a lot more infrastructure. ALA_1 then distinguished between ER and NEE measurements based on photosynthetically active radiation (PAR) measurements. Flux measurements when PAR was smaller than $5 \mu\text{mol m}^{-2} \text{s}^{-1}$ were considered nighttime or ER measurements. We explain this now on L. 43-46 in the Methods. See also **Supp. Discussion #2** for how we checked whether this difference in chamber measurements (i.e. different ‘timing’ of ER measurements) influenced our results. We found no apparent influence of the timing on the ER response.

Referee #2

1. This paper reports on critically important and novel results from a large number of arctic and alpine experimental tundra sites covering a very wide geographic area. Quantifying the responses of soil respiration in the tundra is of great interest to a wide audience, and using detailed and geographically extensive experimental data to do so, as is done here, is vastly superior to other approaches. The results here are therefore of the highest interest and importance, and certainly have the potential to be Nature publishable. Enormous amounts of data were collected, and the authors have analyzed the results thoughtfully. The use of Hedges' g , weighted by precision, and of accepted meta-analysis protocols, are an excellent foundation for the analyses. Nevertheless, I think more can be extracted from this impressive data set. It was difficult not to get lost in the plethora of data points, things measured, and analyses, and it is very easy (and likely) for readers to misinterpret the results. I have a number of questions and suggestions in this regard that may improve the sharpness and accuracy of the analyses.
 - We appreciate the positive feedback on our work, the valuation of our data collection effort and approach to analysing the data. We are equally enthusiastic about the dataset and apparent robustness of the statistical approach. We believe that the additional sensitivity analyses that we have done demonstrate this robustness of the Methods and results. We further hope that we have elucidated our drivers/measurements and the different steps in analysis in this revised version, such that results would not be misinterpreted but instead better understood.
 - We do want to stress that our study evaluates ecosystem respiration or ER instead of soil/microbial respiration. Soil respiration, referred in the manuscript as microbial or heterotrophic respiration is only a component of ER, while the other component that makes up ER is plant or autotrophic respiration. We clarify this key distinction in the main text on L. 110, and believe it should be clearer as well by including supplementary meta-analysis results on these two separate components in response to warming, which were done in response to Reviewer #3's comments (**Ext. Table 4b, Supp. Discussion #1**).
2. The overarching question here is, how did soil respiration in tundra respond to experimental increases in temperature, and the answer is, it increased a great deal, but how much depends on the details. Fair enough, but I think the answer can be sharpened quantitatively with somewhat different Methods, and a different graphical approach. The answer, "by a mean difference of 30%" is likely to be extrapolated and misinterpreted, and is, in fact, a very vague generalization.
 - Indeed, we agree that the 30% ER increase should be more nuanced by highlighting at the same time the significant heterogeneity in the response. We have nuanced this by (i) always including the confidence interval in addition to the mean response, and (ii) including **Fig. 2** now in the main text, which clearly shows the **variability** in the response alongside the **mean** response. (iii) mentioning that our data was based on growing season measurements. Our analyses and manuscript aim exactly at quantifying and attributing this heterogeneity in ER response to warming to carefully selected environmental drivers, so that this variation can be taken into account in e.g. climate models.
 - Thus, we believe our mean 30% increase in ER with warming can still be brought forward as a key result, as it gives an idea of the impact and strong, positive overall response across all datasets, which supports our first hypothesis (*Objectives and study design*, **Ext. Table 4a**). Still, we agree that caution is needed with such 'impact' numbers, which could be used in the wrong

way, and therefore specifically include the following statement on L. 220: “*While the ER response to warming was overall positive and strong, there was significant heterogeneity across the datasets (Q-value 731, $p < 0.001$, Fig. 2, Ext. Table 4), implying that ER responses to warming vary across time and/or space*”.

3. As the authors emphasize, the magnitude of the response is heterogeneous, and depends on context and various soil and other covariates. I did not understand how the within-area heterogeneity was distinguished statistically from the among-area heterogeneity, or how the “hot spots” of sensitivity and enhanced responses were identified; clearly this is important. How was it determined that arctic responses were larger than alpine responses, for instance, or that some areas of Siberia were particularly sensitive to warming? How was this distinguished from ambient warming that is also occurring?

- We respond to the different questions of the reviewer point per point, as they relate to different steps in our analyses. In each response, we highlight how we clarified these methodological aspects in our revised version.
- A) Distinction between Within-site heterogeneity vs. Among-area heterogeneity
 - We used within-site and among-area heterogeneity in a somewhat different way than that the reviewer assumes. We checked for consistency in wording to prevent confusion about this. In the OTC warming experiments, and subsequently in our upscaling analyses, ‘**within-site**’ (experiment) heterogeneity quantifies how warming has affected environmental drivers (e.g. local soil moisture) per experiment. This heterogeneity was quantified as effect sizes of ecological factors in warmed vs. control plots per experiment, and is what we refer to as ‘indirect warming effects’ throughout the manuscript. See **Ext. Table 5b** for an overview of these factors (SMD climate, soil, vegetation, microbial) and a revised explanation in the Methods (L. 77 and following) and **Supp. Methods #1**. Importantly, the within-site heterogeneity (as in variation between replicates) is taken into account when calculating effect sizes.
 - In contrast, the ‘**among-area**’ (across-experiments) heterogeneity reflects the variability in environmental drivers among each experiment or (larger) site, depending on the specific driver. Most of these measurements of among-area environmental conditions, were quantified based on field measurements that we obtained from the control plots. We have clarified the different levels at which these specific environmental drivers were collected in more detail now in **Supp. Methods #1**.
- B) Hotspots of sensitivity and enhanced responses. How was it determined that arctic responses were larger than alpine responses, for instance, or that some areas of Siberia were particularly sensitive to warming?
 - This question showed that the upscaling approach was not entirely clear. We hope that by extending the methodology of our upscaling (which brought forward the results the reviewer mentions), and by adding an additional flowchart that summarizes the method (**Supp. Methods #3**) better explains how these results were obtained.
- B) Influence of ambient warming
 - Please see the next comment for a detailed response on this.

4. Apparently the 30% is the mean across all years and sites (extended Figure 2), which to me did not make sense. There are several reasons I am not convinced this is the right way to summarize the results. First, the controls are warming during the duration of the experiment, by different amounts based on location, because especially arctic tundra is warming rapidly. So, it might make more sense to use the controls from the earlier studies as a baseline, if that is possible (it may not be, I understand, depending on what data is available for which sites). In any case, the simultaneous warming of the controls should be considered.

- Please see our response to your comment #3 on the 30% value.
- In response to the ambient/background warming effects, we do not believe this confounded our results because of the following.
 - First, we use ER Hedges SMD values, i.e. mean differences between ER in the warmed and control plots (divided by pooled standard deviation) for each dataset, hence our response reflects the *additional* ER for each experiment and year exceeding any potential ambient warming effect that may have occurred. We argue that, by focusing on ER response rather than absolute ER values, we overcame the variation that ambient warming potentially could have caused.
 - Second, calendar year itself did not affect the ER response (meta-regression: p-val=0.6 (non-sign), $Q_m=0.4$).
 - Third, the reviewer's suggestion to use 'controls from the earlier studies as a baseline' is interesting, but in our opinion not feasible because baseline data is lacking in most studies, and cannot be transferred across studies (**Ext. Table 1-2**). We believe an analysis of changes in baseline ER would be outside the scope of this manuscript.

5. Second, there is a mention of using years within sites in the meta-regression, which is good, but these are trends (that is, the data are sequential), so it doesn't really make complete sense. If soil respiration is not an immediate response (it isn't), but rather the response is cumulative, somehow describing the shape(s) of that response over time is critical; just describing the duration of warming isn't adequate, either, because the actual year is also important (all responses in, say, 2000, or 2010, share some similarities, as warming, CO₂, rainfall, etc. change over time ambiently). Despite the fact that this is an enormous data set, a lot of information one would want isn't available and I recognize that this can't be modeled precisely, but somehow a sharper picture should be possible to extract, and a much clearer graphical presentation than ext Fig 2 should be created and included right up front in the paper (this to me is the most important data). Perhaps a timeline with each data set shown as a line graph would be clearer? Or a multi-figure graph could be drawn with one graph for each site with responses for each year shown with a single time line, and the graphs for the sites stacked, plus one line averaged by date across all available data across sites for that date? I wanted to see how the responses changed over time (the shapes, by years) for each area (averaged across the plots in that area). How to handle that statistically is admittedly challenging, but at least showing it graphically would be valuable.

- We agree, and have streamlined and clarified the temporal analysis. In short, we specifically include **Fig. 3** (based on **Ext. Table 5a**) which presents our main results for the temporal analysis *across* experiments and include **Supp. Discussion #6** and **Supp. Fig. 5** to visualize *within*-experiment responses as well, as suggested by the reviewer.

- We want to stress, however, that we believe our main strength of our dataset and analysis, are to show *across*-experiment effects of duration/age class, which we do in **Figure 3**, in a similar way as we look at the effect size *across*-experiments in the main meta-analysis. That is, looking at the *within*-experiment effects of duration are less informative in our opinion. This is because of (i) uneven distribution of repeated measurements per experiment (see **Ext. Table 2**: Nr ER years, and the overview table below*), and (ii) lack of experiments stretching the whole duration range that we cover with the across-experiment data (0-25 years).

*Overview table:

A) Overview of the variability in time series length of ER years, i.e. Nr of ER measurement years within an experiment. Note that, although there are 2 long time series of 11 and 13 years of subsequent ER data (ALA_1, GRE_6 – Ext. Table 1-2), other experiments only provided data from between 1-4 years. Note as well that repeated measurements did not always include successive years (Ext. Table 2).	Nr of experiments	Nr of ER years
	21	1
	16	2
	9	3
	8	4
	1	11
1	13	
B) Overview of the range in warming duration age class covered by the different datasets.	Nr of datasets	Warming duration age class (years)
	70	class 1 [0-5)
	28	class 2 [5-10)
	15	class 3 [10-15)
	23	class 4 (≥ 15)

- Further, we believe that looking at across-experiment variation to analyze temporal patterns is appropriate here, and is similar to what we do with our other meta-regression models (**Ext. Table 5b-c**). This is, to our knowledge, the strength of a meta-analysis and -regression approach: to overcome variation across different studies (in our case experiments). We also wish to stress that we do not evaluate absolute ER values in our manuscript, but respiration *responses to warming (effect sizes)*, which we believe is less problematic to evaluate across different experiments than it would be for absolute ER values. The latter would, we believe, indeed be much more influenced by specific measurement times or climatic conditions than the response to warming would be.
- We do not fully agree with the following statements on supposed missing information (*‘a lot of information one would want isn’t available’*), but believe this was related to lack of clarity in the previous manuscript. We attempted to clarify these things as follows:
 - ‘If soil respiration is not an immediate response (it isn’t), but rather the response is cumulative, somehow describing the shape(s) of that response over time is critical; just describing the duration of warming isn’t adequate, either, because the actual year is also important’.
 - Indeed, describing the shape(s) of the ER response over time is highly relevant, and this is exactly what we aimed to do with our temporal analysis (**Ext. Table 5a, Fig. 3**).

Environmental drivers of ~~persistently~~ increased ecosystem respiration with warming in the tundra

- We agree that the actual calendar year can also be important, as the climate is warming over the duration of these experiments, and climatic differences will exist among the calendar years. However, we believe the actual calendar year will not have influenced the majority of our results here, because of our focus on ER response to warming, and not absolute ER over time. Nevertheless, the temporal patterns examined in the manuscript might have been partly obscured by the underlying calendar year differences, as suggested by the reviewer. Because the experiments used in the analysis cover many different starting times and measurement years, however, we believe that the calendar year will only tend to obscure rather than actually contribute to the impacts of warming duration that we have assessed.
 - ‘Perhaps a timeline with each data set shown as a line graph would be clearer? Or a multi-figure graph could be drawn with one graph for each site with responses for each year shown with a single time line, and the graphs for the sites stacked, plus one line averaged by date across all available data across sites for that date? I wanted to see how the responses changed over time (the shapes, by years) for each area (averaged across the plots in that area).’
 - We included something like this in the previous **Supp. Fig. 5**, where we showed linear regression results for individual experiments per age class, combined with a mean regression line per age class. However, it was clear from the reviewer’s comment that we needed to clarify our temporal analysis throughout the manuscript, and hope that the combination of **Fig. 3**, **Supp. Fig. 5** and **Supp. Discussion #6** clarifies the temporal results. Specifically, we looked at both *across*- and *within*-experiment temporal patterns now but believe the across-experiment trends are most informative and therefore report these in our main text.
 - Due to the uneven distribution of ER measurements across experimental age (see overview table above), combining the different datasets proved the best way to test the effect of warming duration on ER response *across* experiments (i.e. space-for-time substitution). Here lies, in our opinion, the greater strength of our datasets to analyze temporal patterns, and we highlight this, as well as the distinction between the two types of temporal responses (across vs. within experiments) now in **Supp. Discussion #6**.
6. It wasn’t completely clear to me why Hedges’ *d* was used instead of response ratio (as the results are reported as a ratio or percent), or how the “ratio of means” differs from a response ratio, or why for some estimates the “ratio of means” was used and for others Hedges’ *g* was used (response ratios and Hedges’ *g* are both perfectly acceptable, just different).
- We focus on Hedges Standardized Mean Differences (SMD) as primary effect size for the meta-analysis and meta-regression models (**Ext. Table 4-5**) and interpretation of results, because Hedges SMD takes into account the precision of the measurements by including the pooled standard deviation in the calculation. The other effect size, log-transformed Ratio of Means (ROM), does not take into account variability in the measurements, but only the means, which is less preferred because of the variability in sampling design and methodology across the datasets (**Supp. Discussion #2**, **Supp. Fig. 2**). We included ROM as a secondary effect size for the main meta-analysis (**Ext. Table 4**) and upscaling, because it allows calculating a percentage change value. Percentage change allowed a more intuitive interpretation of the results, and was

Environmental drivers of ~~persistently~~ increased ecosystem respiration with warming in the tundra

essential to do a spatial upscaling (this could not be done with Hedges SMD). See L. 49-64 of our revised Methods for explanations on these different effect sizes and why and when each of them was used throughout the analyses.

- Importantly, in evaluating the warming-induced changes in microclimate (e.g. soil temperature), soil conditions, and microbial community composition, we also assessed a third effect size, i.e. raw mean difference or RMD, besides Hedges SMD. We do this to get a better feel of how big or small the environmental and vegetation/microbial community changes were caused by the warming treatment (**Ext. Table 3**, Methods L. 139). Since this effect size does not take into account the measurement precision, we always focus on the Hedges SMD instead of RMD in the main text.
7. Using geographically estimated values for some parameters seems appropriate, but where there were other missing values, multiple imputation would provide a more accurate analysis than would omitting the sites that have missing data (it wasn't clear how missing data were handled). Multiple imputation is a relatively new approach in meta-analysis, but appears to be a substantial improvement with respect to missingness. The approaches taken to dealing with the mass of data were courageous, but the one-at-a-time approach used in places is an older one that is lacking in important ways.
- It was unclear for us whether the reviewer suggested to do imputation of the ER response or of the environmental drivers here. We decided not to apply any statistical imputation for either because of the following reasons.
 - First, to our knowledge, imputation techniques are often necessary in meta-analysis in case of missing standard deviation or sample size to compute the sampling variance of the effect size. The lack of standard deviation or sample size is commonly due to primary studies not reporting such information. Yet, in our case, the dataset was compiled from the raw data, i.e. individual ER measurements, which implies that all necessary statistics to calculate our effect sizes and sampling variance calculation were available.
 - Second, imputation of predictors (environmental driver data) used in meta-regression was not performed either because of too large gaps in (some of) the predictor data, which would, in our opinion, not yield realistic results for imputed predictors. See **Supp. Table 2** for a sample size overview for each predictor. We chose to apply a more strict and parsimonious criterion for the predictors by focusing exclusively on what was measured in the field by the data contributors, or what could be collected from databases.
 - Importantly, there was some form of 'gap filling' in our data collection already, since data contributors could submit predictor data (environmental drivers characterizing soil conditions, vegetation, and microbial community) from a certain year to link with (sometimes multiple) ER datasets from another year. Please see explanations on that, and a detailed sensitivity analysis we performed to check that this did not influence the results in **Supp. Discussion #4** and **Supp. Fig 4**.
 - For our reply to the 'one-by-one approach' for meta-regression results, please see our response to your next comment #8.

8. One better approach for dealing with the complications of direct and indirect effects that is so apparent here is SEM meta-regression, pioneered by the social science statistician Mike Cheung (Singapore Univ.). This would be a more precise and elegant approach. In some cases, using multivariate meta-analysis (not meta-regression, but using something like NMDS, PCA, or the like) would help to reduce the dimensionality of the undoubtedly correlated factors instead of the one-at-a-time approach.
 - An SEM approach would be beneficial but given the data structure in our study it is not possible to do a comprehensive SEM attempt because: First, there are large differences in sample size across the different environmental drivers (between 9 and 136 datasets: **Supp. Table 2**), and the multiple imputation gap filling that is needed for SEM models would not yield realistic results in our opinion (see our response to your comment #7). Sample size variation reflects the reality of (sampling) ecological data, as we have more measurements available from environmental drivers that are easier to measure (e.g. air temperature, soil temperature, soil moisture data with climate loggers), and less from drivers that are more difficult/time-consuming to measure (e.g. soil conditions, vegetation/microbial community). Since SEM would have to be done on the most full sample size possible, this implies that we would have to do SEM on a much lower sample size, lowering the power of our analyses.
 - Besides, a limited SEM using drivers for which all datasets had an observation would be relatively comparable to a multi-model meta-regression approach. However, we do not mention the result of this in the main text because (i) it was only possible for a rather limited, random selection of drivers (Zone, soil C stock, pH class, vegetation class and NPP), and (ii) we did not find any significant relationships between these drivers and the ER response with the multi-factor model either (cfr. single-factor model results, **Ext. Table 5c**).
 - We did not reduce dimensionality of our predictors because we deemed all chosen variables equally important for the ecological understanding of our system. See correlation plots below for an overview of the actual correlations among drivers.

Indirect warming effect drivers:

Indirect warming effect drivers:

C Vegetation community

D Microbial community

E Context-dependency drivers

9. Heterogeneity is included in places, but I^2 and Q should be reported more consistently and the method used for calculating τ^2 for the random effects models should be reported; clearly these data are highly heterogeneous. In fact, that heterogeneity is an important part of the story being told here, because it is biologically reasonable and interpretable (at least in part). Showing how accounting for the biologically meaningful differences in response reduces that statistical heterogeneity would be very valuable.

- We agree and reported the strength and the quality of the models in more detail (Ext. Table 4-5). Previously, we only reported regression estimates, Q_m and our calculated R^2 values for meta-regression models. Since we used multi-level meta-analysis and -regression models, including both a random component structure as well as an autocorrelation structure, the estimated heterogeneity in our models is partitioned across these different components¹³⁻¹⁸. Therefore, we expanded our tables with model results to also include (i) the Q -values for heterogeneity, (ii) the variance partitioning of the three-level component across σ^2_1 (for

between-experiments heterogeneity) and σ^2 (for between observation-within-experiment heterogeneity); and (iii) τ^2 and ρ values reflecting the autoregressive component in our meta-models: **Ext. Table 4-5**. We also updated the method description on how we account for heterogeneity (L. 174).

Referee #3 Ivan Janssens

1. The submitted manuscript analyzes an impressive dataset on passive warming experiments by OTCs, focusing on the environmental drivers explaining the variation of the ecosystem respiration (ER) response to the warming. The rationale for the study is that more insight is needed in the response of respiration to warming to obtain better estimates of the land carbon-climate feedback. The key finding is that across all experiments and independent of warming duration, there is a mean increase of ER of 30% for a mean warming of 1.4 °C. Experiments with lower nitrogen concentrations in the mineral soil exhibited a greater stimulation of ER by the placement of OTCs. An upscaling approach using global maps of the key drivers then revealed Western and Eastern Siberia and the Canadian Archipelago as hotspots of warming sensitivity. While this analysis of the response of ER to the OTC placement across a large number of experiments is of course interesting and novel, there are multiple issues with the current version, through which I cannot assess whether or not this study is truly relevant. I will list these issues first and thereafter provide a list of questions/comments that arose while reading the text.
 - We appreciate the reviewer's positive remarks and addressed the outspoken concerns with care below. We do want to stress that our aim was written as "*we urgently need a better quantification and greater mechanistic understanding of how warming influences ER across the tundra biome*" (L. 148). This differs from the aim to obtain *better estimates of the carbon-climate feedback* the reviewer mentioned. This is an important distinction, related as well to the reviewer's comment #3 on lack of GPP data since it was not available. With our dataset on environmental drivers of ER in response to warming, the land carbon-climate feedback is hard to access, but we believe we *can* assess one of the key components of that feedback, i.e. respiration by plants and microbes. We believe this is novel because no synthesis study of respiration changes with warming, and based on field data, has been done for the tundra biome at this scale before. We do describe potentially strong *implications for* (but do not claim estimations of) the land-carbon climate feedback. However, including NPP and GPP is outside the scope of our work here.

Major comments

2. This manuscript deals with warming responses, using the increase in air temperature induced by OTC placement. However, air temperature would only affect aboveground respiration (roughly 20-40% of ER) and not the part that is key to the land carbon-climate feedback. How did soil temperatures change? I was very surprised that Figure 1b, nor any of the Tables give this soil warming data. The majority of experiment placed the OTCs only in the growing season. Because of the passive warming and the vegetation cover in summer, I doubt that substantial soil warming was realized, at least not in the first month. By removing the chambers every August/September, any difference in soil temperature would be undone every Autumn to Spring period, in contrast to the experiments where the OTCs remained in place. Surely this must have affected the soil warming intensity? By averaging out all RE measurements over the growing season, independent of the amount of soil warming, and not taking this into account in the analysis, substantial error may have been introduced. The manuscript also fails to report on the differences in snow cover and wind speed, that have huge impacts on vegetation and thus on ER as well.
 - We did find an overall soil warming which is described on L. 199: "*Warming with OTCs led to a mean 1.4 °C [95% CI 0.9-2.0 °C] (p<0.001; N=77) increase in growing season air*

Environmental drivers of ~~persistently~~ increased ecosystem respiration with warming in the tundra

temperatures, a mean 0.4 °C [CI 0.2-0.7 °C] ($p < 0.001$; $N = 118$) increase in growing season soil temperature, and a mean 1.6% [CI 0.8-2.4%] ($p < 0.001$; $N = 111$) decrease in growing season soil moisture compared to ambient conditions (based on meta-analysis models, *Ext. Table 3*).” We include the effects on soil temperature and moisture in the Visual Abstract now as well:

- Regarding: ‘we did not take into account the amount of soil warming in our analysis, and therefore may have introduced substantial error’. → As part of our main meta-regression models investigating indirect warming effects, we specifically assess the relationships between the ER response to warming (Hedges SMD) and the microclimatic warming effects (Hedges SMD of air temperature, soil temperature, and soil moisture, **Ext. Table 5b**) which is equivalent to the amount of warming the reviewer refers to. Yet, we did not find any significant relationships between the ER effect sizes and the microclimatic effect sizes as described in (L. 250).
 - Finally, regarding: A) ‘I doubt that substantial soil warming was realized, at least not in the first month’. → Soil warming was realized throughout the growing season. We found a mean estimated Hedges SMD of 0.18 [CI 0.08-0.29]***, corresponding to a mean estimated increase of soil temperatures with 0.44°C [CI 0.24-0.65 °C]*** (**Ext. Table 3**). Further, we now checked seasonal variation in the effect size of soil warming, by running a meta-regression model testing the effect of the measurement ‘Day of the year’ on daily ER Hedges SMD. This showed that soil warming became stronger over the season (green box; table below), but was already there at the beginning of the growing season (day 152 – 1st of June). However, we do not believe this has confounded our results because of the following reasons. Second, when we run the meta-regression model that tests for an effect of soil warming (Hedges SMD soil temperature) on ER Hedges SMD separately for the early half vs. late half of the growing season, we find positive ER responses in both parts of the growing season (output with red boxes below).
 - Results of meta-regression model testing the effect of the measurement ‘Day’ on daily ER Hedges SMD per experiment, and statistics and visualization of the result in a scatter plot:

	estimate	se	zval	pval	ci.lb	ci.ub	
	<chi>	<chi>	<chi>	<chi>	<chi>	<chi>	<S3: noquote>
intrcpt	0.0819	0.2030	0.4033	0.6867	-0.3160	0.4798	
Day	0.0018	0.0007	2.4948	0.0126	0.0004	0.0032	*

Environmental drivers of ~~persistently~~ increased ecosystem respiration with warming in the tundra

- *Results of meta-regression models testing the effect of soil warming (Hedges SMD soil temperature) on ER Hedges MSD separately for the early half vs. late half of the growing season:*

Early season [Julian Day 152-197]

	estimate <chr>	se <chr>	zval <chr>	pval <chr>	ci.lb <chr>	ci.ub <chr>	<S3: noquote>
intrcpt	0.6852	0.0725	9.4500	<.0001	0.5431	0.8273	***
SMD_ST	-0.0176	0.0257	-0.6861	0.4927	-0.0679	0.0327	

Late season [Julian Day 198-244]

	estimate <chr>	se <chr>	zval <chr>	pval <chr>	ci.lb <chr>	ci.ub <chr>	<S3: noquote>
intrcpt	0.6440	0.0675	9.5463	<.0001	0.5117	0.7762	***
SMD_ST	0.0822	0.0756	1.0882	0.2765	-0.0659	0.2304	

- B) In response to ‘By removing the chambers every August/September, any difference in soil temperature would be undone every Autumn to Spring period, in contrast to the experiments where the OTCs remained in place. Surely this must have affected the soil warming intensity?’ à Although our analyses focused on the growing season (June-August), when all OTC chambers would have been in place, removing the OTCs in winter could still affect the intensity of the microclimate effects during the growing season due to lagged effects. To assess this, we checked with an additional meta-regression model whether microclimatic effect sizes, i.e. Hedges SMD of air temperature, soil temperature, and soil moisture, were influenced by OTC removal or not. See the distribution of datasets where OTCs were removed or not, as well as the model output below.

- *Air temperature*: no significant effect of OTC removal on our Hedges SMD of air temperature in the growing season.

```
##           estimate      se      zval      pval      ci.lb      ci.ub
## intrcpt           0.2050  0.1214  1.6884  0.0913  -0.0330  0.4431 .
## OTC_RemovalYes  0.1397  0.1435  0.9731  0.3305  -0.1417  0.4210
##
## ---
## Signif. codes:  0 '***' 0.001 '**' 0.01 '*' 0.05 '.' 0.1 ' ' 1
```

- *Soil temperature*: no significant effect of OTC removal on our Hedges SMD of soil temperature in the growing season.

```
##           estimate      se      zval      pval      ci.lb      ci.ub
## intrcpt           0.1737  0.0825  2.1064  0.0352  0.0121  0.3353 *
## OTC_RemovalYes  0.0444  0.1117  0.3975  0.6910  -0.1745  0.2634
##
```

Environmental drivers of ~~persistently~~ increased ecosystem respiration with warming in the tundra

- *Soil moisture*: significant effect of OTC removal on our Hedges SMD of soil moisture in the growing season, with larger SMD values (smaller decreases) when OTCs were removed in winter.

```
##
##               estimate      se      zval      pval      ci.lb      ci.ub
## intrcpt          -0.5672    0.1057   -5.3643   <.0001   -0.7744   -0.3599   ***
## OTC_RemovalYes    0.3321    0.1538    2.1589    0.0309    0.0306    0.6337    *
```

- In sum, there was no effect of removing the OTCs in winter on the increase in air or soil temperature due to warming. However, removal of OTCs in winter did affect the decrease in soil moisture due to warming, with removal of OTCs leading to a smaller decrease in soil moisture with warming in the growing season. Yet, we do not think this confounded our results of the lack of relationship between the ER response and the soil moisture response to warming (**Ext. Table 5b**). Namely, we do not find an interaction between OTC removal and the SMD of soil moisture on the SMD of ER (see below), indicating that relationships between SMD ER and soil moisture are similar for datasets where OTCs were removed, as for datasets where they were not removed.

	estimate <chr>	se <chr>	zval <chr>	pval <chr>	ci.lb <chr>	ci.ub <chr>	<S3: noquote>
intrcpt	0.6011	0.1159	5.1852	<.0001	0.3739	0.8284	***
SMD_SM	-0.0505	0.0714	-0.7069	0.4796	-0.1904	0.0894	
OTC_RemovalYes	0.0142	0.1646	0.0864	0.9312	-0.3083	0.3367	
SMD_SM.OTC_RemovalYes	0.0567	0.1147	0.4938	0.6214	-0.1682	0.2815	

- C) ‘The manuscript also fails to report on the differences in snow cover and wind speed, that have huge impacts on vegetation and thus on ER as well’.
 - Although we agree that snow cover and wind speed confound OTC effects, we believe that this is minimal, because (i) we do not find a difference in the ER response rates with warming between experiments that kept their OTCs in the winter versus not; (ii) although these aspects can indeed affect vegetation structure/height, we do not find that changes or differences in vegetation structure (e.g. SMD shrub cover; SMD mean height) influenced the ER response rates (**Ext. Table 5b**, L. 250); and (iii) our analysis focuses on the growing season, a time where snow cover and high wind speeds are minimal, hence they could have influenced the ER response rates only indirectly through vegetation changes, which we assessed (see (ii)).

3. This brings me to a major issue. The manuscript only reports ER fluxes; I guess also GPP data will be available but being spared for a follow-up paper. The problem is that GPP is a more important

Environmental drivers of ~~persistently~~ increased ecosystem respiration with warming in the tundra

determinant for ER than air temperature and even soil temperature: Plants can only respire more if GPP increases to sustain the higher respiration; microbes that respire more deplete their food source, which results in a shrinking microbial population and eventually reduces microbial respiration to the level of soil carbon inputs under the warmed conditions. Thus: long-term sustained increases in ER must be accompanied by increased GPP and NPP. In this study, there are clear indications that the 30% increase in ER is more driven by a warming-induced productivity increase than directly by warming. For example, there is a clear tendency towards reduced bulk density in both organic and mineral layers, for which the most likely reason is increased soil organic matter inputs and associated microbial- and faunal activity. Bulk density may also have decreased because the OTCs provide warmer and wind-free shelters for soil animals. Total C stocks are not given, but C concentration did on average not decrease, which one would expect if stimulation of ER would be driven by the warming and not by increased GPP. Given that I believe that increased productivity is driving increased ER, plus that soil carbon stocks were not affected by the OTCs, I cannot see the relevance of the 30% increase in ER for the land carbon-climate feedback, which is how the authors currently sell the story.

- We appreciate this comment that led us to thoroughly rephrase our objectives and to collect additional data on ecosystem respiration partitioning to further strengthen our findings and storyline. Below, we reply in detail to the different subsections and statements from the reviewer's comments:

A) 'The manuscript only reports ER fluxes; I guess also GPP data will be available but being spared for a follow-up paper.'

- Although we strongly agree that adding GPP data to our database would allow to explore climate feedbacks and C budgets in detail, it would, at the same time, take our current story away from what it is truly about: mechanistic drivers of changes in ecosystem respiration due to warming. We believe that investigating changes in, and drivers of, increased ER across the tundra is highly relevant and novel to assess on its own because of ER being a key component in the C balance. Adding GPP to the story would not allow us to explore the drivers of RE in this much depth since the two processes (GPP vs. ER) are driven by fundamentally different drivers. Understanding ER is therefore an important aim, and understanding GPP and its drivers could indeed be the next step.
- If we would collect additional **GPP** data, we should also collect **NEE** data to properly assess the C budget/net effects, which stretches beyond our objectives, and would require extensive additional sampling. Therefore, instead of extending our story to focus on net C budgets (NEE) driven by changes in GPP vs. ER, by collecting additional flux responses, we decided to strengthen our story on the ER partitioning (L. 214, 258, 402), since the reviewer suggested that increased ER would mainly be driven by increased R_a and not by increased R_h as well. This showed that both R_a and R_h increased by warming.

B) 'Plants can only respire more if GPP increases to sustain the higher respiration; microbes that respire more deplete their food source, which results in a shrinking microbial population and eventually reduces microbial respiration to the level of soil carbon inputs under the warmed conditions. Thus: long-term sustained increases in ER must be accompanied by increased GPP and NPP.'

- Although studies have shown indeed that climate warming may lead to increased (heterotrophic) respiration first, quickly followed by no significant increase in respiration after 5-10 years, possibly due to a combination of **microbial acclimation and substrate**

depletion^{6,19}, later studies have actually shown that these assumptions may not be correct for all types of ecosystems. In Dorrepaal et al. (2009)²⁰, for instance, ER response in a tundra site continued for at least 8 years, with the main part of this increase coming from heterotrophic decomposition and from **deeper soil layers**. Similarly, long-term consistent increases in respiration with warming and permafrost thaw have been found by Schuur et al. (2009)²¹. At first, it was speculated that these more persistent increases in ER in tundra sites might be a response typical of their organic rich, poorly decomposed soils: the decomposition per g soil C might be too low (still being quite cold) and the total C pool too large to deplete the substrate fast enough. However, even the classic hardwood forest story which proposed the accelerated mineralization and increased respiration response to warming to be short-lived when evaluating the first decade of warming⁶, turned out to be more complex than a simple carbon depletion story when they continued the measurements until 25 years²². Although of course, in the longer term, many more ecosystem-mediated responses start to affect the respiration response. Interestingly, our results demonstrate a similar non-linear trend with cumulative warming as for the hardwood forest: first a dip in responsiveness between 5-10 years of warming (i.e. the ER remains increased with warming, but decreases in magnitude, see L. 225 and **Fig. 3b**), then a recovery in responsiveness, i.e. increase of the response magnitude, which ultimately leads to no significant change in the long run (**Ext. Table 4a**).

- In addition to microbes delving into SOC pools in deeper soil layers, several studies have also shown that warming can lead to increases in respiration from **older soil carbon**, instead of just plant-related respiration and young soil C^{23,24}. These studies showed that soil microbes do not get starved by depletion of soil C very quickly, and that ecosystem-mediated changes in respiration pathways in the longer term may cause breakdown of even older soil C.

C) ‘In this study, there are clear indications that the 30% increase in ER is more driven by a warming-induced productivity increase than directly by warming’.

- Prompted by the reviewer’s hypothesis that the increase in ER is more driven by a warming-induced productivity increase than directly by warming, we collected additional data, i.e. the warming response of the autotrophic vs. heterotrophic components of ER, or the plant-related vs. microbial respiration, for a subset of 9 datasets used in the manuscript. With these new data (see overview below and **Ext. Table 4b, Supp. Discussion #1**), we demonstrate clearly that the 30% increase in ER is likely not driven only by a warming-induced productivity increase leading to increased plant-related respiration, but equally so by a warming-induced increase in microbial respiration. We present these results in the revised manuscript on L. 214, in **Ext. Table 4b**, and explain and visualize them in **Supp. Discussion #1**.
 - Plant-related respiration Ra (N=9)
 - Mean Hedges SMD ↑ 0.44 [0.08, 0.80] ***
 - Mean percentage increase ↑ 56.9 [6.9, 130.2] % ***
 - Heterotrophic respiration Rh (N=9)
 - Mean Hedges SMD ↑ 0.92 [0.36, 1.48] ***
 - Mean percentage increase ↑ 54.5 [31.8, 81.2] % ***
 - ER of the partitioning subset (N=9)
 - Mean Hedges SMD ↑ 0.72 [0.42, 1.02] *** (N=9)
 - Mean percentage increase ↑ 39.2 [19.6, 62.1] %
 - ER (N=136) – full dataset: to demonstrate that ER (N=9) is representative of full dataset trend

Environmental drivers of ~~persistently~~ increased ecosystem respiration with warming in the tundra

- Mean Hedges SMD \uparrow 0.57 [0.44, 0.70] ***
- Mean percentage increase \uparrow 29.8 [22.3, 37.8] % ***

D) ‘For example, there is a clear tendency towards reduced bulk density in both organic and mineral layers, for which the most likely reason is increased soil organic matter inputs and associated microbial- and faunal activity. Bulk density may also have decreased because the OTCs provide warmer and wind-free shelters for soil animals. Total C stocks are not given, but C concentration did on average not decrease, which one would expect if stimulation of ER would be driven by the warming and not by increased GPP.’

- As part of our revision, we have also gone through the entire data used in the manuscript. With this, a few outliers were detected, and removed from the analysis after discussion with the data contributors. Due to this update, the effects of warming on local soil conditions were updated, see **Ext. Table 3**. Hence, the reduced bulk density is no longer significant, and no longer interpreted as a trend in the updated manuscript. Further, we only have total C stock data at experiment level and not in response to the warming treatment (**Ext. Table 2**), so we could not test whether this was affected by the warming treatment. The effects of warming on total C stock are also heavily debated, and could be a paper on its own^{25,26}. Finally, many of our experiments are located on sites with organic soils with very high C content, so detecting the impact of warming on soil C stocks would be difficult here, due to a lack of sufficient spatial variation in soil C content.

E) ‘Given that I believe that increased productivity is driving increased ER, plus that soil carbon stocks were not affected by the OTCs, I cannot see the relevance of the 30% increase in ER for the land carbon-climate feedback, which is how the authors currently sell the story.’

- We show with our additional analyses (see our response to part C) that the reviewer’s hypothesis that ‘increased productivity is (solely) driving increased ER’ is not supported by our data, where increased ER arose from both increased plant *and* microbial respiration. Further, we do not know whether soil C stocks are affected by the OTCs, given that we could not calculate treatment-level stocks (as explained in more detail in comment #10). Based on this evidence, and the reasons explained above, we believe our results are highly valuable and implicational for the land carbon-climate feedback, which we include in more detail now on L. 404. Namely, “... *Namely, while increased autotrophic respiration might be met by an equally strong increase in productivity, a combination of increased autotrophic and heterotrophic respiration with warming is likely to free up more carbon than can be compensated for by increased plant productivity alone. While our analyses pertain to the growing season ER fluxes only, changes in non-growing season fluxes are likely to aggravate future C losses. Our findings therefore imply a potentially strong, persistent positive climate feedback coming from the tundra, even under modest warming during the coming decades, which may accelerate, rather than mitigate the impact of global warming*”.
4. The global scaling depends on the model found across these warming experiments, but does not extrapolate the uncertainty of the model. Then, this model is multiplied with an ISRIC dataset providing the spatial data for the model variables, which has gigantic uncertainties for the arctic region, which could have been quantified given the available site data, but are also fully ignored. In addition, one of the drivers is not even available in the ISRIC dataset but is approximated. Also this proxy was not evaluated using the available empirical data and its uncertainty not extrapolated

in the final product. Best remove this entire part of the manuscript or support it with a thorough uncertainty analysis.

- We have now fully revised the upscaling workflow, addressed the uncertainty not only from the meta-regression model but also from the ISRIC dataset and from using a proxy for C:N-ratio. The revised method is explained in detail in the Methods (*Spatial upscaling*) and summarised in a flow diagram in **Supp. Methods #3**. The main changes to the method are:
 - We now use the native resolution (250 m) soil data, for which ISRIC provides uncertainty estimates (which are not yet available for other resolution (i.e. 1 km and 5 km products)). This allowed us to estimate the uncertainty originating from the ISRIC data. Due to the large study area (10 million km²), we still needed to aggregate the soil data to 1 km resolution as even with this resolution we needed a supercomputer to run the uncertainty analysis.
 - We used the reported C:N-ratio and reported TN and TC from our database (**Supp. Fig 8**) to estimate the uncertainty originating of the proxy using TN and SOC to derive the C:N-ratio, and incorporated this uncertainty in our Monte Carlo uncertainty analysis.
 - In the Monte Carlo uncertainty analysis, we created 100 pairs of TN and C:N-ratio values for each 1 km² grid cell using a truncated multivariate random normal distribution generator, using the mean and standard deviation of these variables.
 - We then used the resulting 100 pairs of TN and C:N-ratios as input for our ROM model for each 1 km² grid cell, resulting in 100 estimations for mean and standard deviation. These were then combined to get the mean percentage change in ER induced by 1.4°C warming and its combined standard deviation.

Detailed comments

5. The duration effect should be studied within ecosystems, not across all experiments, to avoid other drivers confounding the analysis.
 - Please see our response to comment #5 from Reviewer 2 for this, as these comments were similar. In short: we *have* investigated both *within*-and *across*-experiment effects of duration, the former with linear regression models and the latter with meta-regression models which account for methodological differences across the experiments (**Fig. 3**, **Supp. Fig. 5** respectively). However, we think the across-experiments results are most informative, looking at the uneven length of the repeated measurements of ER within experiments we have, and the limited range of duration that most experiments cover, while covering a range of 0-25 years across all experiments. See **Supp. Discussion #6** as well for details.
6. Soil carbon and changes therein are crucial, but little to no Methods were given and changes not reported. Was it sampled uniformly?
 - Please see our response to your comment #2 for this. We *did* analyze changes in soil C concentration (SOC and SOM mg/g soil) due to the warming treatment, and report these in **Ext. Table 3**. We looked at total concentrations (TC, TN, C:N) in the mineral and organic layer separately, where the mineral vs. organic layer was determined by data contributors based on their site and ecosystem knowledge. These data was not sampled across a standardized measurement protocol, but this is very common in meta-analyses. We only included measurements using CNS element analyzers (the majority of measurements) and expressed all values as mg/g soil. We describe this in the Methods (L. 103, see also **Supp. Methods #1**). For

neither of the soil layers did we detect a significant trend in C concentrations with warming in the study sites (based on the mean Hedges SMD results, **Ext. Table 3**).

7. Fig 1: show the soil T response too

- See our response to your comment #2. We now added the soil temperature (and air temperature and soil moisture) effect resulting from experimental warming to the Visual Abstract and in the main text (L. 200).

8. Fig 2: N and C concentrations are given but soils have become lighter by the OTC treatment. Did you account for this also in the sampling? Sampling at the same depth is not appropriate if the have loosened.

- We looked carefully at the data and we detected an mistake in the way BD was provided to us and corrected it. This changed the relation to not significant (Hedges SMD bulk density of mineral layer: -0.24 [CI -0.70, 0.22]).
- Standardized sampling in such multi-participants studies is challenging. We standardized by separating samples from mineral and organic layers. Most contributors are part of the ITEX network which makes use of common protocols.

9. Fig 2: in the graphs you give multiple ER data for single driver observations. Please only show one ER per driver observation.

- We think the reviewer refers here to the fact that in several experiments, replicated environmental driver data from the same measurement year is used, relating this data to distinct, unique ER data points. We clarified this methodological aspect of our driver data collection and how we tested for the possible influence on our results with a sensitivity analysis in **Supp. Discussion #4** and **Supp. Fig. 5**. This showed that, even when we allowed less repetition of environmental driver data, the main results did not change. Please see our response to comment #8 of Reviewer #1 as well.

10. Line 260: this hypothesis could be tested by comparing the change in SOC stock between N-poor and N-richer sites...

- The reviewer refers to the hypothesis that *‘warming might stimulate plant root production and root exudation rates to access nutrients and prime microbial communities to mineralize nutrients (i.e. rhizosphere priming), thus enhancing root and microbial respiration and thus ER response rates.’* We believe the reviewer means we could test whether the change in SOC stock (e.g. Hedges SMD effect size) was significantly affected by the TN content. However, we do not have SOC stock values at treatment level (for both control and warmed plots), only at experiment (site) level, so unfortunately cannot test this.
- Please note that, particularly because one should be very cautious about the specific sampling method for accurate SOC stock quantification, we did not find it appropriate to calculate SOC stock values at the treatment level based on the datasets where we have both bulk density and total C values available. That is, we cannot ascertain the similarity in Methods for the two drivers across experiments, e.g. sampled where in the plots, at which depth, for us to properly assess SOC stock as a driver^{27–31}. Through our meta-regression models, we do account for this sampling variation, but this is based on the original data (e.g. TC, BD) and not derived data

Environmental drivers of ~~persistently~~ increased ecosystem respiration with warming in the tundra

(e.g. SOC stock based on TC and BD). Further, the number of datasets that have both bulk density and TC available, are limited and the majority of those come from the same site, i.e. 6 out of the 9 datasets for mineral layer data from Latnjajaure (experiments SWE_1 to 5), and 9 out of the 15 datasets for organic layer. Even if we did run a meta-regression based on derived SOC stock values, the information to extract from that would not be very informative for our overall dataset.

11. Figure 3: A) was there an interaction between vegetation type and soil warming? B) Did soil warming differ between growing season-only and full year applied infrastructures?

- We checked these questions through additional models and plots and report the results point per point below:

A) Was there an interaction between vegetation type and soil warming?

- We interpret this question as ‘whether the amount of soil warming differed significantly between vegetation classes’. If so, we believe the reviewer thinks that this might have confounded our result that vegetation class did not affect ER Hedges SMD (**Ext. Table 5c**). We tested this by running a meta-regression model evaluating effect of vegetation class on the Hedges SMD of soil temperature. There is no significant effect of vegetation class on these Hedges SMD values for soil temperature ($Q_m=6.24$, $p\text{-val} = 0.18$, see plot below). Because it does not appear like a potential interaction could have confounded our results, we did not add this to our manuscript or story.

B) Did soil warming differ between growing season-only and full year applied infrastructures?

- See our response to your comment #2 for this. To test whether soil warming differed between growing-season only and full-year applied infrastructures, we ran a similar meta-regression model as for A), but testing the effect of OTC removal as predictor. We do not find an effect of removal of OTCs (i.e. growing-season only application) vs. no removal of OTCs (full-year application) on the amount of soil warming (Hedges SMD soil temperature; $Q_m=0.16$, $p\text{-val} = 0.69$, see plot below). Since we find no apparent confounding, we did not add this to our manuscript.

12. Line 294:

A) Surely, the depth from the surface where the mineral soil starts must have influenced the importance of mineral soil N and C/N. Was this analyzed?

B) Also, the role of N turnover is hypothesized, probably correctly, but why wasn't a meta-analysis conducted on the N turnover directly? I assume these data were available at many sites, given that N turnover is well known to be a key determinant of the ecosystem response to warming.

- To your first question A), as earlier mentioned, soil environmental driver data was obtained from past measurements, and represented samples from the mineral or organic soil layer, without specifications of which depth the N / C:N samples were taken at. Slight differences in Methods are common in meta-analysis that gain power by numbers despite sometimes lacking sampling specific information. Moreover, as most of the sites belonged to the ITEX network sampling Methods were most likely relatively standardized compared to other meta-analyses.
- To your second question B): because it would be sensible to collect N mineralization or turnover data according to a standardized protocol³², and at this stage the majority of experiments likely do not have this data yet, we did not collect/analyze this here.

13. Fig 5: the text states that the ER response extends to >100%, then best expand the scale to 100%; although as stated above I have little confidence that this analysis bears any meaning

- The scale has been updated accordingly in **Figure 6**. See our response to your comment #4 for details on how we have revised our upscaling analysis, including adding an uncertainty analysis to it.

14. Line 327: this is not true. It depends on the persistency of an imbalance between C inputs and respiratory losses, not on the persistency of stimulated ER.

- Agreed, and we believe we now improved the discussion on the C balance. We have revised the sentence on L. 375. It now reads: *“The magnitude of the long-term land carbon-climate feedback critically depends on the balance between C inputs through changes in photosynthetic plant uptake and C losses through ecosystem respiration. In this regard, the magnitude and persistency of C losses through ER is a crucial component in evaluating this feedback and how it may change with future warming.”*

15. Line 334: explaining why some of the long-term running sites show increased ER while others show declining ER to me would be a very informative analysis

- Importantly, we believe the reviewer means to say ‘increased ER *responses* to warming while others show declining *responses*’. This is an important difference, as we are not analyzing temporal patterns in absolute values of ER here, but trends in how *warming* affects ER over time (quantified as ER Hedges SMD or effect sizes). We agree that the occurrence and drivers of potentially different temporal patterns between experiments would be a very interesting question to address in a future meta-analysis. However, currently, the temporal resolution, replication and total duration of the available ER and driver data *within* experiments, as well as total replication and standardization of ER measurements *across* experiments are unfortunately not sufficient for this (yet).
- Please see our **Supp. Discussion #6** and **Supp. Fig. 5** though for detailed output of the within-experiment temporal patterns, and a discussion on why we decided that analysing baseline changes of ER were out of the scope for this manuscript in comment #4 of Reviewer #2.

16. Line 345: providing modelers with a more robust understanding would require you to show the responses of GPP and NPP as well.

- Please see our response to your comment #3 for this. We further believe that our partitioning results are equally interesting to modelers as NPP and GPP, since they are the underlying processes and thus give us more mechanistic insight.

17. SI Line 27: the differences in OTC height is not slight; the effect on wind penetrating to the soil surface must be huge.

- The reviewer probably refers here to the previous Methods Line 27 instead of Supp. Info L 27. In the Methods, we stated previously that ‘experiments differed slightly in OTC size (0.3-0.7m height),...’. This information, and how we assessed whether it influenced the ER response rates was also included in Supplementary Discussion section 1.

- We agree that a range of 40 cm height difference should not be considered ‘slight’ in these experiments, so we removed this word on L. 27 in the Methods. However, almost all studies had a height of 0.3-0.5 which is a much smaller range (see figure above).
- Since **OTC height** differences showed no significant effect on the ER Hedges SMD (**Supp. Fig. 3: Qm = 3.1, p = 0.08**), we disregarded it as an important driver. Further, when running an explorative backward selection model on Methodological drivers, OTC height was not selected

Environmental drivers of ~~persistently~~ increased ecosystem respiration with warming in the tundra

in the final model (see **Supp. Discussion #2**). Neither did OTC height significantly affect ER Hedges SMD in a single-factor model.

- Wind circulation effects may not only depend on height. To address the reviewers concern, we also tested whether **OTC plot size** might have influenced our ER Hedges SMD, since the plot size reflects differences in diameters, and this may also affect wind circulation in addition to OTC height. Plot size ranged from 0.25 to 4.3 m² (see histogram below). Plot size did not affect our ER Hedges SMD, therefore we believe this reaffirms our arguments that chamber dimensions did not affect our main findings. See **Supp. Discussion #2** for a detailed explanation on how we checked for, and report on, possible methodological bias in our results.

	estimate <chr>	se <chr>	zval <chr>	pval <chr>	ci.lb <chr>	ci.ub <chr>	<S3: noquote>
intrcpt	0.6510	0.1086	5.9955	<.0001	0.4382	0.8639	****
Plot_Size	-0.0529	0.0557	-0.9498	0.3422	-0.1620	0.0562	

18. Line 41: this description does not suffice. Imagine that the OTCs were placed on day X and measurements started on day X+1, then likely vegetation profited from the reduced cooling at night, but the soil would not have changed. Warming the soil, especially the underlying mineral soil, with passive warming as applied here, must take many weeks/months. Removing the OTCs every autumn undoes the warming, such that even after 10 years this artefact remains. Details should be given on this.

- We are unsure whether the reviewer’s comment really relates to Line 41 in the initial main text since this was about how we calculated ER effect sizes. However, we can reply based on the content of the comment, namely:

A) ‘Warming the soil, especially the underlying mineral soil, with passive warming as applied here, must take many weeks/months.’

- See our response to your comment #2 for this. Passive warming in the experiments did lead to significant soil warming even at the start of the growing season (Julian day 152), so we believe that a potential delay in, or lack of, soil warming was not an issue in our analyses. Further, we checked whether there was seasonal variation in the soil warming response and how this might have influenced the results. Please see our response to your comment #2 for that.

B) ‘Removing the OTCs every autumn undoes the warming, such that even after 10 years this artefact remains. Details should be given on this.’

- See our response to your comment #2 and 11 for this. We checked whether OTC removal in winter influenced 1) the effect size of soil warming, which it did not. Neither did it influence

2) our ER response, so we believe we do not need to account for removal of OTCs in our analysis based on this.

19. Unit for NPP is wrong

- We added year⁻¹ to correct the unit for NPP: kg C m⁻² year⁻¹ in the Methods and **Supp. Methods #1**.

20. Define CAVM

- We defined CAVM in the Methods and **Supp. Methods #1**.

21. Define how soil environmental variables were measured (what depth?)

- Soil temperature is measured for every *ER measurement* level with climate loggers, and hence were measured at a relevant depth for the ER measurement. i.e. every ER measurement had coinciding soil temperature and soil moisture values (if these data were available for the experiment). The ER measurements, and hence temperature measurements were done at different depths. We provide the overview of the depths, and how many datasets and experiments measured at different depths, below and in **Supp. Methods #1** now. Since the **majority of the measurements were done at 5 cm**, we selected the moisture or temperature data from this depth in case multiple measurements were available.

Soil moisture			Soil temperature		
Depth (cm)	Number of datasets	Number of experiments	Depth (cm)	Number of datasets	Number of experiments
2.5	5	5	2	7	6
3	11	5	2.5	1	1
3.5	9	5	3	2	1
3.75	3	1	5	78	39
5	40	17	7.5	8	4
6	13	1	10	22	5
7.5	11	5	total:	118	56
10	19	5			
total:	111	44			

- The other soil environmental drivers were measured at *plot* level, i.e. soil organic matter (SOM, %), total C concentration (TC, %), total N concentration (TN, %), C:N-ratio, bulk density (BD, g cm⁻³), and pH from the mineral or organic layer; as well as organic layer depth (OLdepth, cm). For these drivers, we requested data contributors to provide their data as either from mineral or organic layer, so there are no specified depths at which samples were taken. We trust that the data contributors know their site well enough to correctly define their organic and mineral layers best, and that they provide samples that are representative of that layer.
- Important to note, it is common to have different measurement methodologies across studies in meta-analyses. This is also why we perform our analysis with the meta-analysis and -regression approach, since this statistical technique takes sampling variation into account.

22. Methods should allow the reader to reproduce the analysis, but apart from explaining SMD, I have no idea what exactly was done. I assume you used the difference in seasonal mean ER from an experiment? This gives the effect size, but how did you deal with the differences in warming throughout the season, across soil depths, and across years?

- We have revised the Methods to better explain what we have done. To respond to your specific questions: we indeed used the difference in growing season mean ER from control and warmed plots for every dataset to calculate ER response (SMD) for that dataset. A ‘dataset’ represented here a unique ER measurement year x experiment combination as we often had repeated measurements of ER from the same experiment. This was used as our response variable (primary effect size) for the multivariate meta-analysis and meta-regression models (‘rma.mv’ from metafor package) and is explained on L. 50 in the Methods.
- Further, we performed multivariate meta-regression models to analyze precisely what the reviewer hints at, e.g. to investigate which drivers influenced variation in the mean growing season ER. The meta-regression models, which are explained on L. 142 in the Methods and results shown in **Fig. 3-5** and **Ext. Table 5** are comparable with linear mixed effect models, in that they can account for random effect structure, as well as for temporal autocorrelation structures. But at the same time they increase precision of the estimated effects by weighting each dataset by the inverse of its estimated sampling variance. To answer your specific queries, see below: ‘How did you deal with...’

A) the differences in warming throughout the season?

- Please see our response to your comment #2 on seasonal variation.

B) the differences in warming across soil depths?

- In our response to your comment #21, we show that soil temperature data was measured at a range of soil depths, i.e. between 2.5 and 10 cm deep, but with the majority of experiments measuring soil temperatures at 5 cm deep (17 out of the 44 experiments; 40 out of the 111 datasets). These soil depths are relevant for plant growth and microbial activity, and hence for ER. Soil depth did not influence the Hedges SMD of soil warming (see results below). Detailed studies of how warming at different depths would affect ER is, given our results that deeper soil layers may be important for ER, an interesting avenue for further studies.

	estimate <chr>	se <chr>	zval <chr>	pval <chr>	ci.lb <chr>	ci.ub <chr>
intrcpt	0.1419	0.1515	0.9365	0.3490	-0.1551	0.4389
ST_Depth	0.0103	0.0259	0.3998	0.6893	-0.0404	0.0611

C) the difference in warming across years?

- As part of our meta-regression models (**Ext. Table 5b**: indirect warming effects), we tested whether the Hedges SMD of air temperature, soil temperature (i.e. reflecting the air and soil warming magnitude and direction per dataset) influenced the ER response. They did not, which implies for us that differences in achieved warming across years and experiments does not drive variation in ER response rates.

23. I did not fully understand the statistical approach, so some of the comments above may be unfair.

- We appreciate the honesty of the reviewer, and hope we have adequately addressed the concerns and unclarities to make our manuscript and the analyses we have done more comprehensible.

References

1. Jenkins, D. G. et al. A meta-analysis of isolation by distance: Relic or reference standard for landscape genetics? *Ecography (Cop.)*. 33, 315–320 (2010).
2. Vaessen, T. et al. The association between self-reported stress and cardiovascular measures in daily life: A systematic review. *PLoS One* 16, 1–28 (2021).
3. Signorini, M., Midolo, G., Cesco, S., Mimmo, T. & Borruso, L. A Matter of Metals: Copper but Not Cadmium Affects the Microbial Alpha-Diversity of Soils and Sediments — a Meta-analysis. *Microb. Ecol.* (2022) doi:10.1007/s00248-022-02115-4.
4. Konstantopoulos, S. Fixed effects and variance components estimation in three-level meta-analysis. *Res. Synth. Methods* 2, 61–76 (2011).
5. Trikalinos, T. A. & Olkin, I. Meta-analysis of effect sizes reported at multiple time points: A multivariate approach. *Clin. Trials* 9, 610–620 (2012).
6. Ishak, K. J., Platt, R. W., Joseph, L., Hanley, J. A. & Caro, J. J. Meta-analysis of longitudinal studies. *Clin. Trials* 4, 525–539 (2007).
7. Higgins, J. P. T. & Thompson, S. G. Quantifying heterogeneity in a meta-analysis. *Stat. Med.* 21, 1539–1558 (2002).
8. Nakagawa, S. & Santos, E. S. A. Methodological issues and advances in biological meta-analysis. *Evol. Ecol.* 26, 1253–1274 (2012).
9. Epron, D. Separating autotrophic and heterotrophic components of soil respiration: Lessons learned from trenching and related root-exclusion experiments. *Soil Carbon Dyn. An Integr. Methodol.* 157–168 (2010) doi:10.1017/CBO9780511711794.009.
10. Hicks Pries, C. E. et al. Decadal warming causes a consistent and persistent shift from heterotrophic to autotrophic respiration in contrasting permafrost ecosystems. *Glob. Chang. Biol.* 21, 4508–4519 (2015).
11. Melillo, J. M. et al. Soil warming and carbon-cycle feedbacks to the climate system. *Science (80-.)*. 298, 2173–2176 (2002).
12. Luo, Y., Wan, S., Hui, D. & Wallace, L. L. Acclimatization of soil respiration to warming in a tall grass prairie. *Nature* 413, 4–7 (2001).
13. Dorrepaal, E. et al. Carbon respiration from subsurface peat accelerated by climate warming in the subarctic. *Nature* 460, 616–619 (2009).
14. Schuur, E. A. G. et al. The effect of permafrost thaw on old carbon release and net carbon exchange from tundra. *Nature* 459, 556–559 (2009).
15. Melillo, J. M. et al. Long-term pattern and magnitude of soil carbon feedback to the climate system in a warming world. *Science (80-.)*. 358, 101–105 (2017).
16. Hicks Pries, C. E., Schuur, E. A. G., Natali, S. M. & Crummer, K. G. Old soil carbon losses increase with ecosystem respiration in experimentally thawed tundra. *Nat. Clim. Chang.* 6, 214–218 (2016).
17. Crowther, T. W. et al. Quantifying global soil carbon losses in response to warming. *Nature* 540, 104–110 (2016).
18. van Gestel, N. et al. Predicting soil carbon loss with warming. *Nat. Publ. Gr.* 554, (2018).
19. Nayak, A. K. et al. Current and emerging methodologies for estimating carbon sequestration in agricultural soils: A review. *Sci. Total Environ.* 665, 890–912 (2019).
20. Poeplau, C., Vos, C. & Don, A. Soil organic carbon stocks are systematically overestimated by misuse of the parameters bulk density and rock fragment content. *Soil* 3, 61–66 (2017).
21. Goidts, E., Van Wesemael, B. & Crucifix, M. Magnitude and sources of uncertainties in soil organic carbon (SOC) stock assessments at various scales. *Eur. J. Soil Sci.* 60, 723–739 (2009).
22. Tadiello, T., Perego, A., Valkama, E., Schillaci, C. & Acutis, M. Computation of total soil organic carbon stock and its standard deviation from layered soils. *MethodsX* 9, 101662 (2022).
23. Zhi, J. et al. Estimating soil organic carbon stocks and spatial patterns with statistical and GIS-based Methods. *PLoS One* 9, 30–33 (2014).
24. Rustad, L. E. et al. A meta-analysis of the response of soil respiration, net nitrogen mineralization, and aboveground plant growth to experimental ecosystem warming. *Oecologia* 126, 543–562 (2001).

Reviewer Reports on the Second Revision:

Referees' comments:

Referee #1 (Remarks to the Author):

The authors have provided a clear response letter, and I appreciate the effort and time spent clarifying and improving this manuscript. The strength of this manuscript is the compilation and analysis of this dataset to identify generalities in trends in ecosystem respiration due to warming in the tundra.

Overall summary of the results provided in this study.

1- I agree that the study provides the important result that ER has increased on average by 30% based on the 136 datasets and the meta-analysis. This is a robust result.

2- This increase in ER is (likely) due to a comparable increase in both plant-related and microbial respiration. This is a preliminary result based on 9 studies. Thus, this tone needs to be respected throughout the manuscript.

3- The magnitude of warming effects depended on changes in local soil conditions and context-dependent spatial variation of these conditions. This result is not surprising and confirms expectations based on previous studies. That said, this meta-analysis highlights the challenge in predicting/modeling the warming effects across this heterogeneous region.

4- Information about how nitrogen limitation influences the respiration response is a novel result of this meta-analysis, as it provides insights into mechanisms.

5- The upscaling approach is preliminarily overinterpreted (mainly in the abstract). The upscaling exercise provides an output to visualize the spatial variation of the warming sensitivity and the uncertainty of those estimates. Because the uncertainty is so large, I respectfully disagree that the authors provide confidence in interpreting the respiration increase using this upscaling approach (mainly in the abstract). They acknowledge this limitation in their response to the reviewers, so these results should be interpreted carefully.

There are two possibilities. One is to remove the analysis as has been suggested throughout the review process. Second, the authors could keep the analysis but avoid overinterpretation due to the limitations of these preliminary results (pending editorial decision). They must tone down the wording of their current interpretation (throughout the abstract and text), or they could highlight the need to increase information in other regions of the tundra to potentially reduce the large uncertainty in this upscaling approach.

Comments in detail

Lines 88-90- These results are based on 9 studies, so it needs to be explicit that $n=9$ to avoid misleading readers (assuming this conclusion is based on 136 datasets).

Lines 96-97 – The authors added, “and can be used to improve model estimations.” The authors must be explicit in the manuscript on how this will be possible with their new information.

Line 100 – The upscaling visualizes the spatial variation of the predicted magnitudes and associated uncertainty. Furthermore, the authors must be transparent with the readers that the uncertainty is very large, complicating the interpretation of the results. See alternative comments in point 5.

Lines 156-158 – These hypotheses are not novel, as acknowledged by the authors in the response to the reviewers. However, I think the value is to highlight in the manuscript that this analysis will test (and confirm) these expectations using this meta-analysis approach. This is the wording that the authors used in the response to the reviewers, and I think it is valuable to use it in the main text to show the study's novelty.

Lines 220-224- The authors shared their concerns about the high “apparent” Q10 value reported in this study (see response to the reviewers). This numerical result is biased by the experimental approach, the meta-analysis, and the challenge to measure a “real” Q10 value. Considering the concerns of the authors and reviewers, I strongly suggest removing these Q10 results.

Line 229 – Maybe be explicit in recognizing that these results are based on the 9 available studies. For transparency, consider editing:

“Based on these limited data, the warming-induced increase in ER likely resulted from a significant increase in both plant-related and heterotrophic respiration, each contributing to the overall effect of warming on ER.”

Lines 271-273 – Here, the authors could include a statement saying that this preliminary interpretation should be verified with future ER partitioning studies.

Figure 4: Increase the font size to improve the readability of the figure legends. Remove the title on each figure. Revise that the legend provides appropriate information.

Section “Context-dependencies in respiration response to warming”: As discussed throughout the review process, the hypothesis behind these results is not novel, but what is novel is that the meta-analysis tests this expectation. This idea should be clarified in this section, acknowledging that the current expectation is that the responses should depend on changes in local soil conditions and context-dependent spatial variation, but this study corroborates this with the present analysis.

Lines 355-369 – As discussed throughout the review process, this upscaling approach has several limitations. Arguably, it is an analysis that is too early to perform due to the limited spatial availability of training data as the authors are interpolating across large areas. The Revised Supp. Fig. 9 shows the dispersion of the training data, and clearly, this data cannot reproduce the “real” statistical properties of the gridded data. Removing this analysis is strongly recommended, but if the authors and the editor decide to keep it, several actions are needed.

a) Report the model performance formally in the main text.

b) I recommend reorganizing lines 355-369. First, start recognizing that uncertainty is very large (>100%), that this is a preliminary analysis, and that uncertainty originated from the input data (i.e., lines 367-369). Then, all interpretations must acknowledge that the uncertainty is larger than the estimates and, therefore, tone down the way the text is written (and the abstract). There are many sections that need correction, but one example is line 363. It currently says “..., which corresponds to

a 32% increase”, but it should say “..., which corresponds to a 32% increase with >XX% uncertainty”. Note that the uncertainty is >100%, but the authors should properly calculate that.

Lines 388-391- Be more explicit and point the reader to an example.

Referee #1 (Remarks on code availability):

Currently, the data and the code are in a Google folder, which does not follow FAIR principles. The data and code should be published following FAIR principles.

Response letter 3 – January 2024

Referee #1

1. The authors have provided a clear response letter, and I appreciate the effort and time spent clarifying and improving this manuscript. The strength of this manuscript is the compilation and analysis of this dataset to identify generalities in trends in ecosystem respiration due to warming in the tundra.

Overall summary of the results provided in this study.

1- I agree that the study provides the important result that ER has increased on average by 30% based on the 136 datasets and the meta-analysis. This is a robust result.

2- This increase in ER is (likely) due to a comparable increase in both plant-related and microbial respiration. This is a preliminary result based on 9 studies. Thus, this tone needs to be respected throughout the manuscript.

3- The magnitude of warming effects depended on changes in local soil conditions and context-dependent spatial variation of these conditions. This result is not surprising and confirms expectations based on previous studies. That said, this meta-analysis highlights the challenge in predicting/modeling the warming effects across this heterogeneous region.

4- Information about how nitrogen limitation influences the respiration response is a novel result of this meta-analysis, as it provides insights into mechanisms.

5- The upscaling approach is preliminarily overinterpreted (mainly in the abstract). The upscaling exercise provides an output to visualize the spatial variation of the warming sensitivity and the uncertainty of those estimates. Because the uncertainty is so large, I respectfully disagree that the authors provide confidence in interpreting the respiration increase using this upscaling approach (mainly in the abstract). They acknowledge this limitation in their response to the reviewers, so these results should be interpreted carefully. There are two possibilities. One is to remove the analysis as has been suggested throughout the review process. Second, the authors could keep the analysis but avoid overinterpretation due to the limitations of these preliminary results (pending editorial decision). They must tone down the wording of their current interpretation (throughout the abstract and text), or they could highlight the need to increase information in other regions of the tundra to potentially reduce the large uncertainty in this upscaling approach.

- We thank Reviewer #1 for the constructive comments. We appreciate the thoroughness of their review and concrete suggestions that improved the final manuscript.
- We believe our revised manuscript addresses the remarks that were raised, as outlined below.

Comments in detail

1. Lines 88-90- These results are based on 9 studies, so it needs to be explicit that $n=9$ to avoid misleading readers (assuming this conclusion is based on 136 datasets).
 - Agreed. We have added the sample size with the respective results to the abstract to increase clarity and avoid confusion (L 94, 96).
2. Lines 96-97 – The authors added, “and can be used to improve model estimations.” The authors must be explicit in the manuscript on how this will be possible with their new information.
 - This phrase was removed from the abstract, but we have incorporated a concrete example later in the Main text, see our response to your comment #11.
3. Line 100 – The upscaling visualizes the spatial variation of the predicted magnitudes and associated uncertainty. Furthermore, the authors must be transparent with the readers that the uncertainty is very large, complicating the interpretation of the results. See alternative comments in point 5.
 - Since we followed the suggestion from the Editor to remove the upscaling from the abstract (see their comment #1), this final statement was removed. We keep the upscaling as part of the paper, discussing it in the main text only (L 319 ‘Spatial patterns’ paragraph). See also our response to your comment #10 for how we have adjusted its wording.
4. Lines 156-158 – These hypotheses are not novel, as acknowledged by the authors in the response to the reviewers. However, I think the value is to highlight in the manuscript that this analysis will test (and confirm) these expectations using this meta-analysis approach. This is the wording that the authors used in the response to the reviewers, and I think it is valuable to use it in the main text to show the study's novelty.
 - We agree that the novelty aspect could still be stressed more compared to what previous studies found. We rearranged and rephrased this paragraph. It now reads (L 156):
*“Here we aim to quantify with meta-analysis the magnitude and variability of the impact of warming on ER across the tundra, and to understand with meta-regression models how variation in the ER response depends on warming duration, indirect effects of warming, and context-dependencies. We therefore collated 24035 daytime, growing season ER measurements from 28 sites across the tundra biome, covering 56 passive in situ open-top chamber [OTC] warming experiments across four continents, with warming duration ranging from one summer to 25 years at the time of ER measurements (Fig. 1a; Supp. Table 1). **No single study has assessed the ER response to in situ experimental warming yet across a large spatiotemporal environmental gradient in the tundra, with a standardized experimental setup while including also an extensive set of environmental drivers (see below).** Based on previous studies, we expect an overall positive ER response to warming (both R_a and R_h) across all experiments, but significant variation in the magnitude of the response across time and space. We expect these differences in the magnitude to be driven (i) indirectly by variation in warming-induced changes in local environmental conditions, as well as (ii) directly by context-dependent variation in environmental conditions.”*
5. Lines 220-224- The authors shared their concerns about the high “apparent” Q10 value reported in this study (see response to the reviewers). This numerical result is biased by the experimental

approach, the meta-analysis, and the challenge to measure a “real” Q10 value. Considering the concerns of the authors and reviewers, I strongly suggest removing these Q10 results.

- As mentioned in our previous response letter (Sept. 2023) to your comment #4, we do not object with removing our apparent Q10 from the manuscript, because it would not impact the main conclusions of our study. We thus removed it from the manuscript.
6. Line 229 – Maybe be explicit in recognizing that these results are based on the 9 available studies. For transparency, consider editing: “Based on these limited data, the warming-induced increase in ER likely resulted from a significant increase in both plant-related and heterotrophic respiration, each contributing to the overall effect of warming on ER.”
 - Agreed, we have highlighted the N=136 vs. N=9 more in this paragraph so that it stands out that they are analyses based on different sample sizes (L 211, 217), and rephrased L 220 as suggested.
 7. Lines 271-273 – Here, the authors could include a statement saying that this preliminary interpretation should be verified with future ER partitioning studies.
 - We have highlighted N=9 again on L 253, and inserted a statement on the need for larger-scale partitioning studies (L 254-258).
 8. Figure 4: Increase the font size to improve the readability of the figure legends. Remove the title on each figure. Revise that the legend provides appropriate information.
 - We have adjusted the figures and removed the titles.
 9. Section “Context-dependencies in respiration response to warming”: As discussed throughout the review process, the hypothesis behind these results is not novel, but what is novel is that the meta-analysis tests this expectation. This idea should be clarified in this section, acknowledging that the current expectation is that the responses should depend on changes in local soil conditions and context-dependent spatial variation, but this study corroborates this with the present analysis.
 - We have slightly adjusted the wording in this paragraph (e.g. ‘expected’, ‘corroborated’) to reiterate that context-dependencies were to be expected and that our analyses has confirmed which specific contextual factors drive variation in the ER response to warming across a large spatiotemporal gradient. We also rephrased the sentence on L 265 to highlight the novelty more in this paragraph.
 10. Lines 355-369 – As discussed throughout the review process, this upscaling approach has several limitations. Arguably, it is an analysis that is too early to perform due to the limited spatial availability of training data as the authors are interpolating across large areas. The Revised Supp. Fig. 9 shows the dispersion of the training data, and clearly, this data cannot reproduce the “real” statistical properties of the gridded data. Removing this analysis is strongly recommended, but if the authors and the editor decide to keep it, several actions are needed.
 - a) Report the model performance formally in the main text.
 - b) I recommend reorganizing lines 355-369. First, start recognizing that uncertainty is very large (>100%), that this is a preliminary analysis, and that uncertainty originated from the input data (i.e., lines 367-369). Then, all interpretations must acknowledge that the uncertainty is larger than

the estimates and, therefore, tone down the way the text is written (and the abstract). There are many sections that need correction, but one example is line 363. It currently says "..., which corresponds to a 32% increase", but it should say "..., which corresponds to a 32% increase with >XX% uncertainty". Note that the uncertainty is >100%, but the authors should properly calculate that.

- On L 334 and 337, we specifically add the model performance now, i.e. uncertainty with our results. We further reorganized the paragraph on 'Spatial patterns' (L 319 and following) according to the reviewer's suggestions.

11. Lines 388-391- Be more explicit and point the reader to an example.

- We have rephrased L 341 to be more specific about the type of mechanistic understanding our study provides.

12. Currently, the data and the code are in a Google folder, which does not follow FAIR principles. The data and code should be published following FAIR principles.

- We have deposited our data and code to Zenodo and GitHub, which should adhere to the FAIR principles. We will make it publicly available upon the expected publication date.